# MDNS: Masked Diffusion Neural Sampler via Stochastic Optimal Control

**Yuchen Zhu**[1,*], **Wei Guo**[1,*], **Jaemoo Choi**[1], **Guan-Horng Liu**[2], **Yongxin Chen**[1], **Molei Tao**[1]

[1]Georgia Institute of Technology
[2]FAIR at Meta

{yzhu738, wei.guo, jchoi843, yongchen, mtao}@gatech.edu, ghliu@meta.com

## Abstract

We study the problem of learning a neural sampler to generate samples from discrete state spaces where the target probability mass function $\pi \propto \mathrm{e}^{-U}$ is known up to a normalizing constant, which is an important task in fields such as statistical physics, machine learning, combinatorial optimization, etc. To better address this challenging task when the state space has a large cardinality and the distribution is multi-modal, we propose **M**asked **D**iffusion **N**eural **S**ampler (**MDNS**), a novel framework for training discrete neural samplers by aligning two path measures through a family of learning objectives, theoretically grounded in the stochastic optimal control of the continuous-time Markov chains. We validate the efficiency and scalability of MDNS through extensive experiments on various distributions with distinct statistical properties, where MDNS learns to accurately sample from the target distributions despite the extremely high problem dimensions and outperforms other learning-based baselines by a large margin. A comprehensive study of ablations and extensions is also provided to demonstrate the efficacy and potential of the proposed framework. Our code is available at https://github.com/yuchen-zhu-zyc/MDNS.

## 1 Introduction

Drawing samples from an unnormalized target distribution $\pi \propto \mathrm{e}^{-U}$ on some state space $\mathcal{X}_0$ is a fundamental problem with wide-ranging applications across fields such as statistical physics [LB14], Bayesian inference [Gel+13], machine learning [And+03; WJ08], etc. For decades, Markov chain Monte Carlo (MCMC) methods, such as the Langevin Monte Carlo, the Metropolis-Hastings algorithm, and the Glauber dynamics, have been a cornerstone. These methods simulate a Markov chain whose stationary distribution is the target distribution, and converge provably fast under certain circumstances [Che22]. Despite their widespread adoption, MCMC algorithms face significant challenges, especially when the state space is high-dimensional and the target distribution is multimodal, which hinders efficient exploration and convergence (see, e.g., the lower-bound results in [Ran06; GT06; GLR18; HZ25]).

Recently, inspired by the success of diffusion models in generative modeling for continuous data [HJA20; SME21; Son+21], significant progress has been made in leveraging neural networks to learn a stochastic differential equation (SDE) to generate samples from $\pi$, which we collectively refer to as **neural samplers** (e.g., [ZC22; VGD23; AVE25; Hav+25]). Concurrently, the diffusion paradigm has been effectively extended to discrete state spaces [Aus+21; Cam+22; Sun+23b; LME24], finding broad applications in domains involving sequences of categorical data. However, while neural samplers have been extensively developed for distributions on $\mathbb{R}^d$ and discrete diffusion models

---

[*]Equal contribution, random order.

have shown promise in modeling discrete data, the study of leveraging discrete diffusion models for sampling remains relatively unexplored. Addressing this gap is crucial, as numerous real-world sampling problems in areas such as statistical physics [FL24] and combinatorial optimization [Sun+23a; SHL24; San+25] necessitate specialized methods to navigate the unique challenges they pose. This work aims to develop such a specialized approach.

**Contributions.** Our contributions can be summarized in the following points.

1. We introduce **Masked Diffusion Neural Sampler** (MDNS), a novel framework for training discrete neural samplers that leverages optimal control of CTMCs and masked discrete diffusion models.

2. To cope with the optimization challenges caused by the discontinuity of CTMC trajectories, we introduce several learning objectives that are free of differentiability requirements.

3. To achieve scalable training in high dimensions, we propose a novel training loss named **Weighted Denoising Cross-entropy** (16) that directly implements score learning on the target distribution using importance sampling.

4. We conduct comprehensive experiments and ablation studies on Ising and Potts models to validate the efficacy of our method. MDNS can accurately sample from target distributions even when the state space has cardinality $10^{122}$, substantially surpassing other learning-based methods.

**Related works.** We refer readers to App. A for a detailed discussion of related works.

## 2 Preliminaries

### 2.1 Continuous-time Markov Chains

A **continuous-time Markov chain (CTMC)** $X = (X_t)_{t \in [0,T]}$ is a stochastic process defined on a probability space $(\Omega, \mathcal{F}, \Pr)$ and takes value in a finite state space $\mathcal{X}$. Its law is completely characterized by the **generator** $Q = (Q_t \in \mathbb{R}^{\mathcal{X} \times \mathcal{X}})_{t \in [0,T]}$, defined by

$$Q_t(x,y) = \lim_{\Delta t \to 0} \frac{1}{\Delta t}(\Pr(X_{t+\Delta t} = y | X_t = x) - 1_{x=y}), \tag{1}$$

where $1_A$ is the indicator function of the statement $A$, being one if $A$ holds and zero if otherwise. By definition, the generator satisfies $Q_t(x,y) \geq 0$ for $x \neq y$ and $Q_t(x,x) = -\sum_{y \neq x} Q_t(x,y)$.

The path of $X$, i.e., $t \mapsto X_t(\omega)$, is a piecewise constant and càdlàg[1] function. The **path measure** of a CTMC $X$ is a probability measure on $(\Omega, \mathcal{F})$ defined as $\mathbb{P}^X(A) := \Pr(X \in A)$, which is the distribution of $X$ on $\Omega$. We define $\mathbb{P}_t^X$ as the marginal distribution of $X_t$, and $\mathbb{P}_{t|s}^X(\cdot|x)$ as the distribution of $X_t$ conditional on $X_s = x$. The following lemma is an important result to know from the literature of CTMCs, which shows how to compute the **Radon-Nikodým (RN) derivative** between two path measures driven by CTMCs with different generators and initial distributions, and is crucial for developing our proposed sampling algorithms in (14):

**Lemma 1.** *Given two CTMCs with generators $Q^1, Q^2$ and initial distributions $\mu_1, \mu_2$ on $\mathcal{X}$, let $\mathbb{P}^1, \mathbb{P}^2$ be the associated path measures. Then for any trajectory $\xi = (\xi_t)_{t \in [0,T]}$,*

$$\log \frac{d\mathbb{P}^2}{d\mathbb{P}^1}(\xi) = \log \frac{d\mu_2}{d\mu_1}(\xi_0) + \sum_{t: \xi_{t-} \neq \xi_t} \log \frac{Q_t^2(\xi_{t-}, \xi_t)}{Q_t^1(\xi_{t-}, \xi_t)} + \int_0^T \sum_{y \neq \xi_t} (Q_t^1(\xi_t, y) - Q_t^2(\xi_t, y)) dt. \tag{2}$$

A detailed proof is provided in App. C.3. An intuitive interpretation of (2) is to view the RN derivative between path measures as the limit of density ratios between finite-dimensional joint distributions, and approximate the transition distribution by (1). We remark that for masked diffusion models, (2) can be *precisely* calculated without discretization error, as will be seen later in (14).

---

[1]Acronym of the French phrase standing for "right continuous with left limits".

## 2.2 Discrete Diffusion Models

In discrete diffusion models, one learns the generator of a CTMC that starts from a simple distribution $p_{\text{init}}$ and reaches the target distribution $p_{\text{data}}$ at the final time. This CTMC is typically chosen as the time-reversal of a noising process that converts any distribution to $p_{\text{init}}$. An especially effective subclass of discrete diffusion is the **masked discrete diffusion models** [LME24; Ou+25; Sah+24; Shi+24a; Zhe+25], which corresponds to choosing $p_{\text{init}}$ to be $p_{\text{mask}}$, the Dirac distribution on the fully masked sequence. Adding a mask token $\mathbf{M}$ into the original state space $\mathcal{X}_0 := \{1, 2, ..., N\}^D$ containing length-$D$ sequences with vocabulary size $N$, denote the mask-augmented state space by $\mathcal{X} := \{1, 2, ..., N, \mathbf{M}\}^D$. Throughout the paper, $x = (x^1, ..., x^D)$ denotes a sequence of tokens, $x^{d \leftarrow n}$ represents the sequence constructed by replacing the $d$-th position of $x$ by $n$, and $x^{\text{UM}} = (x^d : x^d \neq \mathbf{M})$ represents the unmasked part of $x \in \mathcal{X}$.

Suppose we need to model sequential data $X \in \mathcal{X}_0$ following the distribution $p_{\text{data}}$. The intuition behind the masked discrete diffusion model is to define a noising CTMC that independently and randomly masks each token according to a certain schedule, then reverse this CTMC to generate data from a sequence of pure mask tokens. It is proved in [Ou+25] that for $x \neq y \in \mathcal{X}$, the generator for the unmasking generative process enjoys a special structure and can be written as

$$Q_t(x, y) = \gamma(t) \Pr_{X \sim p_{\text{data}}} (X^d = n | X^{\text{UM}} = x^{\text{UM}}), \text{ if } x^d = \mathbf{M} \text{ and } y = x^{d \leftarrow n}, \tag{3}$$

and 0 if otherwise, for some noise schedule $\gamma : [0, T] \to [0, \infty)$ such that $\int_0^T \gamma(t) \mathrm{d}t = \infty$. In practice, practitioners typically leverage a neural network $s_\theta$ that takes a partially masked sequence $x \in \mathcal{X}$ as input and outputs $s_\theta(x) \in \mathbb{R}^{D \times N}$ whose $(d, n)$ entry approximates $\Pr_{X \sim p_{\text{data}}}(X^d = n | X^{\text{UM}} = x^{\text{UM}})$ if $x^d = \mathbf{M}$.

A standard way for training a masked discrete diffusion model given i.i.d. samples from $p_{\text{data}}$ is using the **denoising cross-entropy (DCE)** loss [Ou+25; Sah+24; Shi+24a]:

$$\min_\theta \mathbb{E}_{p_{\text{data}}(x)} \mathbb{E}_{\lambda \sim \text{Unif}(0,1)} \left[ w(\lambda) \mathbb{E}_{\mu_\lambda(\widetilde{x}|x)} \sum_{d : \widetilde{x}^d = \mathbf{M}} -\log s_\theta(\widetilde{x})_{d, x^d} \right], \tag{4}$$

where the transition kernel $\mu_\lambda(\cdot | x)$ means independently masking each entry of $x$ with probability $\lambda$, and the weight $w(\cdot)$ can be any positive function. In particular, the loss with $w(\lambda) = \frac{1}{\lambda}$ corresponds to the any-order autoregressive loss [Ou+25, Eq. (3.7)].

Sampling from a masked discrete diffusion model can be achieved through multiple schemes, such as the Euler method, variants of $\tau$-leaping [LME24; Ou+25; Ren+25b; Cam+22], uniformization [CY25], and (semi-)autoregressive sampler [Arr+25; Nie+25]. In this paper, we mainly use an *exact* sampling method known as the **random order autoregressive sampler** [Ou+25, App. J.4], implemented by choosing a uniformly random permutation of $\{1, ..., D\}$ and autoregressively unmasking each position along the permutation conditional on the observed positions.

# 3 Control-based Learning of Discrete Neural Sampler

## 3.1 Discrete Sampling with CTMCs

In this paper, we focus on the task of drawing samples from a **target distribution** $\pi(x) = \frac{1}{Z} e^{-U(x)}$ on the finite state space $\mathcal{X}_0 = \{1, 2, \ldots, N\}^D$. Here, we only have access to the potential function $U$, and the normalizing constant $Z = \sum_{x \in \mathcal{X}_0} \mathrm{e}^{-U(x)}$ is unknown. Moreover, we are interested in using CTMCs to sample from this target distribution, in the sense that we hope to find a generator $Q^* = (Q_t^*)_{t \in [0,T]}$ such that it drives a CTMC $X = (X_t)_{t \in [0,T]}$ with $X_0 \sim p_{\text{init}}$ (a readily sampleable initial distribution) to reach the target distribution at the final time $T$, i.e., $X_T \sim \pi$.

While this setup is simple and ideal, the problem is essentially ill-posed and hard to solve due to the non-uniqueness of the feasible generator $Q^*$. We thus seek to restrict the solution space by enforcing stricter constraints: finding a generator $Q^*$ such that the associated path measure $\mathbb{P}^*$ satisfies

$$\mathbb{P}^*(\xi) = \mathbb{P}^0(\xi_{[0,T)} | \xi_T) \pi(\xi_T) = \mathbb{P}^0(\xi) \frac{\mathrm{d}\pi}{\mathrm{d}\mathbb{P}_T^0}(\xi_T) := \mathbb{P}^0(\xi) \frac{1}{Z} \mathrm{e}^{r(\xi_T)}, \text{ where } r := -U - \log p_{\text{base}},$$
$$\tag{5}$$

for any trajectory $\xi = (\xi_t)_{t \in [0,T]}$. Here, $\mathbb{P}^0$ is a **reference path measure** with a known generator $Q^0$, readily sampleable initial distribution $\mathbb{P}_0^0 =: p_{\text{init}}$, and known final distribution $\mathbb{P}_T^0 =: p_{\text{base}}$.

Such a formulation has a natural connection to the **stochastic optimal control (SOC)** of CTMCs. In fact, parameterizing our candidate generator as $Q^u$ and assuming that it induces a path measure $\mathbb{P}^u$, the learning of $Q^u$ through matching $\mathbb{P}^u$ to $\mathbb{P}^*$ can be understood as controlling a CTMC with generator $Q^0$ to reach a new terminal distribution $\pi$. We demonstrate this connection in detail by considering the following SOC problem on $\mathcal{X}$:

$$\min_u \ \mathbb{E}_{X \sim \mathbb{P}^u} \left[ \int_0^T \sum_{y \neq X_t} \left( Q_t^u \log \frac{Q_t^u}{Q_t^0} - Q_t^u + Q_t^0 \right) (X_t, y) \mathrm{d}t - r(X_T) \right], \tag{6}$$

$$\text{s.t. } X = (X_t)_{t \in [0,T]} \text{ is a CTMC on } \mathcal{X} \text{ with generator } Q^u, \ X_0 \sim p_{\text{init}},$$

It can be shown that the problem (6) is equivalent to minimizing the Kullback–Leibler (KL) divergence between two path measures $\mathbb{P}^u$ and $\mathbb{P}^*$, $\mathrm{KL}(\mathbb{P}^u \| \mathbb{P}^*) = \mathbb{E}_{\mathbb{P}^u} \log \frac{\mathrm{d}\mathbb{P}^u}{\mathrm{d}\mathbb{P}^*}$. Define the **value function** as

$$-V_t(x) = \inf_u \mathbb{E}_{X \sim \mathbb{P}^u} \left[ \int_t^T \sum_{y \neq X_s} \left( Q_s^u \log \frac{Q_s^u}{Q_s^0} - Q_s^u + Q_s^0 \right) (X_s, y) \mathrm{d}s - r(X_T) \, \middle| \, X_t = x \right]. \tag{7}$$

In order to guarantee the existence and uniqueness of the solution to (6) for any $r$ and $p_{\text{init}}$, we need to choose a reference path measure $\mathbb{P}^0$ that is **memoryless** [DE+25], i.e., $X_0$ and $X_T$ are independent for $X \sim \mathbb{P}^0$. In this case, the unique optimal solution $Q^*$ can be expressed as a multiplicative perturbation of the reference generator $Q^0$:

$$Q_t^*(x, y) = Q_t^0(x, y) \exp(V_t(y) - V_t(x)), \quad \forall x \neq y. \tag{8}$$

Moreover, the optimal path measure $\mathbb{P}^*$ associated with $Q^*$ has the identical form as (5): $\frac{\mathrm{d}\mathbb{P}^*}{\mathrm{d}\mathbb{P}^0}(\xi) = \frac{1}{Z} \mathrm{e}^{r(\xi_T)}, \forall \xi$, where $Z = \mathbb{E}_{\mathbb{P}_T^0} \mathrm{e}^r$. These results can be shown using Lem. 1 and standard SOC theory, and we include the proofs in App. C.2 for self-consistency and better readability.

## 3.2 Optimal Control Formulation of Discrete Neural Sampler Training

As discussed in the previous section, if we manage to learn the generator $Q^u$ that produces a path measure $\mathbb{P}^u$ matching $\mathbb{P}^*$, we can sample from the target distribution by simulating the CTMC with this generator. However, minimizing $\mathrm{KL}(\mathbb{P}^u \| \mathbb{P}^*)$ is not the only available choice of objective to reach the goal. In fact, it has some disadvantages due to the inherent discontinuous nature of the problem. Instead, we propose the following general framework for learning the discrete neural sampler:

> **Framework:** Given a target distribution $\pi \propto \mathrm{e}^{-U}$ on $\mathcal{X}_0$, a choice of reference path measure $\mathbb{P}^0$ with generator $Q^0$, learn the generator $Q^u$ to sample from $\pi$ through optimizing an efficiently estimable objective $\mathcal{F}(\mathbb{P}^u, \mathbb{P}^*)$, where $\mathbb{P}^*$ is given in (5) and $Q^* = \mathrm{argmin}_{Q^u} \mathcal{F}(\mathbb{P}^u, \mathbb{P}^*)$.

It is straightforward to see that the problem (6) is a special case of the proposed framework upon choosing $\mathcal{F}(\mathbb{P}^u, \mathbb{P}^*) = \mathrm{KL}(\mathbb{P}^u \| \mathbb{P}^*)$. However, although directly relevant to many theoretical results, naively optimizing the discretized, estimated objective in (6) is not an appropriate approach for learning the desired controlled generator $Q^u$. The estimation of (6) requires the simulation of CTMC using a neural network parameterized generator with parameter $\theta$, yet the objective itself is inherently non-differentiable with respect to $\theta$ since (1) the trajectories of CTMCs are pure jump processes, and (2) the reward function $r$ is non-differentiable, making it difficult to effectively train the generator.

A workaround is proposed in [Wan+25a], where the authors retain the differentiability of the objectives through relaxing the CTMC trajectories from staying on $\mathcal{X}_0$ to sequences of probability vectors on $\mathbb{R}^{D \times N}$, using the Gumbel softmax trick. This partly solves the problem at the cost of introducing a high approximation error into the learning process, since the $\theta$ gradients coming from the Gumbel softmax trick are known to be biased and easily cause numerical instability [JGP17; Liu+23]. This leads to failure to converge to the correct target distribution, even in low dimensions, as empirically validated in our experiments (see App. D.5). Moreover, backpropagation over the entire trajectory is a memory-intensive operation, and we would like to avoid it as much as possible. In the following, we propose several alternative learning objectives that operate without these disadvantages.

**Relative-entropy with REINFORCE (RERF).** One of the key reasons that $\mathrm{KL}(\mathbb{P}^u \| \mathbb{P}^*)$ is intractable for direct optimization originates from the fact that the expectation is with respect to $\mathbb{P}^u$, which tracks the $\theta$ gradient. Alternatively, we can estimate $\nabla_\theta \mathrm{KL}(\mathbb{P}^u \| \mathbb{P}^*)$ by introducing a gradient-nontracking trajectory simulated using $\bar{u} = \mathtt{stopgrad}\,(u)$, also known as the REINFORCE trick [Wil92; RGB14; MG14]. We denote the path measure produced with $\bar{u}$ to be $\mathbb{P}^{\bar{u}}$[2], and express the gradient of the relative-entropy by the following identity: $\nabla_\theta \mathrm{KL}(\mathbb{P}^u \| \mathbb{P}^*) = \nabla_\theta \mathcal{F}_{\mathrm{RERF}}(\mathbb{P}^u, \mathbb{P}^*)$, where

$$\mathcal{F}_{\mathrm{RERF}}(\mathbb{P}^u, \mathbb{P}^*) := \mathbb{E}_{\mathbb{P}^{\bar{u}}} \log \frac{\mathrm{d}\mathbb{P}^*}{\mathrm{d}\mathbb{P}^u} \left( \log \frac{\mathrm{d}\mathbb{P}^*}{\mathrm{d}\mathbb{P}^{\bar{u}}} + C \right), \ \forall C \in \mathbb{R}. \tag{9}$$

See App. C.3 for a detailed derivation. Thus, $\mathcal{F}_{\mathrm{RERF}}$ can be introduced for the purpose of gradient estimation of relative entropy. We remark that, in contrast to the approach proposed in [Wan+25a], (9) yields an *unbiased estimator* of the relative-entropy gradient, which is crucial for accurate learning of the target distribution. However, $\mathcal{F}_{\mathrm{RERF}}(\mathbb{P}^u, \mathbb{P}^*)$ is not a valid "loss function" but only a computational surrogate, in the sense that a reduction in the objective value does not guarantee an improvement in learning performance.

**Log-variance (LV).** Other than KL divergence on path measures, we can also consider the variance type of losses, such as Log-variance (LV). Originally proposed in [NR21] to solve SOC problems for SDEs on $\mathbb{R}^d$, LV considers minimizing the following $v$-dependent objective

$$\mathcal{F}_{\mathrm{LV}}(\mathbb{P}^u, \mathbb{P}^*) := \mathrm{Var}_{\mathbb{P}^v} \log \frac{\mathrm{d}\mathbb{P}^*}{\mathrm{d}\mathbb{P}^u}, \tag{10}$$

where $v$ for now is generic and $\mathbb{P}^v$ is a chosen sampling path measure driven by the generator $Q^v$ that does not require gradient backpropagation. Note that the optimality of the solution (10) is guaranteed under weak regularity assumptions on $\mathbb{P}^v$.[3] In practice, we choose $v = \bar{u}$ as in [NR21] for algorithmic effectiveness. (10) can be efficiently computed for CTMCs leveraging Lem. 1 and (5).

**Cross-entropy (CE).** While the aforementioned RERF and LV objectives have an optimality guarantee of the solution, they often do not enjoy optimization guarantees as in general these objectives have a nonconvex landscape. To mitigate it, we consider cross-entropy between $\mathbb{P}^u$ and $\mathbb{P}^*$,

$$\mathcal{F}_{\mathrm{CE}}(\mathbb{P}^u, \mathbb{P}^*) := \mathrm{KL}(\mathbb{P}^* \| \mathbb{P}^u) = \mathbb{E}_{\mathbb{P}^*} \log \frac{\mathrm{d}\mathbb{P}^*}{\mathrm{d}\mathbb{P}^u} = \mathbb{E}_{\mathbb{P}^v} \frac{\mathrm{d}\mathbb{P}^*}{\mathrm{d}\mathbb{P}^v} \log \frac{\mathrm{d}\mathbb{P}^*}{\mathrm{d}\mathbb{P}^u}. \tag{11}$$

$\mathcal{F}_{\mathrm{CE}}(\mathbb{P}^u, \mathbb{P}^*)$ is *convex* in $\mathbb{P}^u$ due to the convexity of $t \mapsto -\log t$, thus enjoying a benign optimization landscape. However, since $\mathbb{P}^*$ is not directly tractable for simulation, we need to introduce an auxiliary sampling path measure $\mathbb{P}^v$ and equivalently express the objective based on $\mathbb{P}^v$. This can be understood as an importance sampling estimation of the original objective, and in practice we use $v = \bar{u}$.

To sum up, the RERF, LV, and CE losses do not involve the error for approximating the discrete variables by continuous ones, do not require backpropagation over the entire trajectory of states, and thus are more preferable for efficient and stable optimization.

### 3.3 Masked Diffusion Neural Sampler

Besides the objective $\mathcal{F}(\mathbb{P}^u, \mathbb{P}^*)$, another major component in the design space of the proposed training framework is the choice of reference path measure $\mathbb{P}^0$ and the corresponding generator $Q^0$. Recall from Sec. 3.1 that we need the reference path measure to be memoryless to guarantee the existence and uniqueness of the SOC problem (6). We now introduce a method for choosing such a memoryless reference path measure based on a masked discrete diffusion model. This approach, which we term the **M**asked **D**iffusion **N**eural **S**ampler (**MDNS**), forms the core of our framework. The corresponding learning algorithms are subsequently developed based on the objectives proposed in Sec. 3.2. Furthermore, we demonstrate that this framework can be extended to incorporate uniform discrete diffusion models, an adaptation referred to as the **U**niform **D**iffusion **N**eural **S**ampler (**UDNS**), with a detailed discussion deferred to App. F.

We choose $\mathbb{P}^0$ to be the generative process of a masked discrete diffusion model, starting from $p_{\mathrm{init}} \leftarrow p_{\mathrm{mask}}(x) := 1_{x=(\mathbf{M},\dots,\mathbf{M})}$ and terminating at $p_{\mathrm{data}} \leftarrow p_{\mathrm{unif}}(x) := \frac{1}{N^D} 1_{x \in \mathcal{X}_0}$, the uniform

---

[2]This means we sample a trajectory $X \sim \mathbb{P}^u$ and detach it from the computational graph. After that, use the values of the trajectory to compute the objective and do backpropagation.

[3]As long as $\mathbb{P}^v$ is supported almost everywhere on the space of paths.

---

**Algorithm 1** Training of Masked Diffusion Neural Sampler (MDNS)

---

**Require:** score model $s_\theta$, batch size $B$, training iterations $K$, reward $r : \mathcal{X}_0 \to \mathbb{R}$, learning objective
  $\mathcal{F} \in \{\mathcal{F}_{\mathrm{RERF}}, \mathcal{F}_{\mathrm{LV}}, \mathcal{F}_{\mathrm{CE}}, \mathcal{F}_{\mathrm{WDCE}}\}$, (num. replicates $R$, resample frequency $k$ for $\mathcal{F}_{\mathrm{WDCE}}$).
  1: **for** step $= 1$ **to** $K$ **do**
  2:    **if** $\mathcal{F} \in \{\mathcal{F}_{\mathrm{RERF}}, \mathcal{F}_{\mathrm{LV}}, \mathcal{F}_{\mathrm{CE}}\}$ **then**
  3:       $\{X^{(i)}, W^u(X^{(i)})\}_{1 \le i \le B} = \texttt{Sample\_Trajectories}(B).$       $\triangleright$ See Alg. 2 for details.
  4:       Compute $\mathcal{F}$ with $\{X^{(i)}, W^u(X^{(i)})\}_{1 \le i \le B}$.                      $\triangleright$ See (15).
  5:    **else if** $\mathcal{F} = \mathcal{F}_{\mathrm{WDCE}}$ **then**
  6:       **if** step $\bmod\ k = 0$ **then**                      $\triangleright$ Sample new trajectories every $k$ steps.
  7:          $\{X^{(i)}, W^u(X^{(i)})\}_{1 \le i \le B} = \texttt{Sample\_Trajectories}(B).$
  8:          Set replay buffer $\mathcal{B} \leftarrow \{X^{(i)}, W^u(X^{(i)})\}_{1 \le i \le B}.$
  9:       $\{\widetilde{X}^{(i)}, W^u(\widetilde{X}^{(i)})\}_{1 \le i \le BR} = \texttt{Resample\_with\_Mask}(\mathcal{B}; R).$  $\triangleright$ See App. B for details.
 10:       Compute $\mathcal{F}_{\mathrm{WDCE}}$ with $\{\widetilde{X}^{(i)}, W^u(\widetilde{X}^{(i)})\}_{1 \le i \le BR}.$
 11:    Update the parameters $\theta$ based on the gradient $\nabla_\theta \mathcal{F}.$
      **return** trained score model $s_\theta$.

---

distribution on the unmasked data space $\mathcal{X}_0$. Based on (3), the corresponding generator is

$$Q_t^0(x, x^{d \leftarrow n}) = \gamma(t) \Pr_{X \sim p_{\mathrm{unif}}} (X^d = n | X^{\mathrm{UM}} = x^{\mathrm{UM}}) 1_{x^d = \mathbf{M}} = \frac{\gamma(t)}{N} 1_{x^d = \mathbf{M}}. \tag{12}$$

One can prove the following lemma (see App. C.3 for the proof):

**Lemma 2.** *Under the assumption that $\int_0^T \gamma(t)\mathrm{d}t = \infty$, the reference path measure $\mathbb{P}^0$ with generator $Q^0$ defined in (12) and starting from $p_{\mathrm{mask}}$ is memoryless and satisfies $\mathbb{P}_T^0 = p_{\mathrm{unif}}$.*

With such a choice for $\mathbb{P}^0$, the optimal generator $Q^*$ also has a special structure:

**Lemma 3.** *The generator $Q^*$ corresponding to the optimal solution of (6) with $Q^0$ defined as the memoryless reference path measure (12) satisfies*

$$Q_t^*(x, x^{d \leftarrow n}) = \gamma(t) \Pr_{X \sim \pi} (X^d = n | X^{\mathrm{UM}} = x^{\mathrm{UM}}) 1_{x^d = \mathbf{M}}. \tag{13}$$

See App. C.3 for the proof. This suggests that it suffices to parameterize $Q_t^u(x, x^{d \leftarrow n}) = \gamma(t) s_\theta(x)_{d,n} 1_{x^d = \mathbf{M}}$. Here, $s_\theta : \mathcal{X} \to \mathbb{R}^{D \times N}$ is parameterized to have non-negative entries and each row sums up to 1, representing the one-dimensional marginals of the target distribution $\pi$ conditioned on unmasked entries of the input $x$. Moreover, such parameterization implies that the diagonal entries of $Q_t^u$ are

$$\sum_{y \neq x} Q_t^u(x, y) = \sum_{d : x^d = \mathbf{M}} \sum_n Q_t^u(x, x^{d \leftarrow n}) = \gamma(t) \sum_{d : x^d = \mathbf{M}} 1 = \gamma(t)|\{d : x^d = \mathbf{M}\}|,$$

where the first equality is due to (3) and the second equality is due to the parameterization of $Q_t^u$. Similarly, one can also verify that $\sum_{y \neq x} Q_t^0(x, y) = \gamma(t)|\{d : x^d = \mathbf{M}\}|$. This turns out to significantly simplify the expression of the RN derivative, as will be shown later.

With the results above, we use Lem. 1 to derive the RN derivative between the optimal and the current path measures $\mathbb{P}^*$ and $\mathbb{P}^u$, the common term for computing $\mathcal{F}_{\mathrm{RERF}}$, $\mathcal{F}_{\mathrm{LV}}$, and $\mathcal{F}_{\mathrm{CE}}$:

$$\log \frac{\mathrm{d}\mathbb{P}^*}{\mathrm{d}\mathbb{P}^u}(X) = r(X_T) - \log Z + \sum_{t : X_{t-} \neq X_t} \log \frac{Q_t^0}{Q_t^u}(X_{t-}, X_t) + \int_0^T \sum_{y \neq X_t} (Q_t^u - Q_t^0)(X_t, y)\mathrm{d}t$$

$$= r(X_T) + \sum_{t : X_{t-} \neq X_t} \log \frac{1/N}{s_\theta(X_{t-})_{d(t), X_t^{d(t)}}} - \log Z =: W^u(X) - \log Z,$$

where $W^u(X) = r(X_T) + \sum_{t : X_{t-} \neq X_t} \log \frac{1/N}{s_\theta(X_{t-})_{d(t), X_t^{d(t)}}}. \tag{14}$

Here, we assume that the jump from $X_{t-}$ to $X_t$ is at the $d(t)$-th position, and the total number of jumps is $D$. With (14), the aforementioned training objectives (9) to (11) are simplified to

$$\mathcal{F}_{\text{RERF}} = \mathop{\mathbb{E}}_{X \sim \mathbb{P}^{\bar{u}}} W^{\bar{u}}(X) W^u(X), \quad \mathcal{F}_{\text{LV}} = \mathop{\text{Var}}_{X \sim \mathbb{P}^{\bar{u}}} W^u(X), \quad \mathcal{F}_{\text{CE}} = \mathop{\mathbb{E}}_{X \sim \mathbb{P}^{\bar{u}}} \frac{1}{Z} e^{W^{\bar{u}}(X)} W^u(X), \quad (15)$$

where we have removed terms related to $Z$ in $\mathcal{F}_{\text{RERF}}$ and $\mathcal{F}_{\text{LV}}$ without modifying the optimization landscape (in particular, $C$ is chosen as $\log Z$ in (9)). For the CE loss, as in practice the normalizing constant $Z$ may be prohibitively large, removing $\frac{1}{Z}$ from the loss may lead to numerical instability. To avoid this, we propose to estimate $Z$ via the equality $Z = \mathbb{E}_{X \sim \mathbb{P}^{\bar{u}}} e^{W^{\bar{u}}(X)}$ implied by (14), which is equivalent to applying softmax to the weights $\{W^{\bar{u}}(X^{(i)})\}_{1 \leq i \leq B}$ in a batch.

**Scalable training via weighted denoising cross-entropy.** While the learning objectives in (15) successfully avoid backpropagation along all the states $X$ and are thus relatively memory efficient, their implementation still wastes much compute. Note that for computing $W^u(X)$, we need to call the model $s_\theta(\cdot)$ $D$ times, but each time only the $(d(t), X_t^{d(t)})$-th element of the $D \times N$ output matrix is used in optimization. What's worse, the three losses in (15) require backpropagation through all of the $D$ gradient-tracking score outputs in (14), which may be unaffordable due to the GPU memory constraints. To propose a more scalable loss for high-dimensional data, we start with a further simplification of $\mathcal{F}_{\text{CE}}$ by discarding the $u$-independent terms in $W^u(X)$:

$$\min_u \mathbb{E}_{X \sim \mathbb{P}^{\bar{u}}} \frac{1}{Z} e^{W^{\bar{u}}(X)} W^u(X) = \min_u \mathbb{E}_{X \sim \mathbb{P}^{\bar{u}}} \frac{1}{Z} e^{W^{\bar{u}}(X)} \sum_{t: X_{t-} \neq X_t} - \log s_\theta(X_{t-})_{d(t), X_t^{d(t)}}.$$

Here, the key idea is to treat i.i.d. samples from $\mathbb{P}^u$ as *importance weighted* samples from $\mathbb{P}^*$. Instead of only learning the conditional distribution $s_\theta$ on a set of positions and values on the generation trajectory, we can actually forget about the trajectory and only focus on the achieved clean sample $X_T$. By remasking $X_T$ and computing the DCE loss in (4), we arrive at the following **weighted denoising cross-entropy (WDCE) loss**:

$$\mathcal{F}_{\text{WDCE}}(\mathbb{P}^u, \mathbb{P}^*) := \mathop{\mathbb{E}}_{X \sim \mathbb{P}^{\bar{u}}} \left[ \frac{1}{Z} e^{W^{\bar{u}}(X)} \mathop{\mathbb{E}}_{\lambda \sim \text{Unif}(0,1)} \left[ w(\lambda) \mathop{\mathbb{E}}_{\mu_\lambda(\widetilde{x}|X_T)} \sum_{d: \widetilde{x}^d = \mathbf{M}} - \log s_\theta(\widetilde{x})_{d, X_T^d} \right] \right], \tag{16}$$

where we estimate $Z$ by $\mathbb{E}_{X \sim \mathbb{P}^{\bar{u}}} e^{W^{\bar{u}}(X)}$ as in $\mathcal{F}_{\text{CE}}$. It is straightforward to note that (16) is equivalent to the DCE loss (4) with $\pi$ being the data distribution $p_{\text{data}}$, except that $X_T$ are now importance weighted instead of i.i.d. samples from $\pi$. $\mathcal{F}_{\text{WDCE}}$ is much more efficient than $\mathcal{F}_{\text{CE}}$ in that we can use all the output of the score model $s_\theta(\cdot)$ to compute the loss, instead of only one element. Moreover, for each pair of $(X_T, e^{W^{\bar{u}}(X)})$, we can sample *multiple* (say, $R$) partially masked $\tilde{x}$ to compute the loss, so the expensive $O(D)$ computation of the RN derivative can be amortized. Inspired by recent works for reusing samples [Mid+23; Hav+25], we can also create a *replay buffer* $\mathcal{B}$ that stores pairs of final samples and weights $(X_T, e^{W^{\bar{u}}(X)})$ for multiple steps of optimization for further reduction of the computational cost. Our algorithm is summarized in Alg. 1.

### 3.4 Theoretical Guarantees

We can establish the following theoretical guarantees for our learning algorithms (proof in App. C.3):

**Proposition 1** (Guarantee for sampling). *Let $p_{\text{samp}} := \mathbb{P}_T^u$ be the distribution of the samples generated from the learned sampler. Then, to ensure $\text{KL}(p_{\text{samp}} \| \pi) \leq \varepsilon^2$ (resp., $\text{KL}(\pi \| p_{\text{samp}}) \leq \varepsilon^2$), it suffices to train the sampler to reach $\text{KL}(\mathbb{P}^u \| \mathbb{P}^*) \leq \varepsilon^2$ (resp., $\text{KL}(\mathbb{P}^* \| \mathbb{P}^u) \leq \varepsilon^2$).*

**Proposition 2** (Guarantee for normalizing constant estimation). $\widehat{Z} := e^{W^u(X)}$, $X \sim \mathbb{P}^u$ *is an unbiased estimation of $Z$. To guarantee $\Pr\left( \left| \frac{\widehat{Z}}{Z} - 1 \right| \leq \varepsilon \right) \geq \frac{3}{4}$, it suffices to train the sampler to reach $\min\{\text{KL}(\mathbb{P}^u \| \mathbb{P}^*), \text{KL}(\mathbb{P}^* \| \mathbb{P}^u)\} \leq \frac{\varepsilon^2}{2}$. One can boost the probability from $\frac{3}{4}$ to $1 - \zeta$ for any $\zeta \in \left(0, \frac{1}{4}\right)$ by taking the median of $O\left(\log \frac{1}{\zeta}\right)$ i.i.d. estimates.*

## 4 Experiments

In this section, we experimentally validate our proposed frameworks by learning to sample from the Ising model and Potts model on square lattices. At different temperatures, these models exhibit

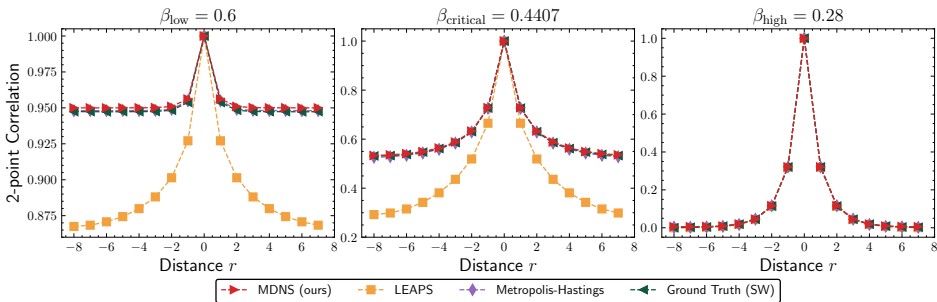

Figure 1: Average of 2-point correlation $C^{\mathrm{row}}(k, k+r)$ of samples from $16 \times 16$ Ising model.

distinct behaviors, providing a rich ground for demonstrating the effectiveness of our algorithms. We use probabilistic metrics (KL divergence, TV distance, $\chi^2$ divergence, etc.) and observables (magnetization, 2-point correlation) originating from statistical physics to benchmark our learning results. We include both learning-based approaches such as LEAPS [HAJ25], and learning-free approaches such as the Metropolis-Hastings (MH) algorithm as benchmarks. For fair comparisons, we run all baselines with a long enough but comparable compute time. To accurately estimate ground truth values for all the observables when an exact computation is intractable, we draw samples by running the Swendsen-Wang (SW) algorithm [SW86; SW87] for a sufficiently long time. We include experiment details such as metric definitions and training hyperparameters in Apps. D and E.

## 4.1 Ising model on Square Lattice

We consider learning to sample from the Ising model on a square lattice with $L$ sites per dimension, $\Lambda = \{1, ..., L\}^2$. The state space is $\mathcal{X}_0 = \{\pm 1\}^\Lambda$, where $\pm 1$ represent the two spin states. Given an interaction parameter $J \in \mathbb{R}$, an external magnetic field $h \in \mathbb{R}$, and an inverse temperature $\beta > 0$, the probability distribution of any configuration $x \in \mathcal{X}_0$ is given by

$$\pi(x) = \frac{1}{Z} \mathrm{e}^{-\beta H(x)}, \text{ where } H(x) = -J \sum_{i \sim j} x^i x^j - h \sum_i x^i \text{ and } Z = \sum_{x \in \mathcal{X}_0} \mathrm{e}^{-\beta H(x)}. \quad (17)$$

In the above equation, $i \sim j$ means that $i, j \in \Lambda$ are adjacent on the lattice. For simplicity, we impose periodic boundary conditions in both the horizontal and vertical directions.

**A case study on the choice of $\mathcal{F}$.** We first compare the effects of different learning objectives $\mathcal{F}$ by learning to sample from a $4 \times 4$ high temperature Ising model with $J = 1$, $h = 0.1$, and $\beta_{\mathrm{high}} = 0.28$. The system size is picked so that the sampling problem is not trivial, but an exact computation of the partition function remains computationally tractable. The results are reported in Tab. 2, and results at other temperatures are available in App. D. For a fair comparison, we train for 1000 steps with a batch size of 256 for each loss, and use $R = 16$ resampling replicates for $\mathcal{F}_{\mathrm{WDCE}}$ so that the effective batch size is the same across all objectives. We draw a sufficiently large number ($2^{20}$) of samples from the learned model to estimate all the reported metrics. We also report the average **effective sample size** (**ESS**, defined in (24)) of the last 100 steps to showcase the learning speed for each objective. We include a baseline produced by running the MH algorithm for sufficiently long as a reference. We can see that all four learning objectives can learn a good model that generates ground-truth-like samples. In all the evaluation metrics except the absolute error of $\log \widehat{Z}$, the effectiveness ranking is uniformly $\mathcal{F}_{\mathrm{LV}} > \mathcal{F}_{\mathrm{WDCE}} > \mathcal{F}_{\mathrm{RERF}} > \mathcal{F}_{\mathrm{CE}}$.

**Scaling to higher dimensions and lower temperatures.** We also run on significantly harder settings by increasing the system size to $16 \times 16$ (so the cardinality of the state space is $2^{256}$). We keep $J = 1, h = 0$ and vary the inverse temperature, ranging from $\beta_{\mathrm{high}} = 0.28$, $\beta_{\mathrm{critical}} = \log(1 + \sqrt{2})/2 = 0.4407$, to $\beta_{\mathrm{low}} = 0.6$. The behaviors of the Ising model are known to be distinct at these temperatures by the phase transition theory [KW41; Ons44]. For the purpose of efficient training, we use $\mathcal{F}_{\mathrm{WDCE}}$ as learning objectives and train on each problem for 50k steps. In Tab. 1, we report the ESS and the **absolute error of magnetization** and **2-point correlation** (respectively defined in (26) and (28)) to the estimated ground truth. We also plot the estimated average 2-point correlation (defined in (27)) between a varying distance $r$ in Fig. 1, and defer further results to

Table 1: Results for learning $16 \times 16$ Ising model at different temperatures, best in **bold**. Mag. and Corr. represent the absolute error of the magnetization and 2-point correlation to ground truth values.

| Temperature | $\beta_{\text{low}} = 0.6$ | | | $\beta_{\text{critical}} = 0.4407$ | | | $\beta_{\text{high}} = 0.28$ | | |
|---|---|---|---|---|---|---|---|---|---|
| Metrics | Mag. ↓ | Corr. ↓ | ESS ↑ | Mag. ↓ | Corr. ↓ | ESS ↑ | Mag. ↓ | Corr. ↓ | ESS ↑ |
| MDNS (ours) | **9.9e−3** | 2.4e−3 | **0.981** | **3.7e−3** | **2.0e−3** | **0.933** | 8.5e−3 | **1.0e−3** | 0.962 |
| LEAPS | 2.4e−2 | 5.8e−1 | 0.261 | 7.4e−3 | 1.6e−1 | 0.384 | 7.4e−3 | 1.6e−3 | **0.987** |
| Baseline (MH) | 1.9e−2 | **7.7e−4** | / | 4.6e−3 | 2.5e−3 | / | **6.1e−3** | 1.1e−3 | / |

Table 2: Results for learning $4 \times 4$ Ising model with $J = 1$, $h = 0.1$, and $\beta_{\text{high}} = 0.28$, best in **bold**.

| Method | ESS ↑ | $\text{TV}(\widehat{p}_{\text{samp}}, \pi) \downarrow$ | $\text{KL}(\widehat{p}_{\text{samp}} \| \pi) \downarrow$ | $\chi^2(\widehat{p}_{\text{samp}} \| \pi) \downarrow$ | $\widehat{\text{KL}}(\mathbb{P}^u \| \mathbb{P}^*) \downarrow$ | Abs. err. of $\log \widehat{Z} \downarrow$ |
|---|---|---|---|---|---|---|
| $\mathcal{F}_{\text{RERF}}$ | 0.9621 | 0.0799 | 0.0380 | 0.0845 | 0.0188 | **0.00003** |
| $\mathcal{F}_{\text{LV}}$ | **0.9713** | **0.0748** | **0.0348** | **0.0714** | **0.0141** | 0.00046 |
| $\mathcal{F}_{\text{CE}}$ | 0.9513 | 0.0833 | 0.0393 | 0.0903 | 0.0248 | 0.00099 |
| $\mathcal{F}_{\text{WDCE}}$ | 0.9644 | 0.0799 | 0.0382 | 0.0868 | 0.0177 | 0.00030 |
| Baseline (MH) | / | 0.0667 | 0.0325 | 0.0628 | / | / |

App. D.3. Our learned distribution accurately matches the theoretically predicted trend of correlation functions, which exponentially decays to the non-zero spontaneous magnetization at $\beta_{\text{low}}$, polynomial decays at $\beta_{\text{critical}}$ and exponential decays to 0 at $\beta_{\text{high}}$. As shown in the table and figure, MDNS outperforms LEAPS by a large margin across almost all metrics. MDNS accurately learns the high-dimensional distribution across all temperatures, as evidenced by low observables errors and high ESS values, whereas LEAPS fails at $\beta_{\text{critical}}$ and $\beta_{\text{low}}$ despite long training.

**Warm-up for lower temperatures.** Instead of only training from scratch, our training includes a **warm-up phase**. When targeting hard distributions such as $\beta_{\text{critical}}$ and $\beta_{\text{low}}$, to help the model better locate the modes of the target distribution, we start the training by fitting easier ones such as $\beta_{\text{high}}$ for a short (20k steps in this example) period of time. An ablation study to demonstrate the importance of this proposed technique is presented in App. D.2. A more systematic study on the warm-up is deferred to future work [Guo+25].

**Preconditioning.** In learning diffusion samplers for a continuous distribution $\nu \propto \text{e}^{-V}$ on $\mathbb{R}^d$, one typically leverages the score information $\nabla \log \nu$ in the neural network architecture to guide the sampling process for faster convergence and better sampling quality, known as preconditioning. We also explore similar techniques to improve the training of MDNS, see App. D.4 for details.

### 4.2 Potts model on Square Lattice

We now consider a harder problem known as the standard Potts model, which has $q (\geq 2)$ spin states on a square lattice with $L$ sites per dimension, $\Lambda = \{1, ..., L\}^2$, i.e., the state space is $\mathcal{X}_0 = \{1, 2, ..., q\}^\Lambda$. Given an interaction parameter $J \in \mathbb{R}$ and an inverse temperature $\beta > 0$, the probability distribution of any configuration $x \in \mathcal{X}_0$ is given by

$$\pi(x) = \frac{1}{Z} \text{e}^{-\beta H(x)}, \text{ where } H(x) = -J \sum_{i \sim j} 1_{x^i = x^j} \text{ and } Z = \sum_{x \in \mathcal{X}_0} \text{e}^{-\beta H(x)}. \tag{18}$$

In the above equation, $i \sim j$ also means that $i, j \in \Lambda$ are adjacent on the lattice, and we impose the same periodic boundary conditions. As a generalization of the Ising model, the Potts model also exhibits diverse behaviors across different temperatures, and the critical inverse temperature is known to be $\beta_{\text{critical}} = \log(1 + \sqrt{q})$ [BDC12]. In the following experiment, we consider the system size to be $L = 16$, fix $q = 3$ and $J = 1$, and vary the inverse temperature $\beta$, ranging from $\beta_{\text{high}} = 0.5$, $\beta_{\text{critical}} = \log(1 + \sqrt{3}) = 1.005$, to $\beta_{\text{low}} = 1.2$.

Similar to the training for the Ising model, we use $\mathcal{F}_{\text{WDCE}}$ and train on each temperature for 100k steps, among which 30k steps are in the warm-up phase for training $\beta_{\text{critical}}$ and $\beta_{\text{low}}$. We report quantitative metrics (defined in (30) and (32)) in Tab. 3 and similarly plot the estimated average 2-point correlation (defined in (31) between a varying distance $r$ in Fig. 2. More results can be

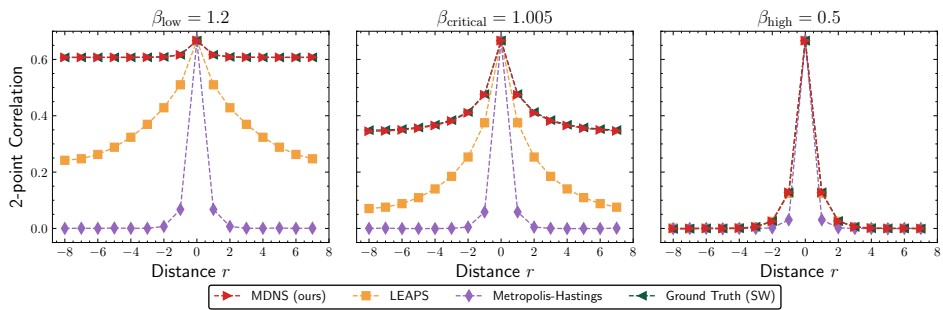

Figure 2: Average of 2-point correlation $C^{\mathrm{row}}(k, k+r)$ of samples from $16 \times 16$ Potts model.

Table 3: Results for learning $16 \times 16$ Potts model at different temperatures, best in **bold**. Mag. and Corr. represent absolute errors of magnetization and 2-point correlation to ground truth values.

| Temperature | $\beta_{\mathrm{low}} = 1.2$ | | | $\beta_{\mathrm{critical}} = 1.005$ | | | $\beta_{\mathrm{high}} = 0.5$ | | |
|---|---|---|---|---|---|---|---|---|---|
| Metrics | Mag. ↓ | Corr. ↓ | ESS ↑ | Mag. ↓ | Corr. ↓ | ESS ↑ | Mag. ↓ | Corr. ↓ | ESS ↑ |
| MDNS (ours) | **1.3e−3** | **8.8e−5** | **0.933** | **4.3e−3** | **2.9e−3** | **0.875** | **2.2e−3** | **5.8e−4** | 0.983 |
| LEAPS | 2.9e−1 | 2.5e−1 | 0.012 | 2.7e−1 | 2.0e−1 | 0.004 | 2.9e−3 | 1.2e−3 | **0.991** |
| Baseline (MH) | 7.4e−1 | 5.6e−1 | / | 5.2e−1 | 3.5e−1 | / | 3.5e−2 | 1.6e−2 | / |

found in App. E. Again, our approach recovers a distribution with a correlation function whose decay patterns are well aligned with the ground truth, whereas other baselines fail to do so. It is worth noting that the MH algorithm cannot mix even after a continuous simulation of more than 20 hours, while it remains a successful approach for all Ising model experiments. Such a difference likely originates from the exponential increase in the state space cardinality $|\mathcal{X}_0|$ despite the system size $L$ remaining the same. For example, a $4 \times 4$ Ising model has around 60k distinct states, while a $4 \times 4$ Potts model with $q = 3$ has 40M distinct states, which clearly highlights the increased difficulty of Potts model sampling. Nevertheless, our approach succeeds with a moderate increase in training iterations. Moreover, the evaluation shows that MDNS performs best across almost all metrics and temperatures, suggesting that our framework and learning objectives are highly scalable.

## 5   Conclusion, Limitations and Future Directions

This paper introduces Masked Diffusion Neural Sampler (MDNS), a novel framework for training discrete neural samplers based on stochastic optimal control and masked diffusion models. While MDNS has proven effective for distributions arising in statistical physics, its performance on other families of discrete distributions, such as those on graphs or related to combinatorial optimization, is unknown. We also propose that the framework can be extended to fine-tune pretrained discrete diffusion models given a (possibly non-differentiable) reward function, which is left for future work [Zhu+25d]. Theoretically, we conjecture that our interpolation between the masked and the target distribution possesses superior properties compared to the geometric annealing $(\pi_\eta \propto \mathrm{e}^{-\eta U})_{\eta \in [0,1]}$ used in LEAPS [HAJ25]. A rigorous theoretical analysis of annealing paths in discrete spaces, potentially building upon insights in continuous spaces [GTC25b; GTC25a], remains a key area for future investigation.

## Acknowledgments and Disclosure of Funding

WG and MT thank Peter Holderrieth and Michael Albergo for insightful discussions on the paper [HAJ25] during ICLR 2025. YZ and MT are grateful for partial support by NSF Grants DMS-1847802, DMS-2513699, DOE Grants NA0004261, SC0026274, and Richard Duke Fellowship. WG, JC, and YC acknowledge support from NSF Grants ECCS-1942523, DMS-2206576, and CMMI-2450378.

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

## A    Related Works

**Neural samplers.**    Recently, a growing trend of research directions has been focusing on training neural samplers [Noé+19; He+25], where one amortizes the sampling procedure through learning a neural network. Most of the work has focused on learning distributions over the continuous Euclidean space $\mathbb{R}^d$. One of the main approaches is to learn a trainable drift term that drives an SDE to approximate the time-reversal of a pre-selected noising process that converts any target distribution $\pi$ to a noise distribution. This includes methods like Path Integral Sampler (PIS, [ZC22]), Neural Schrödinger-Föllmer Flows (NSFS, [Var+23]), Denoising Diffusion Samplers (DDS, [VGD23]), time-reversed DIffusion Sampler (DIS, [BRU24]), iterated Denoising Energy Matching (iDEM, [AS+24]), Diffusion-PINN Sampler [Shi+24b], Adjoint Sampling [Hav+25; Liu+25b; Cho+25], etc. Another popular approach trains the process to transport from a prior distribution $\pi_0$ to $\pi$ along a sequence of marginal distributions whose unnormalized densities are all available, such as the geometric interpolation $\pi_t \propto \pi_0^{1-t} \pi_1^t$. Methods of this type include the paper [MF23], Controlled Monte Carlo Diffusion (CMCD, [Var+24]), Liouville Flow Importance Sampler (LFIS [TPL24]), Non-Equilibrium Transport Sampler (NETS, [AVE25]), Sequential Controlled Langevin Diffusions (SCLD, [Che+25c]), the paper [Che+25a], underdamped diffusion bridges [Ble+25], Accelerated Parallel Tempering (APT, [Zha+25a]), Progessive Tempering with Diffusion (PTSD, [Ris+25]), etc. There are also works motivated by annealed importance sampling [Nea01] and Jarzynski equality [Jar97; VJ08], including Monte Carlo Diffusion (MCD, [Dou+22]) and Langevin Diffusion Variational Inference (LDVI, [GD23]). Despite the progress, these methods are restricted to the setting of $\mathbb{R}^d$ and cannot be easily extended to sampling discrete distributions. This distinguishes our proposed MDNS from all the aforementioned approaches.

In terms of methodology, MDNS is most relevant to PIS [ZC22]. Despite the difference in the underlying state space of the target distributions, PIS resembles our work in terms of method at a high level, since it approaches the sampling problem in $\mathbb{R}^d$ also from an SOC perspective. However, PIS training does not suffer from the non-differentiability of underlying sampling dynamics, which is a serious problem in discrete state spaces that we need to address by proposing new learning objectives that do not require trajectory differentiability.

In terms of task, MDNS is most connected to LEAPS [HAJ25], which focuses on the same problem of training discrete neural samplers using CTMC. We also note the concurrent work [OZL25], which appeared after our submission and proposed a framework, DNFS, similar to LEAPS. Unlike our framework, LEAPS takes a measure transport perspective and trains the neural sampler to follow a sequence of pre-defined discrete distributions. However, LEAPS requires a special inductive bias called locally equivariant networks to efficiently evaluate the training objectives, which greatly limits the algorithm's design flexibility. In contrast, our proposed MDNS does not impose constraints on the choice of score model backbone. Empirically, we also find that escorted-transport-based approaches like LEAPS tend to be ineffective when learning multimodal, high-dimensional distributions, while MDNS remains competitive.

**Discrete diffusion models.**    Diffusion models [SD+15; HJA20; SME21; Son+21] have been achieving state-of-the-art performances on generative modeling of various data modalities [Wat+23; Zhu+25c; Ess+24; Jin+25; Zhu+25a; Chi+23; HRT24; Roj+25b; Che+25b; Ren+25c]. As a natural generalization of diffusion models to discrete state space, discrete diffusion models [Aus+21; Cam+22;

CZH23; Sun+23b; LME24; Gat+24; Sha+25] have gradually attracted the attention of the research community as a competent method for generative modeling of general discrete sequence data. Discrete diffusion models have seen successful applications in numerous tasks, including image synthesis [Cha+22; Bai+25; Shi+25], text generation [Nie+25; Gon+25; Zhe+24; Arr+25], protein [Gru+23; Ala+23; Hay+25; Wan+24; Che+25d; TZC25], graph [Xu+24; Li+25], combinatorial optimization [SHL24; San+25] etc. Motivated by the huge empirical success of discrete diffusion models in practice, researchers have also worked on the theory of discrete diffusion to understand the key behind its success [Ben+24; CY25; Ren+25a; Fen+25; Kim+25; RRY25; HRT25]. An extensive portion of the literature is also devoted to the development of techniques to enhance further the effectiveness of discrete diffusion models, such as guidance [Sch+25; Guo+24; Nis+25; Roj+25a], distillation [Zhu+25b; DG25], training [Zha+25b; Liu+25a], planning [Par+25; Liu+25c], and inference-time scaling [Wan+25b; Tan+25].

Masked discrete diffusion model [Sah+24; Ou+25; Shi+24a; Zhe+25], as a particularly effective variant of discrete diffusion, is most related to our work. Masked diffusion models enjoy many favorable theoretical properties, such as a special structure of score functions that enables a simplified, variance-reduced training objective. Masked diffusion also has a natural connection to any-order autoregressive models and can be *precisely* simulated with a fixed number of inference steps. The training of MDNS also benefits from these additional structures, making it more effective than other existing approaches. However, what we propose is indeed a general framework and can be applied to other discrete diffusion models as well, such as uniform discrete diffusion. We defer the discussion of such extension to App. F.

**Optimal control of stochastic dynamics.**   The literature of solving SOC problems has a rich history, dating back to the work of Richard Bellman [Bel66], which introduces the Hamilton-Jacobi-Bellman (HJB) equation to characterize the optimal control through PDE. To efficiently solve SOC problems in high dimensions, neural networks are first introduced into the solving process by [EHJ17; HJE18] to alleviate the curse of dimensionality suffered by traditional approaches. Later, various learning objectives have been proposed to train the neural network for solving the SOC problems [DE+25; HLVE24]. We refer interested readers to [NR21] and [DE24] for a detailed review of the properties of different objectives.

Although all the works above focus on solving SOC problems in $\mathbb{R}^d$ (i.e., controlling SDEs instead of CTMCs), MDNS is still tightly connected to this literature due to our SOC formulation of the discrete sampling problem. Moreover, in our main training framework, we took a path measure perspective for designing the learning objectives, which generalizes the perspective used in [NR21] to a discrete state space. However, there is a notable difference between the SOC for CTMC (considered in this work) and the SOC for SDEs (widely studied in the literature) due to the non-differentiable nature of CTMCs, which further limits the feasible choice of learning objectives. These differences separate our paper from the works mentioned above.

# B   Details of Algorithms

In Alg. 1, `Resample_with_Mask` means sample random variables $\{\lambda^{(i,r)}\}_{1 \le i \le B, 1 \le r \le R} \overset{\text{i.i.d.}}{\sim}$ $\text{Unif}(0,1)$, and for each $i$ and $r$, obtain $X^{(i,r)}$ by randomly masking each entry of $X^{(i)}$ with probability $\lambda^{(i,r)}$. `Sample_Trajectories` is defined as in Alg. 2, which is also the inference algorithm for MDNS.

# C   Theory of Continuous-time Markov Chain and Stochastic Optimal Control

## C.1   Continuous-time Markov Chain

We refer readers to [Cam+22; LME24; Sun+23b; LME24; CY25; Ren+25a; Ren+25b] for the general theory of CTMC. Below, we present several key lemmas that will be used in our paper.

**Lemma 4.** *Let $X$ be a CTMC with generator $Q$, and let $p_t$ denote the probability distribution of $X_t$, i.e., $p_t(\cdot) = \Pr(X_t = \cdot)$. Then $p$ satisfies the following **Kolmogorov forward equation**:*

$$\partial_t p_t(x) = \sum_y Q_t(y, x) p_t(y) = \sum_{y \ne x} (Q_t(y, x) p_t(y) - Q_t(x, y) p_t(x)), \ \forall x. \tag{19}$$

---

**Algorithm 2** `Sample_Trajectories`: Sample trajectories and compute weights

---

**Require:** score model $s_\theta$, reward function $r : \mathcal{X}_0 \to \mathbb{R}$, batch size $B$.
1: Initialize fully masked sequences $\{X^{(i)} = (\mathbf{M}, ..., \mathbf{M})\}_{1 \le i \le B}$ and weights $\{W^{(i)} = 0\}_{1 \le i \le B}$.
2: Sample $B$ i.i.d. permutations of $\{1, ..., D\}$: $\{\Pi^{(i)} = (\Pi_1^{(i)}, ..., \Pi_D^{(i)})\}_{1 \le i \le B}$.
3: **for** $d = 1$ **to** $D$ **do**
4:     Call the score model and get all the scores $\{s_\theta(X^{(i)})\}_{1 \le i \le B}$.
5:     For each $1 \le i \le B$, sample a random integer $n^{(i)}$ in $\{1, ..., N\}$ following the probability
    distribution $\Pr(n^{(i)} = n) = s_\theta(X^{(i)})_{\Pi_d^{(i)}, n}$, and update the $\Pi_d^{(i)}$-th entry of $X^{(i)}$ as $n^{(i)}$.
6:     For each $1 \le i \le B$, update weights $W^{(i)} \leftarrow W^{(i)} + \log\left(\frac{1}{N} \Big/ s_\theta(X^{(i)})_{\Pi_d^{(i)}, n^{(i)}}\right)$.
7: For each $1 \le i \le B$, update weights with the final reward: $W^{(i)} \leftarrow W^{(i)} + r(X^{(i)})$.
    **return** pairs of sample and weights $\{X^{(i)}, W^u(X^{(i)}) := W^{(i)}\}_{1 \le i \le B}$.

---

*Moreover, the solution $p$ to (19) given boundary condition at either $0$ or $T$ is unique when $[0, T] \ni t \mapsto Q_t \in \mathbb{R}^{\mathcal{X} \times \mathcal{X}}$ is a continuous function.*

*Proof.* By (1), we have

$$p_{t+\Delta t}(x) = \sum_y \Pr(X_{t+\Delta t} = x | X_t = y) p_t(y) = \sum_y \left( 1_{y=x} + \Delta t Q_t(y, x) + O(\Delta t^2) \right) p_t(y)$$

$$= p_t(x) + \Delta t \sum_y Q_t(y, x) p_t(y) + O(\Delta t^2).$$

Therefore, by taking the limit $\Delta t \to 0$, we have

$$\partial_t p_t(x) = \sum_y Q_t(y, x) p_t(y) = \sum_{y \neq x} Q_t(y, x) p_t(y) + Q_t(x, x) p_t(x)$$

$$= \sum_{y \neq x} Q_t(y, x) p_t(y) - \sum_{y \neq x} Q_t(x, y) p_t(x).$$

The uniqueness of the solution follows from the uniqueness of the solution to the linear ODE, by equivalently writing (19) as $\partial_t \boldsymbol{p}_t = Q_t^{\mathrm{T}} \boldsymbol{p}_t$, $t \in [0, T]$, where $\boldsymbol{p}_t = (p_t(x) : x \in \mathcal{X})$ are column vectors in $\mathbb{R}^{|\mathcal{X}|}$. $\qquad\square$

**Lemma 5.** *Let $X$ be a CTMC with generator $Q$. For any bounded test function $\phi : \mathcal{X} \to \mathbb{R}$, define $\phi_t(\cdot) := \mathbb{E}[\phi(X_T)|X_t = \cdot]$. Then $\phi_t$ satisfies the following **Kolmogorov backward equation**:*

$$-\partial_t \phi_t(x) = \sum_y \phi_t(y) Q_t(x, y) = \sum_{y \neq x} (\phi_t(y) - \phi_t(x)) Q_t(x, y), \ \phi_T(x) = \phi(x), \ \forall x \in \mathcal{X}. \quad (20)$$

*Moreover, the solution $\phi$ to (20) is unique when $[0, T] \ni t \mapsto Q_t \in \mathbb{R}^{\mathcal{X} \times \mathcal{X}}$ is a continuous function.*

*Proof.* By (1), we have

$$\phi_t(x) = \mathbb{E}[\mathbb{E}[\phi(X_T)|X_{t+\Delta t}]|X_t = x] = \mathbb{E}[\phi_{t+\Delta t}(X_{t+\Delta t})|X_t = x]$$

$$= \sum_y \phi_{t+\Delta t}(y)(1_{x=y} + \Delta t Q_t(x, y) + O(\Delta t^2))$$

$$= \phi_{t+\Delta t}(x) + \Delta t \sum_y \phi_{t+\Delta t}(y) Q_t(x, y) + O(\Delta t^2).$$

Hence, by taking the limit $\Delta t \to 0$, we have

$$-\partial_t \phi_t(x) = \sum_y \phi_t(y) Q_t(x, y) = \sum_{y \neq x} \phi_t(y) Q_t(x, y) + \phi_t(x) Q_t(x, x) = \sum_{y \neq x} (\phi_t(y) - \phi_t(x)) Q_t(x, y).$$

The uniqueness of the solution also follows from the uniqueness of the solution to the linear ODE, by equivalently writing (20) as $-\partial_t \boldsymbol{\phi}_t = Q_t \boldsymbol{\phi}_t$, $t \in [0, T]$; $\boldsymbol{\phi}_T = \boldsymbol{\phi}$, where $\boldsymbol{\phi}_t = (\phi_t(x) : x \in \mathcal{X})$ and $\boldsymbol{\phi} = (\phi(x) : x \in \mathcal{X})$ are column vectors in $\mathbb{R}^{|\mathcal{X}|}$. $\qquad\square$

**Proof of Lem. 1.**

*Proof.* Let $\Delta t = \frac{T}{K}$ and $t_k = k\Delta t$. We have

$$
\begin{aligned}
\log \frac{\mathrm{d}\mathbb{P}^2}{\mathrm{d}\mathbb{P}^1}(\xi) &= \log \frac{\mathrm{d}\mu_2}{\mathrm{d}\mu_1}(\xi_0) + \sum_{k=0}^{K-1} \log \frac{\mathbb{P}^2_{t_{k+1}|t_k}(\xi_{t_{k+1}}|\xi_{t_k})}{\mathbb{P}^1_{t_{k+1}|t_k}(\xi_{t_{k+1}}|\xi_{t_k})} + O(\Delta t) \\
&= \log \frac{\mathrm{d}\mu_2}{\mathrm{d}\mu_1}(\xi_0) + \sum_{k=0}^{K-1} \left( \log \frac{1_{\xi_{t_{k+1}}=\xi_{t_k}} + \Delta t Q^2_{t_k}(\xi_{t_{k+1}}|\xi_{t_k})}{1_{\xi_{t_{k+1}}=\xi_{t_k}} + \Delta t Q^1_{t_k}(\xi_{t_{k+1}}|\xi_{t_k})} + O(\Delta t^2) \right) + O(\Delta t)
\end{aligned}
$$

By the definition of the generator. If $\xi_{t_{k+1}} \neq \xi_{t_k}$, then

$$
\log \frac{1_{\xi_{t_{k+1}}=\xi_{t_k}} + \Delta t Q^2_{t_k}(\xi_{t_{k+1}}|\xi_{t_k})}{1_{\xi_{t_{k+1}}=\xi_{t_k}} + \Delta t Q^1_{t_k}(\xi_{t_{k+1}}|\xi_{t_k})} = \log \frac{Q^2_{t_k}(\xi_{t_k}, \xi_{t_{k+1}})}{Q^1_{t_k}(\xi_{t_k}, \xi_{t_{k+1}})}.
$$

If otherwise, by Taylor expansion,

$$
\begin{aligned}
\log \frac{1_{\xi_{t_{k+1}}=\xi_{t_k}} + \Delta t Q^2_{t_k}(\xi_{t_{k+1}}|\xi_{t_k})}{1_{\xi_{t_{k+1}}=\xi_{t_k}} + \Delta t Q^1_{t_k}(\xi_{t_{k+1}}|\xi_{t_k})} &= \Delta t(Q^2_{t_k}(\xi_{t_k}, \xi_{t_k}) - Q^1_{t_k}(\xi_{t_k}, \xi_{t_k})) + O(\Delta t^2) \\
&= \Delta t \sum_{y \neq \xi_k} (Q^1_{t_k} - Q^2_{t_k})(\xi_{t_k}, y) + O(\Delta t^2).
\end{aligned}
$$

Hence, taking the limit $K \to \infty$, we have

$$
\begin{aligned}
\log \frac{\mathrm{d}\mathbb{P}^2}{\mathrm{d}\mathbb{P}^1}(\xi) &= \log \frac{\mathrm{d}\mu_2}{\mathrm{d}\mu_1}(\xi_0) + \sum_{k:\xi_{t_{k+1}} \neq \xi_{t_k}} \log \frac{Q^2_{t_k}(\xi_{t_k}, \xi_{t_{k+1}})}{Q^1_{t_k}(\xi_{t_k}, \xi_{t_{k+1}})} \\
&\quad + \Delta t \sum_{k:\xi_{t_{k+1}}=\xi_{t_k}} \sum_{y \neq \xi_k} (Q^1_{t_k} - Q^2_{t_k})(\xi_{t_k}, y) + O(\Delta t) \\
&\to \log \frac{\mathrm{d}\mu_2}{\mathrm{d}\mu_1}(\xi_0) + \sum_{t:\xi_{t-} \neq \xi_t} \log \frac{Q^2_t(\xi_{t-}, \xi_t)}{Q^1_t(\xi_{t-}, \xi_t)} + \int_0^T \sum_{y \neq \xi_t} (Q^1_t(\xi_t, y) - Q^2_t(\xi_t, y)) \mathrm{d}t.
\end{aligned}
$$

$\square$

**Corollary 1.** *Given two CTMCs with generators $Q^1, Q^2$ and initial distributions $\mu_1, \mu_2$ on $\mathcal{X}$, let $\mathbb{P}^1, \mathbb{P}^2$ be the associated path measures. Then the KL divergence between these two path measures can be written as*

$$
\mathrm{KL}(\mathbb{P}^2\|\mathbb{P}^1) = \mathrm{KL}(\mu_2\|\mu_1) + \mathbb{E}_{\xi \sim \mathbb{P}^2} \int_0^T \sum_{y \neq \xi_t} \left( Q^2_t \log \frac{Q^2_t}{Q^1_t} + Q^1_t - Q^2_t \right) (\xi_t, y) \mathrm{d}t.
$$

*Proof.* It suffices to take expectation w.r.t. $\mathbb{P}^2$ on both sides of (2). For the first term on the r.h.s., $\mathbb{E}_{\xi \sim \mathbb{P}^2} \log \frac{\mathrm{d}\mu_2}{\mathrm{d}\mu_1}(\xi_0) = \mathbb{E}_{\mu_2(\xi_0)} \log \frac{\mathrm{d}\mu_2}{\mathrm{d}\mu_1}(\xi_0) = \mathrm{KL}(\mu_2\|\mu_1)$; for the third term, the expectation is straightforward. To compute the expectation of the second term under $\mathbb{P}^2$, we start with the

discrete-time version and take limit when $K \to \infty$:

$$\mathbb{E}_{\xi \sim \mathbb{P}^2}\left[\sum_{k:\xi_{t_{k+1}} \neq \xi_{t_k}} \log \frac{Q_{t_k}^2(\xi_{t_k}, \xi_{t_{k+1}})}{Q_{t_k}^1(\xi_{t_k}, \xi_{t_{k+1}})}\right] = \mathbb{E}_{\xi \sim \mathbb{P}^2}\left[\sum_{k=0}^{K-1} \log \frac{Q_{t_k}^2(\xi_{t_k}, \xi_{t_{k+1}})}{Q_{t_k}^1(\xi_{t_k}, \xi_{t_{k+1}})} 1_{\xi_{t_{k+1}} \neq \xi_{t_k}}\right]$$

$$= \sum_{k=0}^{K-1} \mathbb{E}_{\mathbb{P}_{t_k}^2(\xi_{t_k})\mathbb{P}_{t_{k+1}|t_k}^2(\xi_{t_{k+1}}|\xi_{t_k})}\left[\log \frac{Q_{t_k}^2(\xi_{t_k}, \xi_{t_{k+1}})}{Q_{t_k}^1(\xi_{t_k}, \xi_{t_{k+1}})} 1_{\xi_{t_{k+1}} \neq \xi_{t_k}}\right]$$

$$= \sum_{k=0}^{K-1} \mathbb{E}_{\mathbb{P}_{t_k}^2(\xi_{t_k})} \sum_{y \neq \xi_{t_k}} \mathbb{P}_{t_{k+1}|t_k}^2(y|\xi_{t_k}) \log \frac{Q_{t_k}^2(\xi_{t_k}, y)}{Q_{t_k}^1(\xi_{t_k}, y)}$$

$$= \sum_{k=0}^{K-1} \mathbb{E}_{\mathbb{P}_{t_k}^2(\xi_{t_k})} \sum_{y \neq \xi_{t_k}} \left(\Delta t Q_{t_k}^2(\xi_{t_k}, y) \left[\log \frac{Q_{t_k}^2(\xi_{t_k}, y)}{Q_{t_k}^1(\xi_{t_k}, y)}\right] + O(\Delta t^2)\right)$$

$$\to \int_0^T \mathbb{E}_{\mathbb{P}_t^2(\xi_t)} \sum_{y \neq \xi_t} Q_t^2 \log \frac{Q_t^2}{Q_t^1}(\xi_t, y) \mathrm{d}t = \mathbb{E}_{\xi \sim \mathbb{P}^2} \int_0^T \sum_{y \neq \xi_t} Q_t^2 \log \frac{Q_t^2}{Q_t^1}(\xi_t, y) \mathrm{d}t$$

Thus, the proof is complete. $\qquad \square$

## C.2 Stochastic Optimal Control

Here, we give proof of some key lemmas in Sec. 3.1 concerning the SOC for CTMC. See also [Wan+25a] for a review of results.

The proof of (8) can be found in proving the following lemma.

**Lemma 6.** *The value function satisfies the following **Hamiltonian-Jacobi-Bellman (HJB) equation**:*

$$\partial_t V_t(x) = \sum_{y \neq x} Q_t^0(x, y)(1 - e^{V_t(y) - V_t(x)}) \iff \partial_t e^{V_t(x)} = \sum_{y \neq x} Q_t^0(x, y)(e^{V_t(x)} - e^{V_t(y)}). \quad (21)$$

*Proof.* By the dynamic programming principle, we have

$$-V_t(x) = \inf_u \mathbb{E}_{X \sim \mathbb{P}^u}\left[\left(\int_t^{t+\Delta t} + \int_{t+\Delta t}^T\right) \sum_{y \neq X_s} \star_s(X_s, y) \mathrm{d}s - r(X_T)\bigg| X_t = x\right]$$

$$= \Delta t \inf_u \sum_{y \neq x} \star_t(x, y) + O(\Delta t^2) + \inf_u \mathbb{E}_{X \sim \mathbb{P}^u}[-V_{t+\Delta t}(X_{t+\Delta t})|X_t = x],$$

where $\star. = Q_{\cdot}^u \log \frac{Q_{\cdot}^u}{Q_{\cdot}^0} - Q_{\cdot}^u + Q_{\cdot}^0$. We decompose the last term as

$$\inf_u \mathbb{E}_{X \sim \mathbb{P}^u}[-V_{t+\Delta t}(X_{t+\Delta t})|X_t = x]$$

$$= \inf_u\left[-\sum_y V_{t+\Delta t}(y)\left(1_{x=y} + \Delta t Q_t^u(x, y) + O(\Delta t^2)\right)\right]$$

$$= \inf_u\left[-V_{t+\Delta t}(x) - \Delta t \sum_{y \neq x} V_{t+\Delta t}(y)Q_t^u(x, y) + \Delta t \sum_{y \neq x} V_{t+\Delta t}(x)Q_t^u(x, y) + O(\Delta t^2)\right]$$

$$= -V_{t+\Delta t}(x) + \Delta t \inf_u\left[\sum_{y \neq x} Q_t^u(x, y)(V_{t+\Delta t}(x) - V_{t+\Delta t}(y))\right] + O(\Delta t^2).$$

Hence, by taking the limit $\Delta t \to 0$, we have

$$\partial_t V_t(x) = \inf_u\left[\sum_{y \neq x}\left(Q_t^u \log \frac{Q_t^u}{Q_t^0} - Q_t^u + Q_t^0\right)(x, y) + (V_t(x) - V_t(y))Q_t^u(x, y)\right]. \quad (22)$$

For each pair of $x \neq y$, by optimizing the r.h.s. of (22) with respect to $Q_t^u(x, y)$, we have

$$Q_t^*(x, y) = \underset{Q_t^u(x,y) \geq 0}{\operatorname{argmin}} \left[ \left( Q_t^u \log \frac{Q_t^u}{Q_t^0} - Q_t^u + Q_t^0 \right)(x, y) + (V_t(x) - V_t(y))Q_t^u(x, y) \right]$$
$$= Q_t^0(x, y) \mathrm{e}^{V_t(y) - V_t(x)},$$

and plugging this back into (22) gives the first equality in Lem. 6. The second equality is an immediate consequence of the first one. $\qquad \square$

**Lemma 7.** *Assume that $t \mapsto Q_t^0$ is continuous. Then the following **Feynman-Kac formula** holds:* $\mathrm{e}^{V_t(x)} = \mathbb{E}_{\mathbb{P}^0}[\mathrm{e}^{r(X_T)} | X_t = x]$.

*Proof.* This result follows immediately from the second form of the HJB equation (21) and the Kolmogorov backward equation Lem. 5. $\qquad \square$

**Lemma 8.** *Assume that $t \mapsto Q_t^0$ is continuous. Then the optimal marginal distribution $\mathbb{P}_t^*$ satisfies* $\mathbb{P}_t^*(x) = \frac{1}{Z} \mathbb{P}_t^0(x) \mathrm{e}^{V_t(x)}$, *where $Z = \mathbb{E}_{\mathbb{P}_T^0} \mathrm{e}^r$.*

*Proof.* Let $h_t(x) := \frac{1}{Z} \mathbb{P}_t^0(x) \mathrm{e}^{V_t(x)}$, and obviously $h_T = \mathbb{P}_T^*$. Recall from Lem. 4 that

$$\partial_t \mathbb{P}_t^0(x) = \sum_{y \neq x} (Q_t^0(y, x) \mathbb{P}_t^0(y) - Q_t^0(x, y) \mathbb{P}_t^0(x)).$$

Also, from Lem. 6,

$$\partial_t \mathrm{e}^{V_t(x)} = \sum_{y \neq x} Q_t^0(x, y)(\mathrm{e}^{V_t(x)} - \mathrm{e}^{V_t(y)}).$$

Multiplying the first equation by $\mathrm{e}^{V_t(x)}$ and the second by $\mathbb{P}_t^0(x)$, we can easily derive

$$\partial_t h_t(x) = \sum_{y \neq x} (Q_t^0(y, x) h_t(y) - Q_t^0(x, y) h_t(x)),$$

which is the Kolmogorov forward equation for $\mathbb{P}^*$ driven by generator $Q^*$. By the uniqueness result, we conclude that $h_t(x) = \mathbb{P}_t^*(x)$. $\qquad \square$

**Remark 1.** *Note that $\mathbb{P}_0^*(x) = \frac{1}{Z} \mathbb{P}_0^0(x) \mathrm{e}^{V_0(x)} = \frac{1}{Z} p_{\mathrm{init}}(x) \mathrm{e}^{V_0(x)}$. Thus, $\mathbb{P}_0^* = p_{\mathrm{init}}$ if.f. $V_0(\cdot)$ is a constant, which can be guaranteed when $X_0$ and $X_T$ are independent under $\mathbb{P}^0$ due to the Feynman-Kac formula. Otherwise, $\mathbb{P}_0^* \neq p_{\mathrm{init}}$ leads to a contradiction, which means there is no solution to the SOC problem (6).*

**Lemma 9.** *The following relation between the reference and optimal path measures holds:* $\frac{\mathrm{d}\mathbb{P}^*}{\mathrm{d}\mathbb{P}^0}(\xi) = \frac{1}{Z} \mathrm{e}^{r(\xi_T)}$ *for any $\xi$, where $Z = \mathbb{E}_{\mathbb{P}_T^0} \mathrm{e}^r$.*

*Proof.* By (2), (8), and Lem. 8,

$$\log \frac{\mathrm{d}\mathbb{P}^*}{\mathrm{d}\mathbb{P}^0}(\xi) = \log \frac{\mathrm{d}\mathbb{P}_0^*}{\mathrm{d}\mathbb{P}_0^0}(\xi_0) + \sum_{t:\xi_{t-} \neq \xi_t} \log \frac{Q_t^*(\xi_{t-}, \xi_t)}{Q_t^0(\xi_{t-}, \xi_t)} + \int_0^T \sum_{y \neq \xi_t} (Q_t^0(\xi_t, y) - Q_t^*(\xi_t, y)) \mathrm{d}t$$

$$= V_0(\xi_0) - \log Z + \sum_{t:\xi_{t-} \neq \xi_t} (V_t(\xi_t) - V_t(\xi_{t-})) + \int_0^T \sum_{y \neq \xi_t} Q_t^0(\xi_t, y)(1 - \mathrm{e}^{V_t(y) - V_t(\xi_t)}) \mathrm{d}t.$$

Since $\xi$ is piecewise constant and càdlàg, $t \mapsto V_t(x)$ is continuous for all $x$, suppose the jump times are $0 < t_1 < \cdots < t_n < T$, and let $t_0 = 0$, $t_{n+1} = T$, we can write

$$V_T(\xi_T) - V_0(\xi_0) = \sum_{i=0}^n (V_{t_{i+1}}(\xi_{t_i}) - V_{t_i}(\xi_{t_i})) + \sum_{i=1}^n (V_{t_i}(\xi_{t_i}) - V_{t_i}(\xi_{t_{i-1}}))$$

$$= \sum_{i=0}^n \int_{t_i}^{t_{i+1}} \partial_t V_t(\xi_{t_i}) \mathrm{d}t + \sum_{t:\xi_{t-} \neq \xi_t} (V_t(\xi_t) - V_t(\xi_{t-}))$$

$$= \int_0^T \partial_t V_t(\xi_t) \mathrm{d}t + \sum_{t:\xi_{t-} \neq \xi_t} (V_t(\xi_t) - V_t(\xi_{t-})).$$

On the other hand, by (8) and (21),

$$\int_0^T \sum_{y \neq \xi_t} (Q_t^0(\xi_t, y) - Q_t^*(\xi_t, y)) \mathrm{d}t = \int_0^T \sum_{y \neq \xi_t} Q_t^0(\xi_t, y)(1 - \mathrm{e}^{V_t(y) - V_t(\xi_t)}) \mathrm{d}t = \int_0^T \partial_t V_t(\xi_t) \mathrm{d}t.$$

Finally, as $V_T = r$, the proof is completed. $\qquad\square$

### C.3  Omitted Proofs in the Main Text

**Proof of (9).**

*Proof.* Throughout the proof, we assume regularity conditions that guarantee the validity of swapping the position between the integral and the derivative. We also assume $\mathbb{P}^u$ and $\mathbb{P}^*$ are mutually absolutely continuous. We have

$$\nabla_\theta \operatorname{KL}(\mathbb{P}^u \| \mathbb{P}^*) = \nabla_\theta \mathbb{E}_{\mathbb{P}^{\bar{u}}} \frac{\mathrm{d}\mathbb{P}^u}{\mathrm{d}\mathbb{P}^{\bar{u}}} \log \frac{\mathrm{d}\mathbb{P}^u}{\mathrm{d}\mathbb{P}^*} = \mathbb{E}_{\mathbb{P}^{\bar{u}}} \nabla_\theta \frac{\mathrm{d}\mathbb{P}^u}{\mathrm{d}\mathbb{P}^{\bar{u}}} \log \frac{\mathrm{d}\mathbb{P}^u}{\mathrm{d}\mathbb{P}^*}$$

$$= \mathbb{E}_{\mathbb{P}^{\bar{u}}} \left[ \nabla_\theta \frac{\mathrm{d}\mathbb{P}^u}{\mathrm{d}\mathbb{P}^{\bar{u}}} \cdot \log \frac{\mathrm{d}\mathbb{P}^u}{\mathrm{d}\mathbb{P}^*} + \frac{\mathrm{d}\mathbb{P}^u}{\mathrm{d}\mathbb{P}^{\bar{u}}} \nabla_\theta \log \frac{\mathrm{d}\mathbb{P}^u}{\mathrm{d}\mathbb{P}^*} \right] \quad (\textit{Chain rule for derivative})$$

$$= \mathbb{E}_{\mathbb{P}^{\bar{u}}} \left[ \frac{\mathrm{d}\mathbb{P}^u}{\mathrm{d}\mathbb{P}^{\bar{u}}} \nabla_\theta \log \frac{\mathrm{d}\mathbb{P}^u}{\mathrm{d}\mathbb{P}^{\bar{u}}} \cdot \log \frac{\mathrm{d}\mathbb{P}^u}{\mathrm{d}\mathbb{P}^*} + \frac{\mathrm{d}\mathbb{P}^u}{\mathrm{d}\mathbb{P}^{\bar{u}}} \nabla_\theta \log \frac{\mathrm{d}\mathbb{P}^u}{\mathrm{d}\mathbb{P}^*} \right] \quad (\textit{Gradient of} \log)$$

$$= \mathbb{E}_{\mathbb{P}^{\bar{u}}} \left[ \nabla_\theta \log \frac{\mathrm{d}\mathbb{P}^u}{\mathrm{d}\mathbb{P}^{\bar{u}}} \cdot \log \frac{\mathrm{d}\mathbb{P}^{\bar{u}}}{\mathrm{d}\mathbb{P}^*} + \nabla_\theta \log \frac{\mathrm{d}\mathbb{P}^u}{\mathrm{d}\mathbb{P}^*} \right] \quad (\textit{When not taking gradient,}\ \frac{\mathrm{d}\mathbb{P}^u}{\mathrm{d}\mathbb{P}^{\bar{u}}} \equiv 1,\ \frac{\mathrm{d}\mathbb{P}^u}{\mathrm{d}\mathbb{P}^*} = \frac{\mathrm{d}\mathbb{P}^{\bar{u}}}{\mathrm{d}\mathbb{P}^*})$$

$$= \mathbb{E}_{\mathbb{P}^{\bar{u}}} \left[ \nabla_\theta \log \frac{\mathrm{d}\mathbb{P}^u}{\mathrm{d}\mathbb{P}^*} \left( \log \frac{\mathrm{d}\mathbb{P}^{\bar{u}}}{\mathrm{d}\mathbb{P}^*} + 1 \right) \right] \quad (\nabla_\theta \log \frac{\mathrm{d}\mathbb{P}^u}{\mathrm{d}\mathbb{P}^*} = \nabla_\theta \log \frac{\mathrm{d}\mathbb{P}^u}{\mathrm{d}\mathbb{P}^{\bar{u}}}). \tag{23}$$

The second term is actually zero:

$$\mathbb{E}_{\mathbb{P}^{\bar{u}}} \nabla_\theta \log \frac{\mathrm{d}\mathbb{P}^u}{\mathrm{d}\mathbb{P}^*} = \mathbb{E}_{\mathbb{P}^{\bar{u}}} \frac{\nabla_\theta (\mathrm{d}\mathbb{P}^u / \mathrm{d}\mathbb{P}^*)}{\mathrm{d}\mathbb{P}^u / \mathrm{d}\mathbb{P}^*} = \mathbb{E}_{\mathbb{P}^{\bar{u}}} \frac{\nabla_\theta (\mathrm{d}\mathbb{P}^u / \mathrm{d}\mathbb{P}^*)}{\mathrm{d}\mathbb{P}^{\bar{u}} / \mathrm{d}\mathbb{P}^*} \quad (\textit{When not taking gradient,}\ \frac{\mathrm{d}\mathbb{P}^u}{\mathrm{d}\mathbb{P}^*} = \frac{\mathrm{d}\mathbb{P}^{\bar{u}}}{\mathrm{d}\mathbb{P}^*})$$

$$= \mathbb{E}_{\mathbb{P}^*} \nabla_\theta \frac{\mathrm{d}\mathbb{P}^u}{\mathrm{d}\mathbb{P}^*} = \nabla_\theta \mathbb{E}_{\mathbb{P}^*} \frac{\mathrm{d}\mathbb{P}^u}{\mathrm{d}\mathbb{P}^*} = \nabla_\theta 1 = 0.$$

Therefore, the 1 in (23) can be replaced by any real number $C$. Swapping the positions of $\mathbb{E}_{\mathbb{P}^{\bar{u}}}$ and $\nabla_\theta$ completes the proof. $\qquad\square$

**Proof of Lem. 2.**

*Proof.* Note that under the generator $Q_t$ defined as (12), it is obvious that each token evolves independently. Therefore, it suffices to consider the transition probability of a single token, $Q^{\mathrm{tok}} = (Q_t^{\mathrm{tok}} \in \mathbb{R}^{(N+1) \times (N+1)})_{t \in [0, T]}$.

Let the mask token $\mathbf{M}$ be $N + 1$. By definition, $Q_t^{\mathrm{tok}}$ can be expressed as $Q_t^{\mathrm{tok}} = \gamma(t) u v^{\mathrm{T}}$, where $u = (0, ..., 0, 1)^{\mathrm{T}}$ and $v = \left( \frac{1}{N}, ..., \frac{1}{N}, -1 \right)^{\mathrm{T}}$ are both $(N+1)$-dimensional vectors. Thus, by the Kolmogorov forward equation (Lem. 4), the transition probability matrix from $s$ to $t$ is given by $\exp \left( \left( \int_s^t \gamma(r) \mathrm{d}r \right) u v^{\mathrm{T}} \right)$. To compute the matrix exponential, note that $(u v^{\mathrm{T}})^k = u (v^{\mathrm{T}} u)^{k-1} v^{\mathrm{T}} = (-1)^{k-1} u v^{\mathrm{T}}$, and hence

$$\mathrm{e}^{\lambda (u v^{\mathrm{T}})} = I + \sum_{k=1}^\infty \frac{\lambda^k}{k!} (u v^{\mathrm{T}})^k = I + \sum_{k=1}^\infty \frac{\lambda^k}{k!} (-1)^{k-1} u v^{\mathrm{T}}$$

$$= I - \sum_{k=1}^\infty \frac{(-\lambda)^k}{k!} u v^{\mathrm{T}} = I + (1 - \mathrm{e}^{-\lambda}) u v^{\mathrm{T}}.$$

Therefore, we have $\mathbb{P}_{T|0}^0(n|\mathbf{M}) = \frac{1}{N} \left( 1 - \exp \left( \int_0^T \gamma(r) \mathrm{d}r \right) \right)$ for $n \in \{1, ..., N\}$. Since the initial sample is completely masked, $\int_0^T \gamma(r) \mathrm{d}r = \infty$ guarantees $\mathbb{P}_T^0 = p_{\mathrm{unif}}$. Moreover, as the initial distribution is a point mass, it is obvious that $X_0$ and $X_T$ are independent, i.e., $\mathbb{P}^0$ is memoryless. $\qquad\square$

**Proof of Lem. 3.**

*Proof.* From (8) and (12), we only need to consider the values of $Q_t^*(x, x^{d \leftarrow n})$ for $x^d = \mathbf{M}$, as otherwise the value is $0$.

We first compute the conditional distribution of $X_T$ given $X_t = x \in \mathcal{X}$ under $\mathbb{P}^0$. We know from the proof of Lem. 2 that each token unmasks into $\{1, ..., N\}$ uniformly and independently, and remains unchanged as long as it is unmasked. Thus, the conditional distribution of $X_T$ given $X_t = x$ is the uniform distribution over the subset of $\mathcal{X}_0$ whose tokens at the unmasked positions of $x$ are the same as $x$, i.e.,

$$\mathbb{P}^0_{T|t}(\widetilde{x}|x) = \frac{1}{N^{|\{d : x^d = \mathbf{M}\}|}} \prod_{d : x^d \neq \mathbf{M}} 1_{\widetilde{x}^d = x^d}.$$

Hence, by Lem. 7, we have

$$\mathrm{e}^{V_t(x)} = \frac{1}{N^{|\{d : x^d = \mathbf{M}\}|}} \sum_{\widetilde{x} \in \mathcal{X}_0 : \, \widetilde{x}^d = x^d, \, \forall d : x^d \neq \mathbf{M}} \mathrm{e}^{r(\widetilde{x})}.$$

Notably, $V_t(x)$ is independent of $t$. Now, suppose $x^i = \mathbf{M}$, we have

$$
\begin{aligned}
\frac{\mathrm{e}^{V_t(x^{i \leftarrow n})}}{\mathrm{e}^{V_t(x)}} &= \frac{\frac{1}{N^{|\{d : (x^{i \leftarrow n})^d = \mathbf{M}\}|}} \sum_{\widetilde{x} \in \mathcal{X}_0 : \, \widetilde{x}^d = (x^{i \leftarrow n})^d, \, \forall d : (x^{i \leftarrow n})^d \neq \mathbf{M}} \mathrm{e}^{r(\widetilde{x})}}{\frac{1}{N^{|\{d : x^d = \mathbf{M}\}|}} \sum_{\widetilde{x} \in \mathcal{X}_0 : \, \widetilde{x}^d = x^d, \, \forall d : x^d \neq \mathbf{M}} \mathrm{e}^{r(\widetilde{x})}} \\
&= N \frac{\sum_{\widetilde{x} \in \mathcal{X}_0 : \, \widetilde{x}^d = x^d, \, \forall d : x^d \neq \mathbf{M}; \, \widetilde{x}^i = n} \pi(\widetilde{x})}{\sum_{\widetilde{x} \in \mathcal{X}_0 : \, \widetilde{x}^d = x^d, \, \forall d : x^d \neq \mathbf{M}} \pi(\widetilde{x})} \quad \left(\textit{Since } \mathrm{e}^{r(x)} = \frac{Z \pi(x)}{p_{\mathrm{unif}}(x)}\right) \\
&= N \Pr_{X \sim \pi}(X^i = n | X^{\mathrm{UM}} = x^{\mathrm{UM}}),
\end{aligned}
$$

which, together with (12), completes the proof. $\qquad\square$

**Proof of Prop. 1.**

*Proof.* The claim follows from the data processing inequality $\mathrm{KL}(p\|q) \geq \mathrm{KL}(T_\sharp p \| T_\sharp q)$, where $T$ is an arbitrary measurable mapping and $T_\sharp p$ is the law of $T(X)$ given $X \sim p$. $\qquad\square$

**Proof of Prop. 2**

*Proof.* The identity (14) directly implies that $Z = \mathbb{E}_{X \sim \mathbb{P}^u} \mathrm{e}^{W^u(X)}$, and hence $\widehat{Z}$ is an unbiased estimation of $Z$. By the Markov inequality,

$$
\begin{aligned}
\Pr\left(\left|\frac{\widehat{Z}}{Z} - 1\right| \geq \varepsilon\right) &= \mathbb{P}^u\left(\left|\frac{\mathrm{d}\mathbb{P}^*}{\mathrm{d}\mathbb{P}^u} - 1\right| \geq \varepsilon\right) \leq \frac{1}{\varepsilon} \mathbb{E}_{\mathbb{P}^u}\left|\frac{\mathrm{d}\mathbb{P}^*}{\mathrm{d}\mathbb{P}^u} - 1\right| \\
&= \frac{1}{\varepsilon} \int \left|\frac{\mathrm{d}\mathbb{P}^*}{\mathrm{d}\mathbb{P}^u} - 1\right| \frac{\mathrm{d}\mathbb{P}^u}{\mathrm{d}\mathbb{Q}} 1_{\frac{\mathrm{d}\mathbb{P}^u}{\mathrm{d}\mathbb{Q}} > 0} \mathrm{d}\mathbb{Q} \quad \left(\forall \mathbb{Q} \text{ that dominates both } \mathbb{P}^* \text{ and } \mathbb{P}^u, \text{ e.g., } \frac{1}{2}(\mathbb{P}^* + \mathbb{P}^u)\right) \\
&\leq \frac{1}{\varepsilon} \int \left|\frac{\mathrm{d}\mathbb{P}^u}{\mathrm{d}\mathbb{Q}} - \frac{\mathrm{d}\mathbb{P}^*}{\mathrm{d}\mathbb{Q}}\right| \mathrm{d}\mathbb{Q} \quad (\textit{Product rule of RN derivatives and } 1_A \leq 1, \, \forall A) \\
&= \frac{\mathrm{TV}(\mathbb{P}^u, \mathbb{P}^*)}{2\varepsilon} \quad (\textit{By definition of TV distance}).
\end{aligned}
$$

By the Pinsker's inequality $\mathrm{KL}(p\|q) \geq 2\,\mathrm{TV}(p,q)^2$, the probability can thus be bounded by $\frac{1}{4}$. The last claim follows from the **median trick**, see [GTC25a, Lem. 29]. $\qquad\square$

## D  Experimental Details and Additional Results: Learning Ising model

### D.1  General Training Hyperparameters and Model Architecture

**Model backbone.**  We use vision transformers (ViT, [Dos+21]) to serve as the backbone for the discrete diffusion model. In particular, we use the DeiT (Data-efficient image Transformers)

framework [Tou+21] with 2-dimensional rotary position embedding [Heo+25], which better captures the 2-dimensional spatial structure of the Ising model.

For $L = 4$, we use 2 blocks with a 32-dimensional hidden space and 4 attention heads, and the whole model contains 26k parameters. For $L = 16$, we use 6 blocks with a 64-dimensional embedding space and 4 attention heads, and the whole model contains 318k parameters.

**Training.** Among all the training tasks, we choose the batch size as 256, and use the AdamW optimizer [LH19] with a constant learning rate of 0.001. We always use exponential moving average (EMA) to stabilize the training, with a decay rate of 0.9999. All experiments of learning Ising model are trained on an NVIDIA RTX A6000. The training of $L = 4$ target distributions takes around 10 minutes while the training of $L = 16$ target distributions takes around 20 hours (or equivalently, 4 hours on an NVIDIA RTX A100). For $16 \times 16$ Ising model, we use $\mathcal{F}_{\mathrm{WDCE}}$ with a resampling frequency $k = 10$ and replicates $R = 8$ for a total of 50k iterations, among which 20k is warm-up training.

**Generating baseline and ground truth samples.** For the learning-based baseline, we train LEAPS on $16 \times 16$ Ising model for up to 150k steps for each temperature, which is comparable to, if not more than, our requirement on this task, to ensure a fair comparison. We also run the Metropolis-Hastings (MH) algorithm for a sufficiently long time to serve as a baseline. For $L = 4$ (resp., $L = 16$), we use a batch size 1024, and warm up the algorithm with $2^{10}$ (resp., $2^{20}$) burn-in iterations. After that, we collect the samples every $2^{10}$ (resp., $2^{16}$) steps to ensure sufficient mixing, and collect for $2^{10}$ rounds, so the final number of samples is $2^{20}$. For $L = 16$, we run the Swendsen-Wang (SW) algorithm on each temperature to generate examples accurately distributed as the ground truth. We use a batch size of 128, and warm up the algorithm with $2^{13}$ burn-in iterations. After that, we collect samples every 128 steps to ensure sufficient mixing of the chain, and collect for 32 rounds to gather a total of $2^{12}$ samples. For $L = 16$, the MH sampling takes around 3 hours while the SW sampling takes around 2 hours on a CPU.

## D.2   Evaluation and Additional Results on $4 \times 4$ Ising model

### D.2.1   Definition and Discussion on the Effective Sample Size

The effective sample size is a metric commonly used to evaluate the sampling quality and does *not* rely on the normalized target probability mass function or ground truth samples.

As in earlier works (e.g., [ZC22; AVE25; HAJ25]), given any control $u$, suppose we have $M$ trajectories $X^{(1)}, ..., X^{(M)} \overset{\text{i.i.d.}}{\sim} \mathbb{P}^u$, we can associate each $X^{(i)}$ with weight

$$\omega(X^{(i)}) = \frac{\mathrm{e}^{W^u(X^{(i)})}}{\sum_{j=1}^{M} \mathrm{e}^{W^u(X^{(j)})}},$$

so that the weighted empirical distribution $\sum_{i=1}^{M} \omega(X^{(i)}) \delta_{X^{(i)}}$ serves as a consistent approximation of $\mathbb{P}^*$ as $M \to \infty$. We define the (normalized) **effective sample size (ESS)** as

$$\mathrm{ESS} \coloneqq \left( M \sum_{i=1}^{M} \omega^2(X^{(i)}) \right)^{-1} \in \left[ \frac{1}{M}, 1 \right]. \tag{24}$$

A higher ESS typically means better sampling quality, but this is not always true as ESS is difficult to detect whether the samples miss a mode in the target distribution. For instance, suppose $\mathbb{P}^*$ is a uniform distribution on two points $\{a, b\}$ and $\mathbb{P}^u$ is a delta distribution on $a$, then sampling from $\mathbb{P}^u$ would always output $a$, and thus $\omega(X^{(i)} = a) \equiv \frac{1}{M}$, resulting in ESS = 1. This phenomenon is due to the fact $\mathbb{P}^*$ is not dominated by (i.e., absolutely continuous with respect to) $\mathbb{P}^u$. In fact, by the strong law of large numbers, one can show that

$$\mathrm{ESS} = \frac{\left( \frac{1}{M} \sum_{i=1}^{M} \frac{\mathrm{d}\mathbb{P}^*}{\mathrm{d}\mathbb{P}^u}(X^{(i)}) \right)^2}{\frac{1}{M} \sum_{i=1}^{M} \left( \frac{\mathrm{d}\mathbb{P}^*}{\mathrm{d}\mathbb{P}^u}(X^{(i)}) \right)^2} \xrightarrow{\text{a.s.}} \frac{\left( \mathbb{E}_{\mathbb{P}^u} \frac{\mathrm{d}\mathbb{P}^*}{\mathrm{d}\mathbb{P}^u} \right)^2}{\mathbb{E}_{\mathbb{P}^u} \left( \frac{\mathrm{d}\mathbb{P}^*}{\mathrm{d}\mathbb{P}^u} \right)^2}.$$

The r.h.s. is $\frac{1}{1+\chi^2(\mathbb{P}^* \| \mathbb{P}^u)}$ if $\mathbb{P}^*$ is dominated by $\mathbb{P}^u$, which aligns with the definition of ESS in [ZC22]. But if the learned path measure $\mathbb{P}^u$ only covers one of the high-probability regions in $\mathbb{P}^*$ and misses

the others, then $\mathbb{P}^*$ may not be dominated by $\mathbb{P}^u$ and ESS does not reveal the correct sampling quality. Therefore, only relying on ESS as the evaluation metric may be problematic.

### D.2.2 Evaluation of the $4 \times 4$ Learned Models.

For each step during training, after doing a gradient update to the model and updating the model parameters stored in EMA, we evaluate the current model by sampling using the parameters stored in EMA and computing the weights $W^u$ along the sampled trajectories. The batch size for sampling is 256. The ESS reported in Tab. 2 are the average ESS of the last 100 steps.

As $|\mathcal{X}_0| = 2^{L^2} = 65536$, the partition function $Z$ can be computed explicitly, and thus we can obtain the whole probability distribution $\pi$. We use the random order autoregressive sampler to sample from the learned model, and then compute the empirical distribution of the samples $\widehat{p}_{\mathrm{samp}}$. Recall that for two categorical distributions $p$ and $q$ on $\mathcal{X}_0$, the total variation (TV) distance is defined as $\mathrm{TV}(p,q) := \frac{1}{2} \sum_{x \in \mathcal{X}_0} |p(x) - q(x)|$, the KL divergence is defined as $\mathrm{KL}(p\|q) := \sum_{x \in \mathcal{X}_0: q(x)>0} p(x) \log \frac{p(x)}{q(x)}$, and the $\chi^2$ divergence is defined as $\chi^2(p\|q) := \sum_{x \in \mathcal{X}_0: q(x)>0} \frac{p(x)^2}{q(x)} - 1 = \sum_{x \in \mathcal{X}_0: q(x)>0} \frac{(p(x)-q(x))^2}{q(x)}$. Note that in order to make $\mathrm{KL}(p\|q)$ and $\chi^2(p\|q)$ divergences well-defined, we require $p$ to be dominated by $q$, i.e., $q(x) > 0$ for all $x \in \mathcal{X}_0$ such that $p(x) > 0$, or equivalently, $q(x) = 0 \implies p(x) = 0$. Therefore, we do not choose $\mathrm{KL}(\pi\|\widehat{p}_{\mathrm{samp}})$ and $\chi^2(\pi\|\widehat{p}_{\mathrm{samp}})$ as the evaluation metrics.

As the partition function $Z$ can be computed explicitly, we can approximate the relative-entropy between the paths, $\mathrm{KL}(\mathbb{P}^u\|\mathbb{P}^*) = \log Z - \mathbb{E}_{\mathbb{P}^u} W^u(X)$, by the empirical means of the weights $W^u(X)$ for $X \sim \mathbb{P}^u$. Finally, as is discussed in Prop. 2, an unbiased estimation of $Z$ can be obtained by $\widehat{Z} = \mathrm{e}^{W^u(X)}$, $X \sim \mathbb{P}^u$.

### D.2.3 Results of Learning the $4 \times 4$ Ising model at Other Temperatures

We also provide results for learning distributions at critical ($\beta_{\mathrm{critical}} = 0.4407$) and low ($\beta_{\mathrm{low}} = 0.6$) temperatures in Tabs. 4 and 5. The same model structure, training configurations, and evaluation methods as discussed above are used, except that for learning the distribution at $\beta_{\mathrm{critical}}$, we train the model from scratch with 2000 steps, and for learning the distribution at $\beta_{\mathrm{low}}$, we train the model for 1000 steps starting from the 1000-step checkpoint for learning the high-temperature distribution with the LV loss. All four learning objectives are capable of training the model to generate high-quality samples compared with the baseline (MH algorithm).

Table 4: Results for learning $4 \times 4$ Ising model with $J = 1$, $h = 0.1$ and $\beta_{\mathrm{critical}} = 0.4407$, best in **bold**.

| Method | ESS $\uparrow$ | TV($\widehat{p}_{\mathrm{samp}}, \pi$) $\downarrow$ | KL($\widehat{p}_{\mathrm{samp}}\|\pi$) $\downarrow$ | $\chi^2(\widehat{p}_{\mathrm{samp}}\|\pi)$ $\downarrow$ | $\widehat{\mathrm{KL}}(\mathbb{P}^u\|\mathbb{P}^*)$ $\downarrow$ | Abs. err. of $\log \widehat{Z}$ $\downarrow$ |
|---|---|---|---|---|---|---|
| RERF | 0.8480 | 0.0841 | 0.0521 | 0.1691 | 0.0565 | 0.00150 |
| LV | **0.9809** | **0.0301** | **0.0222** | **0.0830** | **0.0083** | 0.00106 |
| CE | 0.9545 | 0.0454 | 0.0327 | 0.1824 | 0.0259 | 0.00175 |
| WDCE | 0.9644 | 0.0789 | 0.0375 | 0.0839 | 0.0177 | **0.00010** |
| Baseline (MH) | / | 0.0223 | 0.0193 | 0.0615 | / | / |

### D.2.4 Ablation Study of the Warm-up Training Strategy

As an ablation study of the effectiveness of the warm-up strategy, in Tab. 5 we also present the results of the model trained from scratch for learning the distribution at $\beta_{\mathrm{low}}$, and we can see that warm-up generally improves the training quality. Note that at low temperature, high ESS does not necessarily indicate good sampling quality, as the target distribution is highly concentrated on two modes: in fact, the all-positive configuration has probability 0.7530, the all-negative one has probability 0.1104, and all the remaining $2^{16} - 2 = 65534$ configurations occupy the rest portion 0.1366. As a result, if the samples are all concentrated on the all-positive configuration and do not cover the other mode, then the ESS would be close to 1, yet the overall distribution is far from the target. Thus, we need to rely on a diversified set of evaluation metrics to comprehensively evaluate the learned model, such as the

Table 5: Results for learning $4 \times 4$ Ising model with $J = 1$, $h = 0.1$ and $\beta_{\text{low}} = 0.6$ (with ablation study of the effectiveness of warm-up in training), best in **bold**.

| Method | Use warm-up | ESS $\uparrow$ | TV$(\widehat{p}_{\text{samp}}, \pi) \downarrow$ | KL$(\widehat{p}_{\text{samp}}\|\pi) \downarrow$ | $\chi^2(\widehat{p}_{\text{samp}}\|\pi) \downarrow$ | $\widehat{\text{KL}}(\mathbb{P}^u\|\mathbb{P}^*) \downarrow$ | Abs. err. of $\log \widehat{Z} \downarrow$ |
|---|---|---|---|---|---|---|---|
| RERF | ✓ | 0.9196 | 0.0320 | 0.0200 | 0.4071 | 0.0263 | 0.00788 |
| | ✗ | 0.9276 | 0.8692 | 2.0487 | 6.7966 | 0.0352 | 2.00995 |
| LV | ✓ | 0.9722 | 0.0177 | **0.0098** | **0.1864** | 0.0092 | **0.00257** |
| | ✗ | 0.9594 | 0.1515 | 0.1619 | 0.1933 | 0.0164 | 0.13500 |
| CE | ✓ | 0.9855 | **0.0147** | 0.0138 | 2.8388 | 0.0098 | 0.00259 |
| | ✗ | 0.9694 | 0.0159 | 0.0287 | 60.6136 | 0.0259 | 0.00887 |
| WDCE | ✓ | 0.9465 | 0.0418 | 0.0282 | 1.6582 | 0.0336 | 0.00373 |
| | ✗ | **0.9927** | 0.1365 | 0.1505 | 2.4919 | **0.0057** | 0.13001 |
| Baseline (MH) | / | 0.0068 | 0.0044 | 0.0418 | / | / | |

TV distance, KL divergence, and $\chi^2$ divergence reported in Tabs. 4 and 5 as these metrics in general do not suffer from the aforementioned problem.

Additionally, it is worth mentioning that this warm-up training strategy is significantly different from the warm-up used in LEAPS. The code implementation of LEAPS also has a warm-up stage in the first 20k steps during the training, where they gradually increase $t_{\text{final}}(k)$ from 0 to 1 while the step $k$ increases from 0 to the maximum warm-up steps. During the warm-up phase, the PINN objective of LEAPS is evaluated on time up to $t_{\text{final}}(k)$. Unlike LEAPS, we do not use partial trajectories for training during the warm-up phase, which is a major difference.

### D.2.5  Ablation Study of the Choice of Number of Replicates $R$ in $\mathcal{F}_{\text{WDCE}}$

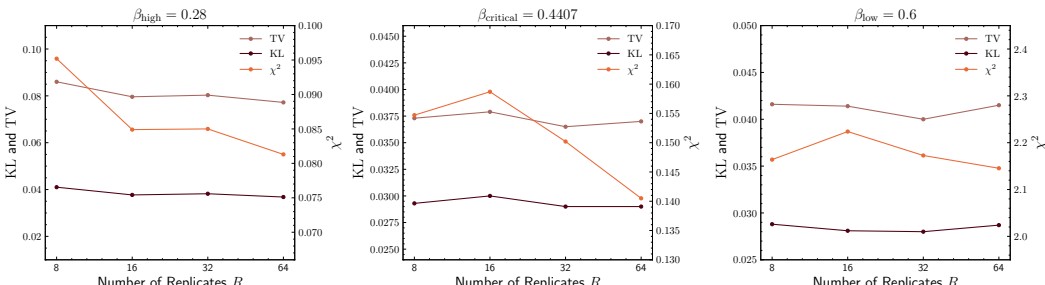

Figure 3: Visualization of learning performance across the number of replicates $R$ for learning $4 \times 4$ Ising model with $J = 1$ and $h = 0.1$ using the WDCE loss. The metrics reported are TV$(\widehat{p}_{\text{samp}}, \pi)$, KL$(\widehat{p}_{\text{samp}}\|\pi)$, and $\chi^2(\widehat{p}_{\text{samp}}\|\pi)$.

We provide an ablation study of the number of replicates $R$ in the WDCE loss in Fig. 3, showing that the performance of training is insensitive to $R$. Therefore, we only choose a relatively small number $R = 8$ in learning the $16 \times 16$ distributions to guarantee the efficiency as well as the efficacy of the algorithm.

### D.3  Evaluation and Additional Results on the $16 \times 16$ Ising model

#### D.3.1  Evaluation of the $16 \times 16$ Learned Models

As the cardinality of the state space is much larger ($2^{L^2=256} \approx 10^{77}$), we cannot exactly compute the whole probability distribution. Instead, we report the values and errors for observables well-studied in statistical physics such as magnetization and 2-point correlation. For any probability distribution $\nu$ on $\{\pm 1\}^\Lambda$ (we recall that $\Lambda = \{1, ..., L\}^2$), these observables can be defined as follows.

**Magnetization.**  The magnetization of a state $i \in \Lambda$ under $\nu$ is defined as $M_\nu(i) := \mathbb{E}_{\nu(x)} x^i$. The average magnetization of all states is defined as $M_\nu := \mathbb{E}_{\nu(x)} \left[ \frac{1}{L^2} \sum_i x^i \right]$. For our target distribution $\pi$ in (17), $M_\pi(i) = 0$ for all $i \in \Lambda$ since $H(x) = H(-x)$ when $h = 0$.

Moreover, we can define **row-wise magnetization** (i.e., along the $x$-axis) $M_\nu^{\text{row}}$ or **column-wise magnetization** (i.e., along the $y$-axis) $M_\nu^{\text{col}}$ by summing up the magnetization of states on the subset of a specific row or column on the square lattice. They are formally defined as,

$$M_\nu^{\text{row}}(k) = \sum_{i \in \text{row}(k)} M_\nu(x^i), \; M_\nu^{\text{col}}(k) = \sum_{i \in \text{col}(k)} M_\nu(x^i), \quad \forall k \in \left\{ -\left\lfloor \frac{L}{2} \right\rfloor, ..., 0, ..., \left\lfloor \frac{L}{2} \right\rfloor \right\}. \tag{25}$$

Based on these observables, we define the **absolute magnetization error** (abbreviated as Mag. in Tab. 1) of the learned distribution $\nu$ compared with the target distribution $\pi$ as

$$\frac{1}{2L} \sum_{k \in \left\{ -\left\lfloor \frac{L}{2} \right\rfloor, ..., 0, ..., \left\lfloor \frac{L}{2} \right\rfloor \right\}} \left| M_\nu^{\text{row}}(k) - M_\pi^{\text{row}}(k) \right| + \left| M_\nu^{\text{col}}(k) - M_\pi^{\text{col}}(k) \right|. \tag{26}$$

**2-Point Correlation.** The 2-point correlation of two states $i, j \in \Lambda$ under $\nu$ is defined as $C_\nu(i, j) = \mathbb{E}_{\nu(x)}[x^i x^j] - \mathbb{E}_{\nu(x)}[x^i] \mathbb{E}_{\nu(x)}[x^j]$. For our target distribution $\pi$ in (17), $C_\pi(i, j) = \mathbb{E}_{\pi(x)}[x^i x^j]$ due to the zero magnetization.

The average magnetization of all states differing with vector $r \in \left\{ -\left\lfloor \frac{L}{2} \right\rfloor, ..., 0, ..., \left\lfloor \frac{L}{2} \right\rfloor \right\}^2$ is defined as

$$C_\nu(r) = \frac{1}{L^2} \left( \sum_i \mathbb{E}_{\nu(x)}[x^i x^{i+r}] - \mathbb{E}_{\nu(x)}[x^i] \mathbb{E}_{\nu(x)}[x^{i+r}] \right),$$

where the addition is performed element-wise under modulo $L$ due to the periodic boundary condition.

Similar to the magnetization, we can define the **row-wise correlation** (i.e., along the $x$-axis) and **column-wise correlation** (i.e., along the $y$-axis) by summing up the 2-point correlations between states on the subset of specific pairs of rows or columns. They are formally defined as

$$C_\nu^{\text{row}}(k, l) = \sum_{\substack{i \in \text{row}(k), \; j \in \text{row}(l), \\ i,j \text{ same col}}} C_\nu(i, j), \quad C_\nu^{\text{col}}(k, l) = \sum_{\substack{i \in \text{col}(k), \; j \in \text{col}(l), \\ i,j \text{ same row}}} C_\nu(i, j), \tag{27}$$

where $k, l \in \left\{ -\left\lfloor \frac{L}{2} \right\rfloor, ..., 0, ..., \left\lfloor \frac{L}{2} \right\rfloor \right\}$. Based on these observables, we can also define the **absolute 2-point correlation error** (abbreviated as Corr. in Tab. 1) for the learned distribution $\nu$ compared with the target distribution $\pi$ by

$$\frac{1}{L^2} \sum_{(k,l)} \left| C_\nu^{\text{row}}(k, l) - C_\pi^{\text{row}}(k, l) \right| + \left| C_\nu^{\text{col}}(k, l) - C_\pi^{\text{col}}(k, l) \right|. \tag{28}$$

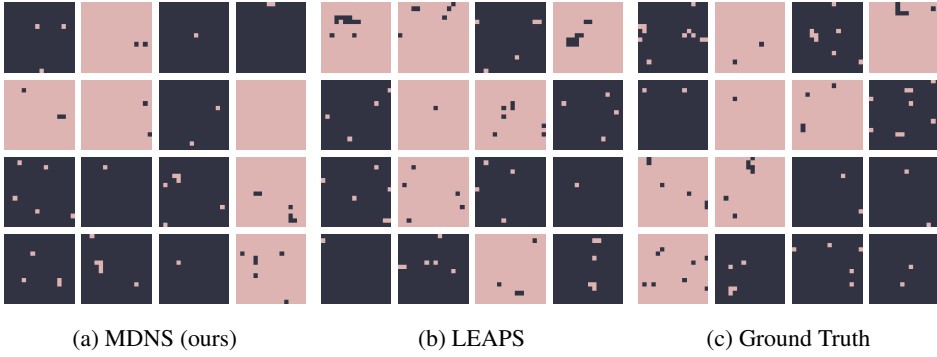

(a) MDNS (ours)      (b) LEAPS      (c) Ground Truth

Figure 4: Visualization of non-cherry-picked samples from the learned $16 \times 16$ Ising model with $J = 1$, $h = 0$, and $\beta_{\text{low}} = 0.6$. (a) MDNS. (b) LEAPS. (c) Ground Truth (simulated with SW algorithm).

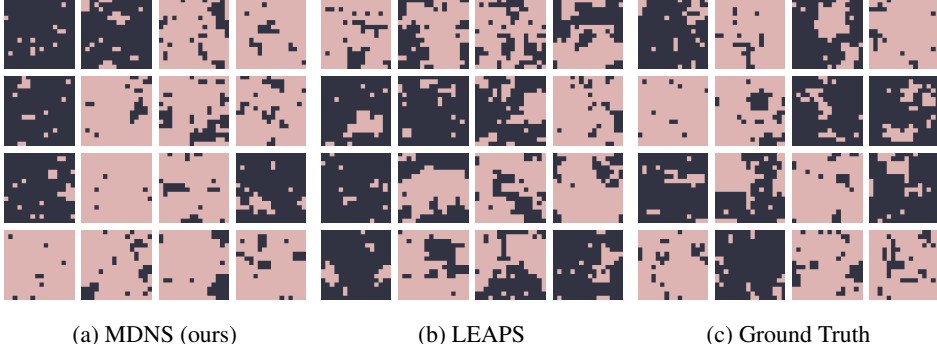

(a) MDNS (ours)          (b) LEAPS          (c) Ground Truth

Figure 5: Visualization of non-cherry-picked samples from the learned $16 \times 16$ Ising model with $J = 1$, $h = 0$, and $\beta_{\text{critical}} = 0.4407$. (a) MDNS. (b) LEAPS. (c) Ground Truth (simulated with SW algorithm).

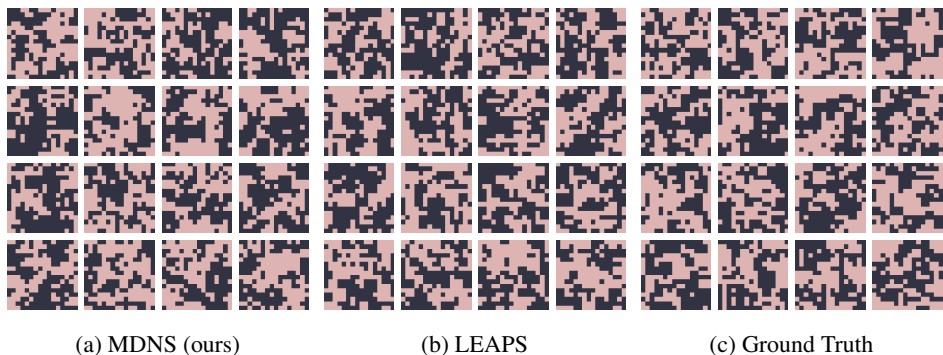

(a) MDNS (ours)          (b) LEAPS          (c) Ground Truth

Figure 6: Visualization of non-cherry-picked samples from the learned $16 \times 16$ Ising model with $J = 1$, $h = 0$, and $\beta_{\text{high}} = 0.28$. (a) MDNS. (b) LEAPS. (c) Ground Truth (simulated with SW algorithm).

### D.3.2 Training Curves for Learning the $16 \times 16$ Ising Model

The training ESS curves of the score models for learning the $16 \times 16$ Ising model are provided in Fig. 7, where the models for $\beta_{\text{critical}}$ and $\beta_{\text{low}}$ are initialized at the 20k-step checkpoint of the model trained for $\beta_{\text{high}}$ to implement the warm-up training strategy. Due to the gigantic mismatch between the reward functions used in the warm-up phase and the formal training phase, the ESS experiences a sudden drop to near 0 at iteration 20k for $\beta_{\text{low}}$ and $\beta_{\text{critical}}$.

### D.3.3 Additional Results for Learning $16 \times 16$ Ising Model

To provide a more comprehensive evaluation of our learned model for $16 \times 16$ Ising model, we visualize the generated samples in Figs. 4 to 6. From the plots, we can see the high fidelity of our generated samples as they follow a statistically similar pattern to the ground truth samples produced by running the SW algorithm. We also plot the 2-point correlation function along the $y$-axis in Fig. 8. Additionally, we visualize a distribution of absolute error of the estimated $M^{\text{row}}$ and $M^{\text{col}}$ to ground truth along each column or row of the square lattice in Fig. 9. All of these results demonstrate a superior performance of our proposed MDNS compared to other benchmarks.

### D.4 Effects of Preconditioning in MDNS Training

In learning a continuous neural sampler to sample from a target distribution $\nu \propto e^{-V}$ on $\mathbb{R}^d$, typical approaches such as PIS [ZC22] and DDS [VGD23] leverage the score information $\nabla \log \pi$ in the neural network, which is known as **preconditioning**. Efficient preconditioning is shown to facilitate the convergence of training and achieve smaller sampling errors [He+25]. In our experiments, we already achieve good performance *without* any information of the target distribution when

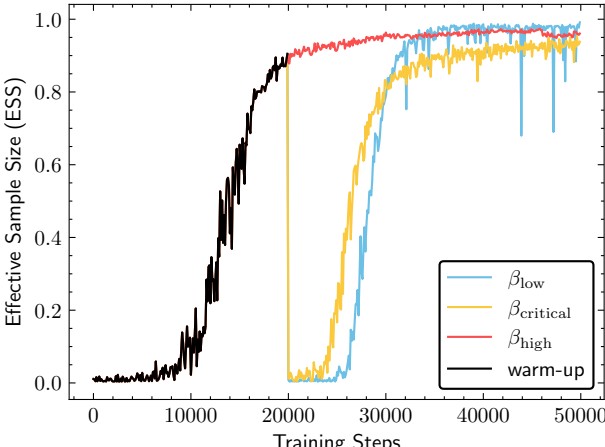

Figure 7: Training curves of the effective sample size (ESS) for learning $16 \times 16$ Ising model at different temperatures. A closer value to 1 generally indicates a better performance.

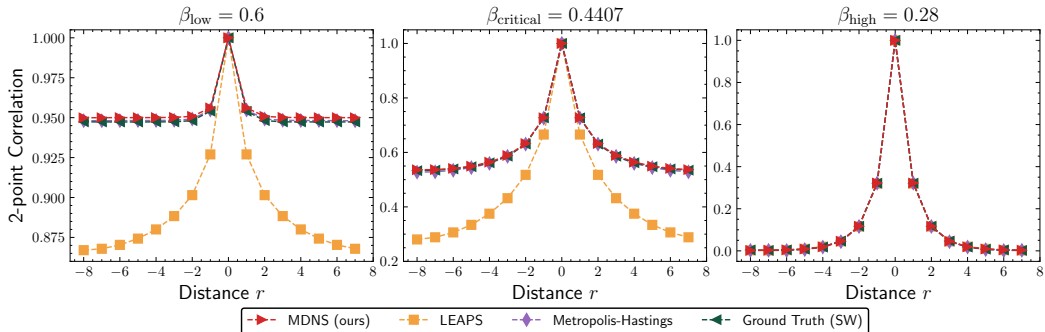

Figure 8: Average of 2-point correlation $C^{\mathrm{col}}(k, k+r)$ of samples from $16 \times 16$ Ising model.

parameterizing the score model; here, we explore a way to do preconditioning for learning to sample from the Ising model, and compare its effectiveness during training.

Recall that in the target distribution of the Ising model (17), the distribution of $x^i$ conditional on all the remaining positions $x^{-i} := (x^j : j \neq i)$ can be computed as follows:

$$\pi(x^i | x^{-i}) \propto_{x^i} e^{\beta J\left(\sum_{j:j\sim i} x^j\right)x^i + \beta h x^i},$$

$$\implies \pi(x^i | x^{-i}) = \frac{e^{\beta J\left(\sum_{j:j\sim i} x^j\right)x^i + \beta h x^i}}{e^{\beta J \sum_{j:j\sim i} x^j + \beta h} + e^{-\beta J \sum_{j:j\sim i} x^j - \beta h}}, \; x^i \in \{\pm 1\}.$$

As the model needs to learn the conditional distribution of $x^i$ given a *partially masked* $x^{-i}$, we can naturally treat the mask value as $0$ and use the above formula to approximate the conditional distribution. Specifically, given a partially masked $x \in \mathcal{X} = \{\pm 1, \mathbf{M} = 0\}^\Lambda$ with $x^i = \mathbf{M}$, we use the following form of preconditioning:

$$\Pr_{X\sim\pi}(X^i = n | X^{\mathrm{UM}} = x^{\mathrm{UM}}) \approx s_\theta(x)_{i,n}$$

$$:= \texttt{softmax}(\Phi_\theta(x) + P(x), \texttt{dim} = -1)_{i,n}, \; i \in \Lambda, \; n \in \{\pm 1\},$$

where $\Phi_\theta : \mathcal{X} \to \mathbb{R}^{(L \times L) \times 2}$ is a free-form neural network and the precondition matrix $P : \mathcal{X} \to \mathbb{R}^{(L \times L) \times 2}$ is defined as $P(x)_{i,-1} = \log \pi(x^i \leftarrow -1 | x^{-i})$ and $P(x)_{i,1} = \log \pi(x^i \leftarrow 1 | x^{-i})$.

In Fig. 10, we train models from scratch to learn from $16 \times 16$ Ising model under $\beta_{\mathrm{high}}$ and $\beta_{\mathrm{critical}}$ using the WDCE loss, both with and without preconditioning. It is obvious that applying preconditioning facilitates the convergence of the training in terms of the ESS. However, unlike in the case on

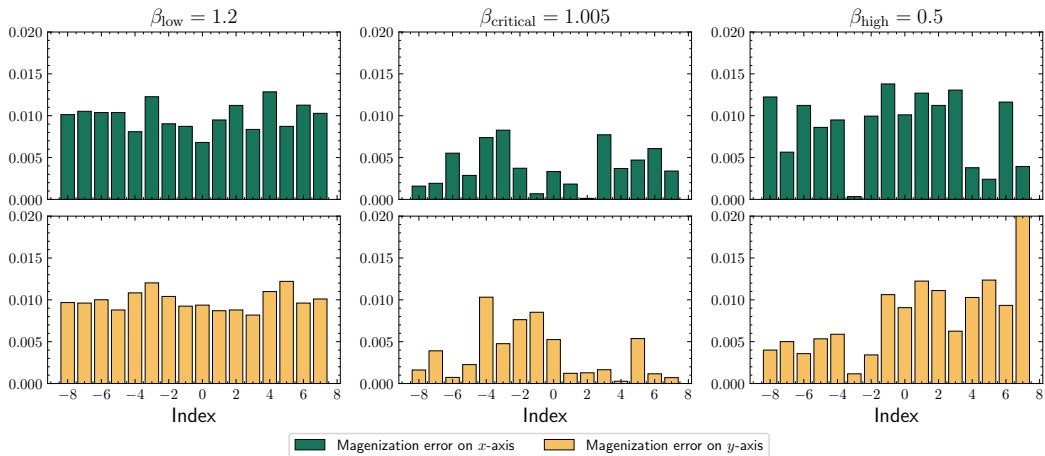

Figure 9: Distribution of the the absolute error $M^{\text{row}}(k)$ and $M^{\text{col}}(k)$ to the ground values against index $k$ for $16 \times 16$ Ising model.

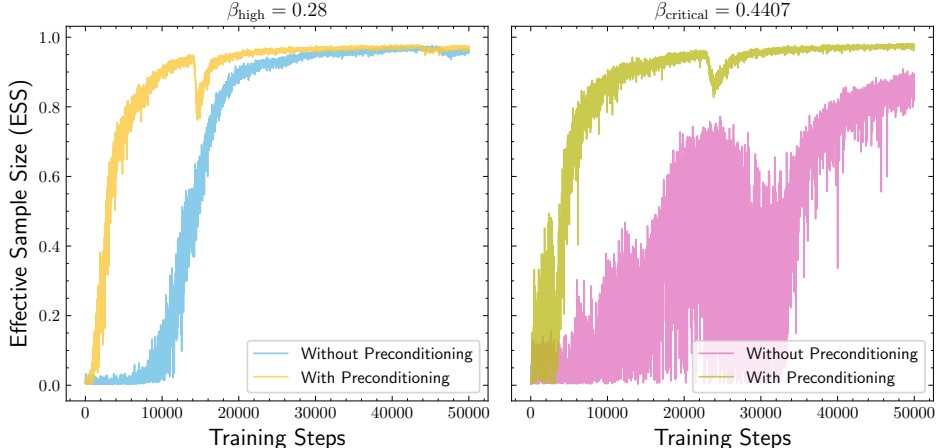

Figure 10: Comparison of learning a $16 \times 16$ Ising model with $J = 1$ and $h = 0$ using the WDCE loss, with and without preconditioning.

$\mathbb{R}^d$ where we can directly leverage the score information $\nabla \log \pi$, the above example heavily replies on the availability of the closed-form solution of the conditional distributions $\{\pi(x^i | x^{-i}), \forall i \in \Lambda\}$, which seriously limits the applicability of this preconditioning method. The study of preconditioning methodologies that are capable of dealing with target distributions whose conditional distributions are unavailable is left for future work.

### D.5 Failure of DRAKES in Learning Neural Samplers.

Finally, we argue that while DRAKES [Wan+25a] might be useful for fine-tuning a pretrained masked discrete diffusion model to maximize a certain reward function, it is not suitable for learning a diffusion sampler due to the error of approximating states using the Gumbel softmax trick.

The training dynamics of the $4 \times 4$ Ising model with $J = 1$, $h = 0.1$, $\beta_{\text{high}} = 0.28$ using DRAKES are presented in Figs. 11 and 12, where we use the same model structure, training configurations, and evaluation methods as in the experiment of Tab. 2, except that we now use a smaller learning rate of $0.0001$ with gradient clipping for numerical stability and train for 3000 steps. We test two different Gumbel softmax temperatures, 1 and 0.1, to show the possible effect of the Gumbel softmax temperature. As DRAKES leverages the Euler sampler [LME24; Ou+25] in training, we use $32(= 2L^2)$ steps to generate each sample and let the gradient back-propagate over the whole

trajectory with all 32 intermediate states without truncation. As shown in the figures, while the loss decreases and reward (defined by $-\beta_{\text{high}}H(x)$ in (17)) increases during training, indicating that the model is learning, the ESS remains at a low level, which means the learned model is not able to sample from the correct target distribution.

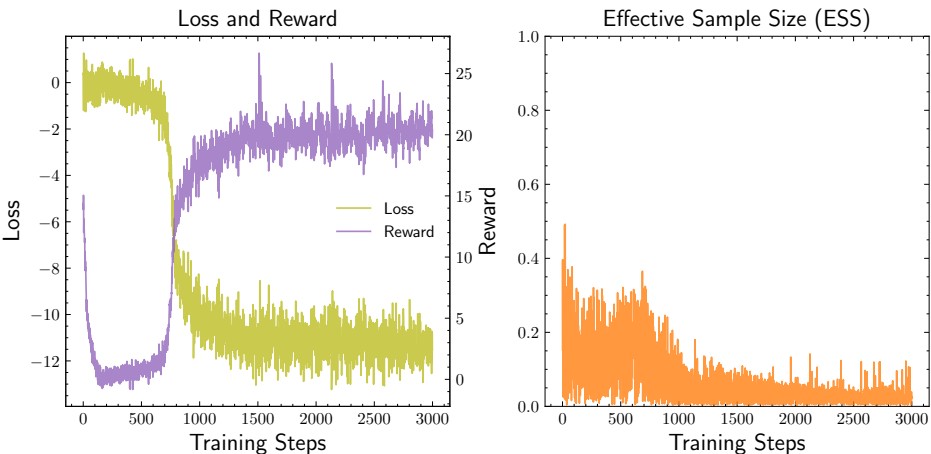

Figure 11: Learning $4 \times 4$ Ising model with $J = 1$, $h = 0.1$, and $\beta_{\text{high}} = 0.28$ via DRAKES (with Gumbel softmax temperature 1). Reward corresponds to $-\beta_{\text{high}}H(x)$.

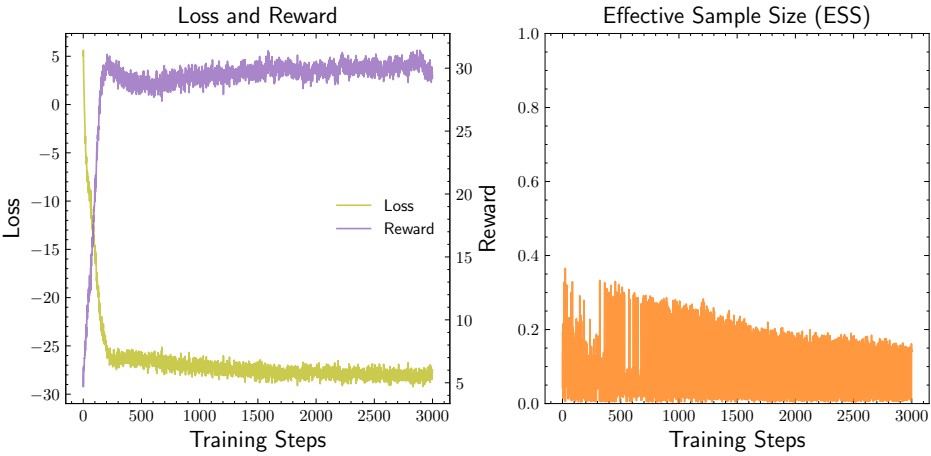

Figure 12: Learning $4 \times 4$ Ising model with $J = 1$, $h = 0.1$, and $\beta_{\text{high}} = 0.28$ via DRAKES (with Gumbel softmax temperature 0.1). Reward corresponds to $-\beta_{\text{high}}H(x)$.

# E   Experimental Details and Additional Results: Learning Potts model

## E.1   General Training Hyperparameters and Model Architecture

**Model backbone.**   Similar to the experiments on Ising model, we also use ViT with 2-dimensional rotary position embedding to serve as the backbone for the score model. To ensure enough model representation power, we adopt a slightly larger model with a 128-dimensional embedding space, 4 blocks and 4 attention heads, which sum up to 829k trainable parameters in total.

**Training.**   Among all the training tasks for Potts model, we choose the batch size as 256, and use the AdamW optimizer [LH19] with a constant learning rate of 5e−4. Like in the experiment of Ising model, we also use EMA to stabilize the training, with a decay rate of 0.9999. All experiments of learning the Potts model are run on an NVIDIA RTX A100. For $16 \times 16$ Potts model of all three

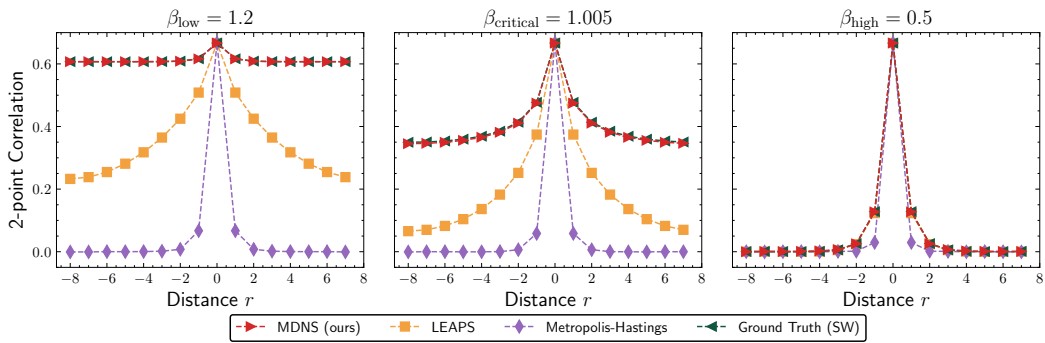

Figure 13: Average of 2-point correlation $C^{\mathrm{col}}(k, k + r)$ of samples from $16 \times 16$ Potts model.

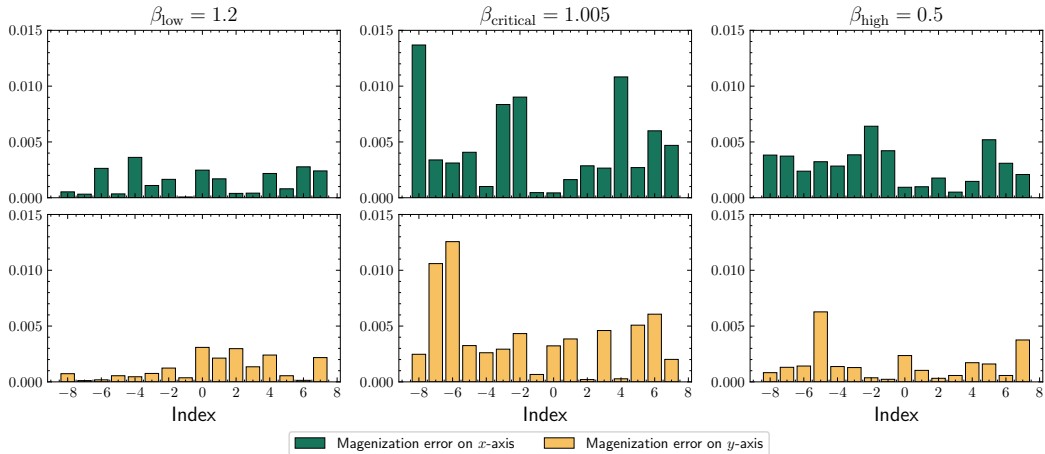

Figure 14: Distribution of the the absolute error $M^{\mathrm{row}}(k)$ and $M^{\mathrm{col}}(k)$ to the ground values against index $k$ for $16 \times 16$ Potts model.

temperatures, we use $\mathcal{F}_{\mathrm{WDCE}}$ with a resampling frequency $k = 10$ and replicates $R = 8$ for a total of 100k iterations, among which 30k is warm-up training. The total training time is about 20 hours.

**Generating baseline and ground truth samples.** For the learning-based baseline, we train LEAPS on $16 \times 16$ the Potts model for up to 100k steps for each temperature, which is comparable to, if not more than, our requirement on this task, to ensure a fair comparison. We also run the MH algorithm for a sufficiently long time to serve as a baseline. For $L = 16$, we use a batch size 32, and warm up the algorithm with $2^{27}$ burn-in iterations. After that, we collect the samples every $2^{16}$ steps to ensure sufficient mixing, and collect for 128 rounds, so the final number of samples is $2^{12}$. Note that for the Potts model with $L = 16$, the MH algorithm is not capable of sampling from the correct distribution even at the high temperature, and thus we have to resort to the SW algorithm to generate ground truth samples. We use a batch size 128, and warm up the algorithm with $2^{16}$ burn-in iterations. After that, we collect samples every 128 steps to ensure sufficient mixing of the chain, and collect for 32 rounds to gather a final number of total $2^{12}$ samples. The MH and SW sampling takes about the same time as the Ising model on a CPU.

### E.2 Evaluation and Additional Results on $16 \times 16$ **Potts model**

Similar to the case of $16 \times 16$ Ising model, a $16 \times 16$ Potts model has an intractably large cardinality of the state space $3^{L^2 = 256} \approx 10^{122}$, so we cannot compute the exact distribution and thus rely on values of observables in statistical physics to evaluate the performance. As the Potts model is a generalization of the Ising model, statistical physics observables such as magnetization and 2-point

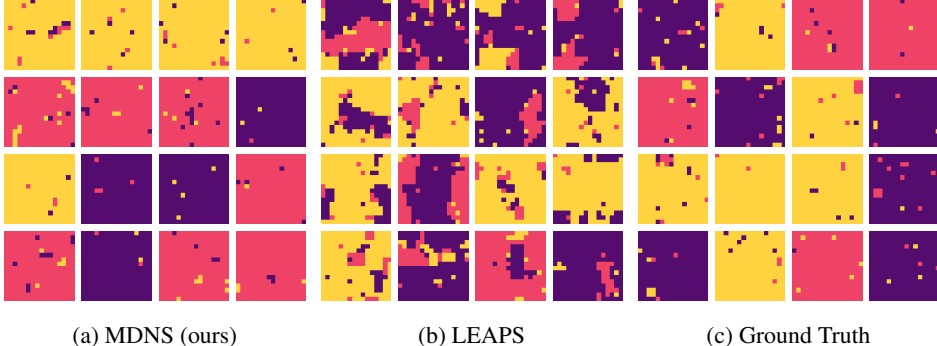

| (a) MDNS (ours) | (b) LEAPS | (c) Ground Truth |

Figure 15: Visualization of non-cherry-picked samples from the learned $16 \times 16$ Potts model with $J = 1$, $q = 3$, and $\beta_{\text{low}} = 1.2$. (a) MDNS. (b) LEAPS. (c) Ground Truth (simulated with SW algorithm).

correlation are also well defined, with different definitions and formulae but share a similar idea in spirit.

For any probability distribution $\nu$ on $\{1, ..., q\}^\Lambda$ (we recall that $\Lambda = \{1, ..., L\}^2$),

**Magnetization.** The **magnetization** of a state $i \in \Lambda$ under $\nu$ given $n$ i.i.d. samples from $\nu$ is defined as

$$M_\nu^{\text{potts}}(i) := \frac{q \max_{1 \leq c \leq q}(n_c^i/n) - 1}{q - 1},$$

where $n_c^i$ is the number of samples whose $i$-th element is $c$. The average magnetization of all states is defined as $M_\nu := \frac{1}{L^2} \sum_i M_\nu^{\text{potts}}(i)$.

Similar to the case of Ising model, we can also define **row-wise magnetization** (i.e., along the $x$-axis) or **column-wise magnetization** (i.e., along the $y$-axis) by summing the magnetization of states on the subset of a specific row or column on the square lattice. They are formally defined as

$$M_\nu^{\text{row}}(k) = \sum_{i \in \text{row}(k)} M_\nu^{\text{potts}}(x^i), \quad M_\nu^{\text{col}}(k) = \sum_{i \in \text{col}(k)} M_\nu^{\text{potts}}(x^i), \quad \forall k \in \left\{ -\left\lfloor \frac{L}{2} \right\rfloor, ..., 0, ..., \left\lfloor \frac{L}{2} \right\rfloor \right\}.$$
(29)

Based on these observables, we define the absolute magnetization error (abbreviated as Mag. in Tab. 3) for learnt distribution $\nu$ by computing the following

$$\frac{1}{2L} \sum_{k \in \left\{ -\left\lfloor \frac{L}{2} \right\rfloor, ..., 0, ..., \left\lfloor \frac{L}{2} \right\rfloor \right\}} \left| M_\nu^{\text{row}}(k) - M_\pi^{\text{row}}(k) \right| + \left| M_\nu^{\text{col}}(k) - M_\pi^{\text{col}}(k) \right|,$$
(30)

where $\pi$ is the target optimal distribution and $\nu$ is the learned distribution.

**2-Point Correlation.** Potts model has the following definition of 2-point correlation:

$$C_\nu^{\text{potts}}(i, j) = \frac{1}{L^2} \sum_i \mathbb{E}_{\nu(x)} \left[ 1_{x^i = x^j} - \frac{1}{q} \right], \quad \forall i, j \in \Lambda.$$

Comparing to the definition for Ising model, the above definition has an additional offset $\frac{1}{q}$, which causes the maximal correlation to be strictly smaller than 1.

We can define **row-wise correlation** (i.e., along the $x$-axis) or **column-wise correlation** (i.e., along the $y$-axis) by summing the correlation between states on the subset of specific pairs of rows or columns. It's formally defined as,

$$C_\nu^{\text{row}}(k, l) = \sum_{\substack{i \in \text{row}(k), \ j \in \text{row}(l), \\ i, j \text{ same col}}} C_\nu^{\text{potts}}(i, j), \quad C_\nu^{\text{col}}(k, l) = \sum_{\substack{i \in \text{col}(k), \ j \in \text{col}(l), \\ i, j \text{ same row}}} C_\nu^{\text{potts}}(i, j), \quad (31)$$

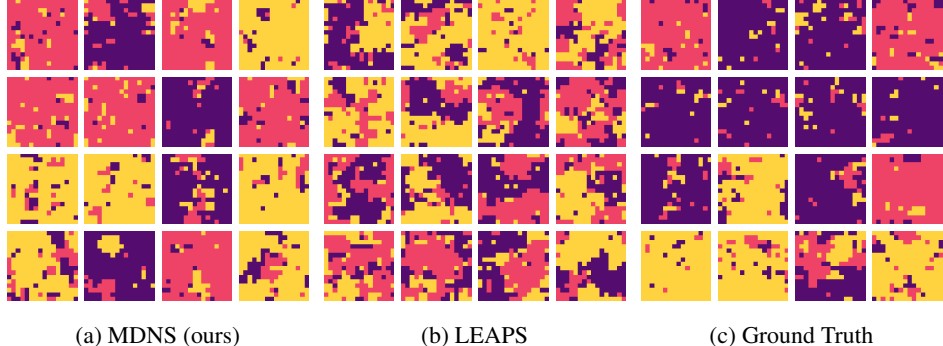

(a) MDNS (ours)      (b) LEAPS      (c) Ground Truth

Figure 16: Visualization of non-cherry-picked samples from the learned $16 \times 16$ Potts model with $J = 1$, $q = 3$, and $\beta_{\text{critical}} = 1.005$. (a) MDNS. (b) LEAPS. (c) Ground Truth (simulated with SW algorithm).

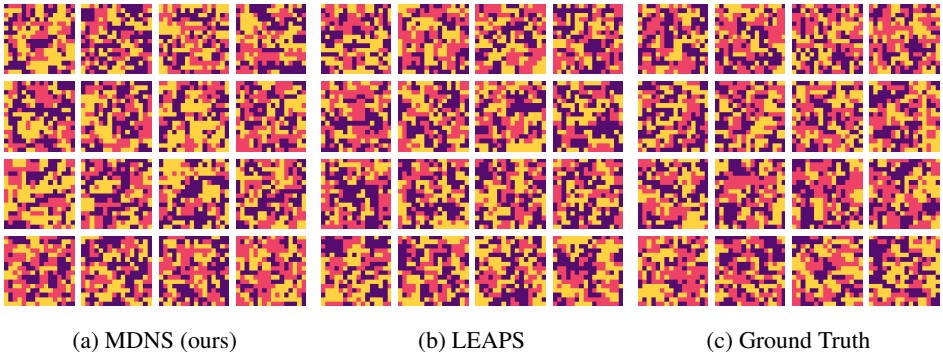

(a) MDNS (ours)      (b) LEAPS      (c) Ground Truth

Figure 17: Visualization of non-cherry-picked samples from the learned $16 \times 16$ Potts model with $J = 1$, $q = 3$, and $\beta_{\text{high}} = 0.5$. (a) MDNS. (b) LEAPS. (c) Ground Truth (simulated with SW algorithm).

where $k, l \in \left\{ -\left\lfloor \frac{L}{2} \right\rfloor, ..., 0, ..., \left\lfloor \frac{L}{2} \right\rfloor \right\}$. Based on these observables, we can also define the absolute correlation error (abbreviated as Corr. in Tab. 3) for learned distribution $\nu$ by

$$\frac{1}{L^2} \sum_{(k,l)} \left| C_\nu^{\text{row}}(k, l) - C_\pi^{\text{row}}(k, l) \right| + \left| C_\nu^{\text{col}}(k, l) - C_\pi^{\text{col}}(k, l) \right|. \tag{32}$$

### E.3 Training Curves for Learning $16 \times 16$ Potts Model

The training ESS curves of the model for learning the $16 \times 16$ Potts model are provided in Fig. 18, where the models for $\beta_{\text{critical}}$ and $\beta_{\text{low}}$ are initialized at the 30k-step checkpoint of the model trained for $\beta_{\text{high}}$ to implement the warm-up training strategy. Again, due to the mismatch between reward functions at different temperatures, the ESS experiences a sudden drop to near 0 at iteration 30k for $\beta_{\text{low}}$ and $\beta_{\text{critical}}$.

### E.4 Additional Results for $16 \times 16$ Potts Model

We visualize the generated samples in Figs. 15 to 17. From the plots, we can see the high fidelity of our generated samples as they follow a statistically similar pattern to the ground truth samples produced by running the SW algorithm. We also plot the average 2-point correlation function along the $y$-axis in Fig. 13, and visualize the distribution of absolute error of the estimated $M^{\text{row}}$ and $M^{\text{col}}$ to ground truth along each column or row of the square lattice in Fig. 14. These results suggest that MDNS manages to learn to sample from multimodal, high-dimensional discrete distributions accurately.

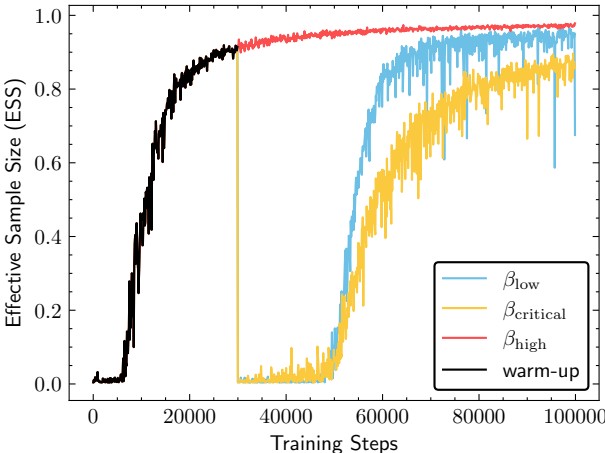

Figure 18: Training curves of effective sample size (ESS) of $16 \times 16$ Potts model under different temperatures. A closer value to 1 generally indicates a better performance.

## F    UDNS: Extension of MDNS to Uniform Discrete Diffusion Models

In this section, we discuss an extension of our MDNS framework to uniform discrete diffusion models [LME24; Sch+25]. We present the theory in App. F.1 and discuss the strategy of preconditioning in App. F.2. Experimental results are presented in App. F.3.

### F.1    Theory of Uniform Diffusion Neural Sampler

We choose $T = 1$ and construct the reference path measure $\mathbb{P}^0$ by the CTMC that always keeps $p_{\text{unif}}(x) = \frac{1}{N^D} 1_{x \in \mathcal{X}_0}$ at all times $t \in [0, 1]$. This can be achieved by initializing $\mathbb{P}^0_0 = p_{\text{unif}}$ and setting the generator as $Q^0_t(x, x^{d \leftarrow n}) = \frac{\gamma(t)}{N}, \forall n \neq x^d$. Note that each dimension evolves independently under $\mathbb{P}^0$. Let $\overline{\gamma}(t) = \int_t^1 \gamma(s) \mathrm{d}s$. One can compute the following transition distribution from time $t$ to 1:

$$\mathbb{P}^0_{1|t}(x_1|x_t) = \prod_{d=1}^{D} \left( \mathrm{e}^{-\overline{\gamma}(t)} 1_{x_1^d = x_t^d} + \frac{1 - \mathrm{e}^{-\overline{\gamma}(t)}}{N} \right). \tag{33}$$

By similarly assuming $\overline{\gamma}(0) = \infty$ like in Lem. 2, we can guarantee that $\mathbb{P}^0_{1|0}(x_1|x_0) = p_{\text{unif}}(x_1)$, so under $\mathbb{P}^0$, $X_0$ and $X_1$ are independent, making the reference process memoryless. A direct implication is $\mathrm{e}^{V_0(\cdot)} = \mathbb{E}_{\mathbb{P}^0}[\mathrm{e}^{r(X_1)}|X_0 = \cdot] = \mathbb{E}_{p_{\text{unif}}} \mathrm{e}^r = Z$ is a constant, just like the mask case.

Moreover, from Lem. 8 we have $\mathbb{P}^*_t(x) = \frac{1}{Z} \mathbb{P}^0_t(x) \mathrm{e}^{V_t(x)} = \frac{1}{ZN^D} \mathrm{e}^{V_t(x)}$, so (8) now reads

$$Q^*_t(x, x^{d \leftarrow n}) = Q^0_t(x, x^{d \leftarrow n}) \frac{\mathrm{e}^{V_t(x^{d \leftarrow n})}}{\mathrm{e}^{V_t(x)}} = \frac{\gamma(t)}{N} \frac{\mathbb{P}^*_t(x^{d \leftarrow n})}{\mathbb{P}^*_t(x)}, \ \forall n \neq x^d.$$

We can thus parameterize $Q^u_t(x, x^{d \leftarrow n}) = \frac{\gamma(t)}{N} s_\theta(x, t)_{d,n}$, where the neural network $s_\theta$ takes $x \in \mathcal{X}_0$ and $t \in [0, 1]$ as input and outputs a non-negative $D \times N$ matrix.

We approximate the continuous-time process by the following (naïve) Euler discretization scheme: for $k = 0, 1, ..., K - 1$, let $\Delta t = \frac{1}{K}$ and $t_k = k \Delta t$. To sample from $\mathbb{P}^u$, first sample $X_0 \sim p_{\text{unif}}$, and then, for $k = 0, 1, ..., K - 1$, approximate the transition probability as

$$\mathbb{P}^u_{t_{k+1}|t_k}(x_{t_{k+1}}|x_{t_k}) \approx 1_{x_{t_{k+1}} = x_{t_k}} + \Delta t Q^u_{t_k}(x_{t_k}, x_{t_{k+1}}).$$

Here, we only allow $x_{t_k}$ and $x_{t_{k+1}}$ to differ *at most one* entry in order to correctly compute $Q^u_{t_k}(x_{t_k}, x_{t_{k+1}})$, as otherwise the value would be zero. The RN derivative $\frac{\mathrm{d}\mathbb{P}^*}{\mathrm{d}\mathbb{P}^u}(x)$ is similarly

approximated by

$$\log \frac{\mathrm{d}\mathbb{P}^*}{\mathrm{d}\mathbb{P}^u}(x) = \log \frac{\mathrm{d}\mathbb{P}^*}{\mathrm{d}\mathbb{P}^0}(x) - \log \frac{\mathrm{d}\mathbb{P}^u}{\mathrm{d}\mathbb{P}^0}(x)$$

$$\approx r(x_1) - \log Z - \sum_{k=0}^{K-1} \log \frac{\mathbb{P}^u_{t_{k+1}|t_k}(x_{t_{k+1}}|x_{t_k})}{\mathbb{P}^0_{t_{k+1}|t_k}(x_{t_{k+1}}|x_{t_k})} =: W^u(x) - \log Z. \quad (34)$$

Next, if $x_{t_{k+1}} \neq x_{t_k}$,

$$\log \frac{\mathbb{P}^u_{t_{k+1}|t_k}(x_{t_{k+1}}|x_{t_k})}{\mathbb{P}^0_{t_{k+1}|t_k}(x_{t_{k+1}}|x_{t_k})} \approx \log \frac{Q^u_{t_k}(x_{t_k}, x_{t_{k+1}})}{Q^0_{t_k}(x_{t_k}, x_{t_{k+1}})} = \log s_\theta(x_{t_k}, t_k)_{d,n} \quad \text{when } x_{t_{k+1}} = x_{t_k}^{d\leftarrow n}. \quad (35)$$

Otherwise, if $x_{t_{k+1}} = x_{t_k}$, we have

$$\log \frac{\mathbb{P}^u_{t_{k+1}|t_k}(x_{t_k}|x_{t_k})}{\mathbb{P}^0_{t_{k+1}|t_k}(x_{t_k}|x_{t_k})} \approx \log \frac{1 + \Delta t Q^u_{t_k}(x_{t_k}, x_{t_k})}{1 + \Delta t Q^0_{t_k}(x_{t_k}, x_{t_k})}$$

$$= \log \frac{1 - \Delta t \sum_{d=1}^D \sum_{n \neq x_{t_k}^d} Q^u_{t_k}(x_{t_k}, x_{t_k}^{d\leftarrow n})}{1 - \Delta t \sum_{d=1}^D \sum_{n \neq x_{t_k}^d} Q^0_{t_k}(x_{t_k}, x_{t_k}^{d\leftarrow n})}$$

$$= \log \frac{1 - \frac{\Delta t \gamma(t)}{N} \sum_{d=1}^D \sum_{n \neq x_{t_k}^d} s_\theta(x_{t_k}, t_k)_{d,n}}{1 - \frac{\Delta t \gamma(t)}{N} \sum_{d=1}^D \sum_{n \neq x_{t_k}^d} 1}. \quad (36)$$

Here, unlike in the proof of Lem. 1, we do not use Taylor expansion to approximate $\log(1 - \Delta t\star)$ by $-\Delta t\star$ in order not to introduce further approximation error. Notably, due to our parameterization of $Q^u_t$, this only requires one call to the score model $s_\theta$ without any specifically designed model architecture such as the locally equivariant network introduced in [HAJ25].

Now, with the approximation of RN derivative, we can easily derive the LV, CE and RERF losses just by plugging in $W^u(X)$ into (15). We can similarly derive the WDCE loss as follows. First, as $\mathbb{P}^0_t = p_{\text{unif}}$ for all $t$, the CTMC with path measure $\mathbb{P}^0$ is reversible, and hence we know from (33) that $\mathbb{P}^0_{1|t}(x_1|x_t) = \mathbb{P}^0_{t|1}(x_t|x_1)$. Next, due to the property that $\mathbb{P}^*(x) = \frac{1}{Z} e^{r(x_1)} \mathbb{P}^0(x)$ from (5), by conditioning on $x_1$, we have

$$\mathbb{P}^*_{t|1}(x_t|x_1) = \mathbb{P}^0_{t|1}(x_t|x_1) = \prod_{d=1}^D \left( e^{-\overline{\gamma}(t)} 1_{x_1^d = x_t^d} + \frac{1 - e^{-\overline{\gamma}(t)}}{N} \right), \quad (37)$$

which means we can easily sample from the transition kernel $\mathbb{P}^*_{t|1}(\cdot|x_1)$ by independently replacing each entry of $x_1$ with a token from $\text{Unif}\{1, ..., N\}$ with probability $1 - e^{-\overline{\gamma}(t)}$. To derive the WDCE loss, one can first prove the DCE trick [LME24, Thm. 3.4]: for any function $f$,

$$\mathbb{E}_{\mathbb{P}^*_1(X_1)\mathbb{P}^*_{t|1}(X_t|X_1)} \sum_{y \neq X_t} \frac{\mathbb{P}^*_t(y)}{\mathbb{P}^*_t(X_t)} f(X_t, t, y) = \mathbb{E}_{\mathbb{P}^*_1(X_1)\mathbb{P}^*_{t|1}(X_t|X_1)} \sum_{y \neq X_t} \frac{\mathbb{P}^*_{t|1}(y|X_1)}{\mathbb{P}^*_{t|1}(X_t|X_1)} f(X_t, t, y). \quad (38)$$

In fact, simple calculation shows that both sides equal $\sum_{X_t, y \neq X_t} \mathbb{P}^*_t(y) f(X_t, t, y)$. We can therefore derive the WDCE loss as follows:

$$\text{KL}(\mathbb{P}^*\|\mathbb{P}^u) = \mathbb{E}_{\mathbb{P}^*(X)} \int_0^1 \sum_{y \neq X_t} \left( Q^*_t \log \frac{Q^*_t}{Q^u_t} + Q^u_t - Q^*_t \right)(X_t, y)\mathrm{d}t \quad (\text{By Cor. 1})$$

$$= \mathbb{E}_{t, \mathbb{P}^*(X)} \sum_{y \neq X_t} (Q^u_t - Q^*_t \log Q^u_t)(X_t, y) + \text{const}$$

$$= \mathbb{E}_{t, \mathbb{P}^*(X)} \sum_d \sum_{n \neq X_t^d} (Q^u_t - Q^*_t \log Q^u_t)(X_t, X_t^{d\leftarrow n}) + \text{const},$$

where $t \sim \text{Unif}(0,1)$, and $\text{const}$ represents terms that do not depend on $\theta$. Next, we leverage the parameterization of $Q^u$ and $Q^*$ as well as the DCE trick (38):

$$\text{KL}(\mathbb{P}^*\|\mathbb{P}^u)$$

$$= \mathbb{E}_{t,\mathbb{P}_1^*(X_1)\mathbb{P}_{t|1}^*(X_t|X_1)} \frac{\gamma(t)}{N} \sum_d \sum_{n \neq X_t^d} \left( s_\theta(X_t,t)_{d,n} - \frac{\mathbb{P}_t^*(X_t^{d \leftarrow n})}{\mathbb{P}_t^*(X_t)} \log s_\theta(X_t,t)_{d,n} \right) + \text{const}$$

$$= \mathbb{E}_{t,\mathbb{P}_1^*(X_1)\mathbb{P}_{t|1}^*(X_t|X_1)} \frac{\gamma(t)}{N} \sum_d \sum_{n \neq X_t^d} \left( s_\theta(X_t,t)_{d,n} - \frac{\mathbb{P}_{t|1}^*(X_t^{d \leftarrow n}|X_1)}{\mathbb{P}_{t|1}^*(X_t|X_1)} \log s_\theta(X_t,t)_{d,n} \right) + \text{const}$$

$$= \mathbb{E}_{\mathbb{P}^{\bar{u}}(\overline{X})} \frac{\mathrm{d}\mathbb{P}^*}{\mathrm{d}\mathbb{P}^{\bar{u}}}(\overline{X}) \, \mathbb{E}_{t,\mathbb{P}_{t|1}^*(X_t|\overline{X}_1)} \left[ \frac{\gamma(t)}{N} \right.$$

$$\left. \sum_d \sum_{n \neq X_t^d} \left( s_\theta(X_t,t)_{d,n} - \frac{\mathbb{P}_{t|1}^*(X_t^{d \leftarrow n}|\overline{X}_1)}{\mathbb{P}_{t|1}^*(X_t|\overline{X}_1)} \log s_\theta(X_t,t)_{d,n} \right) \right] + \text{const},$$

We can simplify the ratio of conditional probabilities according to (37):

$$\frac{\mathbb{P}_{t|1}^*(X_t^{d \leftarrow n}|\overline{X}_1)}{\mathbb{P}_{t|1}^*(X_t|\overline{X}_1)} = \frac{\mathrm{e}^{-\overline{\gamma}(t)}1_{\overline{X}_1^d=n} + \frac{1-\mathrm{e}^{-\overline{\gamma}(t)}}{N}}{\mathrm{e}^{-\overline{\gamma}(t)}1_{\overline{X}_1^d=X_t^d} + \frac{1-\mathrm{e}^{-\overline{\gamma}(t)}}{N}}.$$

Finally, recall that $\frac{\mathrm{d}\mathbb{P}^*}{\mathrm{d}\mathbb{P}^{\bar{u}}}(\overline{X}) = \frac{1}{Z}\mathrm{e}^{W^{\bar{u}}(\overline{X})}$ can be computed via (34), and we can estimate $Z$ by $\mathbb{E}_{\overline{X} \sim \mathbb{P}^{\bar{u}}} \mathrm{e}^{W^{\bar{u}}(\overline{X})}$ as in both CE and WDCE losses for MDNS. We thus arrive at the WDCE loss for UDNS:

$$\mathcal{F}_{\text{WDCE}}(\mathbb{P}^u,\mathbb{P}^*) := \mathbb{E}_{\overline{X} \sim \mathbb{P}^{\bar{u}}} \frac{1}{Z}\mathrm{e}^{W^{\bar{u}}(\overline{X})} \, \mathbb{E}_{t,\mathbb{P}_{t|1}^*(X_t|\overline{X}_1)} \left[ \frac{\gamma(t)}{N} \right.$$

$$\left. \sum_d \sum_{n \neq X_t^d} \left( s_\theta(X_t,t)_{d,n} - \frac{\mathrm{e}^{-\overline{\gamma}(t)}1_{\overline{X}_1^d=n} + \frac{1-\mathrm{e}^{-\overline{\gamma}(t)}}{N}}{\mathrm{e}^{-\overline{\gamma}(t)}1_{\overline{X}_1^d=X_t^d} + \frac{1-\mathrm{e}^{-\overline{\gamma}(t)}}{N}} \log s_\theta(X_t,t)_{d,n} \right) \right]. \tag{39}$$

We summarize the training of the UDNS in Alg. 3. Here, `Resample_with_Unif` means for sample random variables $\{t^{(i,r)}\}_{1 \leq i \leq B, 1 \leq r \leq R} \overset{\text{i.i.d.}}{\sim} \text{Unif}(0,1)$, and for each $i$ and $r$, first randomly masking each entry of $X^{(i)}$ with probability $1 - \mathrm{e}^{-\overline{\gamma}(t^{(i,r)})}$, and then replacing each masked entry independently with a token from $\text{Unif}\{1,...,N\}$.

---

**Algorithm 3** Training of Uniform Diffusion Neural Sampler (UDNS)

---

**Require:** score model $s_\theta$, batch size $B$, training iterations $K$, reward function $r : \mathcal{X}_0 \to \mathbb{R}$, learning objective $\mathcal{F} \in \{\mathcal{F}_{\text{RERF}}, \mathcal{F}_{\text{LV}}, \mathcal{F}_{\text{CE}}, \mathcal{F}_{\text{WDCE}}\}$, (number of replicates of each sample $R$, resample frequency $k$ for $\mathcal{F}_{\text{WDCE}}$).
1: **for** step $= 1$ **to** $K$ **do**
2:     **if** $\mathcal{F} \in \{\mathcal{F}_{\text{RERF}}, \mathcal{F}_{\text{LV}}, \mathcal{F}_{\text{CE}}\}$ **then**
3:         $\{X^{(i)}, W^u(X^{(i)})\}_{1 \leq i \leq B} = \texttt{Sample\_Trajectories\_Unif}(B)$.        ▷ See Alg. 4.
4:         Compute $\mathcal{F}$ with $\{X^{(i)}, W^u(X^{(i)})\}_{1 \leq i \leq B}$.              ▷ See (15).
5:     **else if** $\mathcal{F} = \mathcal{F}_{\text{WDCE}}$ **then**
6:         **if** step $\mod k = 0$ **then**             ▷ Sample new trajectories every $k$ steps.
7:             $\{X^{(i)}, W^u(X^{(i)})\}_{1 \leq i \leq B} = \texttt{Sample\_Trajectories\_Unif}(B)$.
8:             Set replay buffer $\mathcal{B} \leftarrow \{X^{(i)}, W^u(X^{(i)})\}_{1 \leq i \leq B}$.
9:         $\{\widetilde{X}^{(i)}, W^u(\widetilde{X}^{(i)})\}_{1 \leq i \leq BR} = \texttt{Resample\_with\_Unif}(\mathcal{B}; R)$.
10:        Compute $\mathcal{F}_{\text{WDCE}}$ with $\{\widetilde{X}^{(i)}, W^u(\widetilde{X}^{(i)})\}_{1 \leq i \leq BR}$.
11:     Update the parameters $\theta$ based on the gradient $\nabla_\theta \mathcal{F}$.
    **return** trained score model $s_\theta$.

---

**Algorithm 4** `Sample_Trajectories_Unif`: Sample trajectories and compute weights for UDNS.

---

**Require:** score model $s_\theta$, reward function $r : \mathcal{X}_0 \to \mathbb{R}$, batch size $B$, number of time-intervals $K$, functions $\gamma$, early starting parameter $\epsilon \approx 0$.

1: Initialize uniformly random sequences $X_{t_0}^{(i)} \overset{\text{i.i.d.}}{\sim} p_{\text{unif}}$ and weights $W^{(i)} = 0$, $1 \le i \le B$.
2: Define $t_k = \epsilon + k\Delta t$, $0 \le k \le K$, where $\Delta t = \frac{1-\epsilon}{K}$. ▷ $\gamma(0) = \infty$ so do not initialize at $t_0 = 0$.
3: **for** $k = 0$ **to** $K - 1$ **do**
4:     Call the score model and get all the scores $\{s_\theta(X_{t_k}^{(i)}, t_k)\}_{1 \le i \le B}$.
5:     For each $1 \le i \le B$, sample an update from the approximate transition distribution: $X_{t_{k+1}}^{(i)} \leftarrow (X_{t_k}^{(i)})^{d \leftarrow n}$ with probability $\frac{\Delta t \gamma(t)}{N} s_\theta(X_{t_k}^{(i)}, t_k)_{d,n}$ for $n \ne (X_{t_k}^{(i)})^d$, and $X_{t_{k+1}}^{(i)} \leftarrow X_{t_k}^{(i)}$ with probability $1 - \frac{\Delta t \gamma(t)}{N} \sum_d \sum_{n \ne (X_{t_k}^{(i)})^d} s_\theta(X_{t_k}^{(i)}, t_k)_{d,n}$.
6:     For each $1 \le i \le B$, based on if $X_{t_{k+1}}^{(i)} \ne X_{t_k}^{(i)}$ or not, update weights $W^{(i)} \leftarrow W^{(i)} - \log \frac{\mathbb{P}_{t_{k+1}|t_k}^u(x_{t_{k+1}}|x_{t_k})}{\mathbb{P}_{t_{k+1}|t_k}^0(x_{t_{k+1}}|x_{t_k})}$ according to (35) or (36), respectively.
7:     For each $1 \le i \le B$, update weights with the final reward: $W^{(i)} \leftarrow W^{(i)} + r(X^{(i)})$.
    **return** pairs of sample and weights $\{X^{(i)}, W^u(X^{(i)}) := W^{(i)}\}_{1 \le i \le B}$.

---

Table 6: Results for learning $4 \times 4$ Ising model with $J = 1$, $h = 0.1$, and $\beta = 0.28$ using UDNS (with an ablation study of the effect of preconditioning), best in **bold**.

| Method | Use precond. | ESS ↑ | TV$(\widehat{p}_{\text{samp}}, \pi)$ ↓ | KL$(\widehat{p}_{\text{samp}}\|\pi)$ ↓ | $\chi^2(\widehat{p}_{\text{samp}}\|\pi)$ ↓ | $\widehat{\text{KL}}(\mathbb{P}^u\|\mathbb{P}^*)$ ↓ | Abs. err. of $\log \widehat{Z}$ ↓ |
|---|---|---|---|---|---|---|---|
| $\mathcal{F}_{\text{RERF}}$ | ✓ | 0.9671 | 0.0787 | 0.0357 | 0.0738 | 0.0182 | 1.45745 |
|  | ✗ | 0.9769 | 0.0726 | 0.0341 | 0.0696 | 0.0110 | 1.45376 |
| $\mathcal{F}_{\text{LV}}$ | ✓ | **0.9900** | **0.0710** | **0.0332** | **0.0647** | **0.0053** | 0.07752 |
|  | ✗ | 0.9798 | 0.0712 | 0.0334 | 0.0678 | 0.0099 | **0.00025** |
| $\mathcal{F}_{\text{WDCE}}$ | ✓ | 0.9204 | 0.0878 | 0.0397 | 0.0847 | 0.0450 | 0.01911 |
|  | ✗ | 0.9301 | 0.0814 | 0.0383 | 0.0841 | 0.0385 | 0.00932 |
| Baseline (MH) |  | / | 0.0667 | 0.0325 | 0.0628 | / | / |

## F.2 Preconditioning

Inspired by path integral sampler [ZC22], we propose to use the following form of preconditioning for learning the score model to sample from $\pi$:

$$\log \frac{\mathbb{P}_t^*(x^{i \leftarrow n})}{\mathbb{P}_t^*(x)} \approx \log s_\theta(x, t)_{i,n} := \Phi_\theta(x, t)_{i,n} + \sigma_\theta(t) \log \frac{\pi(x^{i \leftarrow n})}{\pi(x)}, \ \forall i \in \{1, ..., L\}^2, \ n \in \{\pm 1\} \backslash \{x^i\},$$

where $\Phi_\theta : \mathcal{X}_0 \times [0,1] \to \mathbb{R}^{(L \times L) \times 2}$ and $\sigma_\theta : [0,1] \to \mathbb{R}$ are free-form neural networks. For the Ising model (17),

$$\log \frac{\pi(x^{\sim i})}{\pi(x)} = -2\beta J \left( \sum_{j:j \sim i} x^j \right) x^i - 2\beta h x^i,$$

where $x^{\sim i}$ represents the configuration obtained by flipping the sign of $x^i$. Again, we emphasize that this design of preconditioning replies on the special structure of the target probability distribution, namely the closed-form expression of $\frac{\pi(x^{i \leftarrow n})}{\pi(x)}$ for all $i$ and $n$. The study of the preconditioning for more general target distributions is left as future work.

## F.3 Experiments

We use UDNS to learn the same $4 \times 4$ Ising model as in Tab. 2 (with $J = 1$, $h = 0.1$, and $\beta = 0.28$), and report the quantitative results in Tab. 6. We use the same model backbone and similar training configurations as described in App. D.1, except that (1) the total number of steps is 3000 to ensure all methods fully converge, and (2) for preconditioning, we parameterize $\sigma_\theta : [0,1] \to \mathbb{R}$ with a multilayer perceptron with hidden sizes $32, 32, 32$ and SiLU activation, whose total number of

parameters is 2k. The number of time points $K$ for discretization is chosen as 50 for both training and evaluation, and the function $\gamma$ is chosen as $\gamma(t) = \frac{1}{t}$, which implies $\overline{\gamma}(t) = \log \frac{1}{t}$ and $e^{-\overline{\gamma}(t)} = t$. The number of replicates $R$ used in $\mathcal{F}_{\mathrm{WDCE}}$ is set as 32. During training, we find that $\mathcal{F}_{\mathrm{CE}}$ fails to learn the correct target distribution, and hence its performance is not reported in the table. We can see from Tab. 6 that all three learning objectives are capable of generating samples with quality comparable to the ones generated from the MH algorithm, and the general effectiveness ranking of them is $\mathcal{F}_{\mathrm{LV}} > \mathcal{F}_{\mathrm{RERF}} > \mathcal{F}_{\mathrm{WDCE}}$. Moreover, we find surprisingly that applying preconditioning is generally beneficial when using $\mathcal{F}_{\mathrm{LV}}$, but is prone to worsen the performance when using $\mathcal{F}_{\mathrm{RERF}}$ and $\mathcal{F}_{\mathrm{WDCE}}$. However, compared with MDNS, UDNS has a higher computational cost in both training and evaluation due to the requirement of time-discretization, and hence is generally less preferable in practice.

