# OpenReview forum: "MDNS: Masked Diffusion Neural Sampler via Stochastic Optimal Control"
_NeurIPS.cc/2025/Conference — NeurIPS 2025 poster_

### Official Review · Reviewer_ctjr · 2025-06-24

**Clarity:** 3
**Significance:** 3
**Originality:** 3
**Rating:** 5
**Confidence:** 4

**Summary:**

The authors propose to leverage a Stochastic Optimal Control (SOC) of Continuous Time Markov Chains (CTMC) for solving the problem of sampling from an unnormalized density  $\pi(x) \propto exp(U(x))$ on a discrete state space. The authors state that sampling from an unnormalized density problem can be formulated as an SOC problem and propose computationally efficient methods to solve the corresponding SOC instance of the problem. In contrast to the previous approach for solving the SOC of CTMC, the authors eliminate differentiation through the sample generation trajectory and provide a total of 4 optimization objectives, i.e., REINFORCE trick-based, Log Variance Divergence-based, two Cross Entropy-based, that require only trajectory samples, but do not differentiate through them.

The algorithm for solving the SOC of CTMC for sampling with a Masked (and Uniform in Appendix F) prior process is proposed, called MNDS (and UNDS), and tested on a series of statistical physics-based benchmarks. The method is compared to the state-of-the-art neural sampler LEAPS [1] and learning-free Metropolis Hastings baseline, and shows itself on par on high temperature distributions and superior on lower temperature distributions.

**Questions:**

- It would be nice if the authors added LEAPS to the comparison in theTable 2.
- Why is WDCE loss used for experiments on 16x16 Ising and Potts models, while the LV loss showed itself to be the best on the training objectives study, see Sec 4.1?
- In the Proposition 2 main paper, there seems to be a missing expectation. I think it should be $\hat{Z} = \mathbb{E}_{X \sim P^u}e^{W^u(x)}$
- How do you ensure the memoryless condition $\int_0^T \gamma(t)dt=\infty$ in practice? In the case of masked process MNDS memoryless property seems obvious (since you start from a point), but for the UDLM (Appendix F), in my opinion, it is not obvious how to ensure this condition.

**Ethical Concerns:**

["NO or VERY MINOR ethics concerns only"]

**Final Justification:**

The authors have adressed my concerns, but the score of 5 is enough in my opinion. Overall good paper would like to see it accepted.

**Limitations:**

Would like to see the mention of method not being simulation-free in the limitations section.

**Paper Formatting Concerns:**

No major formatting issues in this paper,

**Quality:**

3

**Strengths And Weaknesses:**

Strengths:

- The proposed approach is quite simple and effective.
- SOC for CTMC development helps to bridge the gap between CTMC models and diffusion models and, in my opinion, is valuable for the CTMC community.
- The MNDS method seems to outperform other methods on lower temperature distributions.
- Theoretical guarantees on sampling quality and normalizing constant are shown.
- The paper is mainly well-written and easy to follow.
- Experimental evaluation is through with a lot of quantitative metrics, which is rare for an unnormalized density on a discrete state space sampler paper.

Weaknesses:

- In Section 3.2, when LogVar and CE losses are presented, it is hard to understand where exactly the stop gradient can be placed and why it is so. I think it should be explained more clearly.
- Would like to see the code in the sake of reproducibility, although sufficient experimental details are described in the Appendix.
- No information on the computational complexity of the method compared to LEAPS and MH. It would be nice if the authors provided computational complexity analysis and training/inference time comparisons.
- Performance seems a bit behind the MH on Ising 4x4, see Table 2.
- I can’t find the preconditioning strategy mentioned in the main text, only in the Appendix. I think it should be mentioned in the Sec 4.
- The model was evaluated only on the statistical physics benchmarks. It would be nice to see it’s performance on other type of problems, e.g., combinatorial optimization.

---

> ### Author Rebuttal · Authors · 2025-07-30
>
> We sincerely appreciate the reviewer's time and the constructive suggestions to help us improve the quality of this manuscript. We provide itemized responses below to each of them and hope they can help clarify your questions and concerns.
>
> > **Weakness 1: Stop gradient for LV and CE losses**
>
> We apologize for the potential confusion regarding the stop gradient operator. As we use $v=\overline{u}$ in both losses, we can write the LV loss as $\mathcal{F} _ \mathrm{LV}(\mathbb{P}^u,\mathbb{P}^ * )=\mathrm{Var} _ {\mathbb{P}^{\overline{u}}}\log\frac{\mathrm{d}\mathbb{P}^ * }{\mathrm{d}\mathbb{P}^u}$, which means we first use the current policy to sample a batch of trajectories $X^{(i)}\sim\mathbb{P}^u$, $1\le i\le B$, detach them from the computational graph, and then compute all $\log\frac{\mathrm{d}\mathbb{P}^ * }{\mathrm{d}\mathbb{P}^u}(X^{(i)})$ using Eq. (14), and finally do back propagation on the loss $\mathrm{Var}\left(\log\frac{\mathrm{d}\mathbb{P}^ * }{\mathrm{d}\mathbb{P}^u}(X^{(i)}):1\le i\le B\right)$. The same applies to the CE loss. The reason for applying the stop gradient operator is to ensure we compute the correct gradient of the loss, and to avoid taking gradient over the sampled trajectories $X\sim\mathbb{P}^u$ (which is not differentiable due to the discrete nature of the trajectories). We will provide a clearer explanation of this in the revised manuscript.
>
> > **Weakness 2**: Codes for reproducibility
>
> We understand the importance of reproducibility and appreciate that you asked. Unfortunately, due to NeurIPS policy, all the texts we post (rebuttal, discussion) cannot contain any links to external pages, and we are unable to upload supplemental materials at this stage. We will make our code publicly available when the anonymous period ends. Thank you for your understanding!
>
> > **Weakness 3**: Algorithm computational complexity compared to LEAPS and MH
>
> We describe the training details in line 886-891 for Ising model and line 1052-1057 for Potts model. The training time for $16\times16$ Ising model on an A100 is around 4 hours, and for $16\times16$ Potts model on an A100 is 20 hours. For generating ground truth samples, we run the MH algorithm for 3 hours to sample from $16 \times 16$ Ising model, and the SW algorithm for 2 hours to sample from $16 \times 16$ Potts model on CPU. For the MH baseline in Potts model, we run it for 30 hours on CPU. For all experiments with LEAPS, we run it for a time comparable to that of MDNS to ensure a fair comparison. We will add these information and discussion into the paper during revision.
>
> > **Weakness 4**: Performance on Ising 4x4
>
> We agree that the TV distance and KL and $\chi^2$ divergence of the samples generated by MDNS on $4\times 4$ Ising model are slightly larger than those from MH algorithm. However, we believe this is mostly due to the simplicity of the problem as well as the numerical errors. In general, MDNS performs comparably to MH on Ising models and outperforms it on Potts models (see Fig 2 and Tab 3).
>
> > **Weakness 5**: Preconditioning should be mentioned in the Sec 4
>
> Thank you for your suggestion. We will add a brief discussion in the main text regarding the exploration of the preconditioning techniques.
>
> > **Weakness 6**: Performance on other type of problems
>
> Thank you for the great suggestion! We will include results on combinatorial optimization problems, such as finding maximum cut and maximal independent set of graphs, in the revision.
>
> > **Question 1**: Add LEAPS to Table 2
>
> Thank you for your suggestion. The purpose of Table 2 was mainly to display the ablation results over different choices of training objectives, and therefore we do not include LEAPS in the comparison. We have compared with LEAPS on experiments with a larger scale and results can be found in Figures 1, 2 and Tables 1, 3. We hope these comparisons have sufficiently demonstrated that MDNS outperforms LEAPS in numerical performance.
>
> > **Question 2**: LV loss not used for 16x16 experiments
>
> The reason that we do not adopt $\mathcal{F} _ {\mathrm{LV}}$, $\mathcal{F} _ {\mathrm{CE}}$ or $\mathcal{F} _ {\mathrm{RERF}}$ for the high dimensional experiments is because they are too memory-intensive. These objectives share the similarity that they all requries using the full trajectory to estimate the objectives, making the loss computation requires at least $BD$ gradient-tracking score values in one step, where $B$ is the number of trajectories and $D$ is the length of the trajectory. For the $16 \times 16$ experiments, this would be unafforably expensive unless we pick extremely small values for $B$, which potentially would incur large variance detrimental to optimization. On the other hand, using $\mathcal{F} _ {\mathrm{WDCE}}$ is free of such concern and has a more controllable, affordable memory usage for high dimensional example. Therefore, we stick with only $\mathcal{F} _ {\mathrm{WDCE}}$ in the high-dimensional settings.
>
> In the revision, we will highlight more this difference in GPU memory consumption and explain clearly why only WDCE is used in large-scale experiments. Thank you for bringing up this question!
>
> > **Question 3**: Missing expectation in Proposition 2
>
> Thank you for carefully reading our paper. Following Eq. (14), we know that $Z=\mathbb{E} _ {X\sim\mathbb{P}^u}\mathrm{e}^{W^u(X)}$. Thus, we can sample $X\sim\mathbb{P}^u$, and use the *random variable* $\widehat{Z}=\mathrm{e}^{W^u(X)}$ as an unbiased estimator of $Z$. Therefore, we do not add an expectation here.
>
> > **Question 4**: Ensuring the memoryless condition for UDNS
>
> Indeed, we require $\int _ 0^1\gamma(t)\mathrm{d}t=\infty$ to ensure the memoryless condition (see line 1119-1121). For UDNS, as is discussed on line 1163, we choose $\gamma(t)=\frac{1}{t}$ in our experiments. In practice, we use early-stopping (which is a standard practice in diffusion model training) and only train on the interval $[\epsilon, 1]$. However, by choosing a reasonably small $\epsilon$, we can ensure the memoryless property to a satisfactory degree that does not affect the correctness of the algorithm.
>
> > **Limitation**: The method is not simulation-free
>
> Thank you for pointing this out. Just for us to better answer, may we clarify which "simulation-free" were referred to in this comment, as it could mean several things across different communities --
>
> - If being simulation-free refers to the algorithm does not requires differentiate through trajectories to perform optimization, then our algorithm is indeed simulation-free in this sense, as it is a necessary requirement for SOC of CTMCs.
> - If being simulation-free refers to the algorithm does not requries simulating the current policy to generate trajectories, then indeed our algorithm is not simulation-free, and we are happy to discuss this in the limitation section. However, we believe that requiring rolling out policy is a shared limitation of many of the neural samplers. We will commit to explore discrete neural samplers with less reliance on simulations in the future.
>
> ---
> Thank you again for your positive review and constructive feedback regarding the paper. We'd appreciate to discuss further if you have any additional questions or suggestions.

---

> > ### Comment · Reviewer_ctjr · 2025-08-01
> > **After Rebuttal answer**
> >
> > I sincerely thank the authors for taking the time to answer my questions and address my concerns. I am satisfied with most of the answers. Please incorporate the discussed changes and clarifications into the final version of the paper. In particular, I would be glad to see combinatorial optimization results, GPU consumption discussion and preconditioning mentioned. Though I will not raise my score since it is already quite high.

---

> > > ### Author Response · Authors · 2025-08-05
> > >
> > > Thank you again for your thorough and insightful review and your recommendation for acceptance. We are pleased to hear that our rebuttal has successfully clarified your questions and concerns. Your constructive feedback and thoughtful insights have helped improve the quality of the paper, and we appreciate your recognition of this work's contribution to the field of discrete neural sampler training. We will incorporate the clarifications and changes in the final version of the manuscript to reflect the fruitful discussion.
> > >
> > > Best regards,
> > >
> > > Authors

---

### Official Review · Reviewer_Qq8S · 2025-06-25

**Clarity:** 4
**Significance:** 4
**Originality:** 3
**Rating:** 5
**Confidence:** 4

**Summary:**

This paper considers the task of training a neural sampler without data, on discrete distributions by only using the energy function. This paper combines many approaches in continuous space and applies them to CTMC. This paper also proposed an efficient training objection that does not require simulation every gradient update. The authors provided comprehensive evaluation to demonstrate the efficientness of the proposed approach.

**Questions:**

1. I am not generally familiar with the setup of these systems. Therefore, I might be curious why some of the parameters are set in that way (for example, line 301)? Is it to make the system easy (like reducing phase transitions) or make it harder? Is it a convention?


2. I am a bit lost in the equation below line 202 (is there also a typo?). Could you please explain this to me?

**Ethical Concerns:**

["NO or VERY MINOR ethics concerns only"]

**Final Justification:**

The reply has addressed my concern and hence I will support for acceptance.

**Limitations:**

Yes,  the authors have adequately addressed the limitations and potential negative societal impact of their work.

**Quality:**

4

**Strengths And Weaknesses:**

## Strength:

1. The writing is clear, pleasant to read, and easy to follow. It is a pleasure to review and read this paper.

2. The paper not only proposed one approach. Instead, it provides a systematic recipe with several loss choices one can consider. While most of  these are not novel, it makes the paper complete.

3. The proposed simulation-free loss is quite interesting and effective.

4. Experiments are convincing. Comparing different losses is also helpful.



## Weakness:

1. I might suggest adding citations and discussion on previous works that use similar ideas to the buffer and weighted loss. As far as I know, FAB[1] and Adjoint Sampler[2]'s buffers have similar ideas. In FAB, one can do IS to reweight samples from the buffer. In AS, one collects buffer using current control and train the network with TSM.

2. By convention, one define potential /  Energy $U$ that $p \propto \exp(-U)$. However, in this paper, it was defined as $p \propto \exp(U)$ andm hence it might be beneficial to follow that notation. I do not think this is a weakness (but I do not have other places to put this).



[1] Midgley, Laurence Illing, et al. "Flow Annealed Importance Sampling Bootstrap." ICLR,

[2] Havens, Aaron, et al. "Adjoint sampling: Highly scalable diffusion samplers via adjoint matching." ICML.

---

> ### Author Rebuttal · Authors · 2025-07-30
>
> We sincerely appreciate the reviewer's time and the constructive suggestions to help us improve the quality of this manuscript. We provide itemized responses below to each of them and hope they can help clarify your questions and concerns.
>
> > **Weakness 1**: Related works using similar ideas to the buffer and weighted loss
>
> Thank you for pointing out related works. Our approach indeed shares similarities with these works, and we will definitely add a discussion in the related work section of a revised version.
>
> > **Weakness 2**: Notation on potential / energy
>
> Thank you for your suggestion regarding the notation. We will change it following the convention in the revised manuscript.
>
> > **Question 1**: Choice of hyperparameters
>
> These hyperparameters are chosen based on the nature of the target distributions. First, for each distribution in statistical physics that is considered in this work, there exists a critical inverse temperature $\beta _ \mathrm{critical}$ at which the distribution undergoes a phase transition. The target distributions demonstrate three largely different behaviors at different position across this temperature spectrum, making it an ideal testbed for our algorithms.
>
> Generally speaking, higher temperatures than the critical threshold lead to more uniform distributions and are thus easier to sample and learn from, while lower temperatures than the critical threshold add challenges in sampling and learning due to mode separation. For both statiscial physics models, we respectively consider the distribution below, at, and above the critical inverse temperature. For Ising model, the critical inverse temperature is $\beta _ {\mathrm{critical}} \approx 0.4407$. For Potts model with $3$-spin state, $\beta _ {\mathrm{critical}} \approx 1.005$.
>
> For system size, we mostly experiment on the square lattice of $16\times 16$, which incurs problems of a reasonable high dimension. This setup enables a comprehensive evaluation of algorithm performance. MDNS is successful on both examples across all three temperatures, suggesting its excellent robustness to hyperparameter settings as well as great scalability across dimensions.
>
> > **Question 2**: The equation below line 202
>
> In the equation below line 202, we are computing the sum of the non-diagonal elements in $Q _ t^u$, which shows up as in the term $\sum _ {y\ne X _ t}(Q _ t^u-Q _ t^0)(X _ t,y)$ in the RN derivative $\log\frac{\mathrm{d}\mathbb{P}^ * }{\mathrm{d}\mathbb{P}^u}$ in Eq. (14).
>
> The first equality is due to the fact that for $x\ne y$, $Q _ t^u(x,y)\ne0$ if and only if $x$ and $y$ only differ in one position (say, $d$), and $x^d=\mathbf{M}\ne y^d$. The second equality follows from the parameterization due to Eq. (3), where $Q _ t^u(x,x^{d\gets n})=\gamma(t)s _ \theta(x) _ {d,n}$ and $s _ \theta(x) _ {d,\cdot}\in\mathbb{R}^N$ is a probability vector. We will clarify the purpose of this equation as well as add a short explanation in the revised manuscript, and please do let us know if you have any further questions on this equation.
>
> ---
> Thank you again for your positive review and constructive feedback regarding the paper. We'd appreciate to discuss further if you have any additional questions or suggestions.

---

> > ### Comment · Reviewer_Qq8S · 2025-08-02
> >
> > Thank you for the clarification, which addressed my concern. Therefore, I will keep my score.  Best of luck.

---

> > > ### Author Response · Authors · 2025-08-05
> > >
> > > Thank you again for your thorough and insightful review and your recommendation for acceptance. We are pleased to hear that our rebuttal has successfully clarified your questions and concerns. Your constructive feedback and thoughtful insights have helped improve the quality of the paper, and we appreciate your recognition of this work's contribution to the field of discrete neural sampler training.
> > >
> > > Best regards,
> > >
> > > Authors

---

### Official Review · Reviewer_7ypT · 2025-07-02

**Clarity:** 3
**Significance:** 3
**Originality:** 4
**Rating:** 5
**Confidence:** 4

**Summary:**

This paper extends stochastic optimal control for continuous-time Markov chains (CTMCs) to discrete state spaces, introducing the Masked Diffusion Neural Sampler (MDNS)-–a neural sampler tailored to discrete domains. It proposes a weighted denoising cross-entropy loss, a scalable objective for training masked-diffusion models in sampling tasks. The theoretical framework generalizes existing continuous-space neural samplers to discrete settings, while a replay-buffer–based training algorithm improves computational efficiency. Experiments are comprehensive, benchmarking MDNS against both learning-based and classical non-learning Markov-chain methods across diverse probabilistic metrics.

**Questions:**

1. **Replay buffer**: Does it materially improve training? Have you explored prioritized replay?
2. **Trajectory length**: Can the algorithm scale to very long trajectories?
3. **CTMC justification**: Why is a CTMC formulation needed in discrete spaces?
4. **Variance control**: What techniques could you use to reduce training variance?

**Ethical Concerns:**

["NO or VERY MINOR ethics concerns only"]

**Final Justification:**

This paper bring interesting formulation based on stochastic control into discrete sampling and gives high performances. My questions and concerns are resolved and want to keep high score (5: accept).

**Quality:**

4

**Strengths And Weaknesses:**

**Strengths**

1. Strong theory and a solid contribution that extends CTMC-based stochastic optimal control to discrete spaces.
2. Introduces an efficient loss function and accompanying algorithm.

**Weaknesses**

1. The paper does not report how trajectory-level loss computation affects the variance of learning.
2. The importance weights may further amplify training variance, yet no variance-reduction estimator is provided.
3. Motivation is thin: why adopt a CTMC framework in discrete spaces when established approaches like GFlowNets already serve as effective discrete samplers without relying on CTMC dynamics?

---

> ### Author Rebuttal · Authors · 2025-07-30
>
> We sincerely appreciate the reviewer's time and the constructive suggestions to help us improve the quality of this manuscript. We provide itemized responses below to each of them and hope they can help clarify your questions and concerns.
>
> > **Weakness 1, 2 & Question 4**: Variance reduction of training objectives and importance weight estimators
>
> Thanks for pointing out the issue for variance reduction. Throughout our experiments, we observe that the variance of the loss is controlled and the training is almost always stable, thus we do not incorporate specific variance reduction techniques.
>
> One possible trick of variance reduction in the logits could be to leverage the following equivalent formulation of the WDCE loss inspired by [1], i.e.,
> $$\min _ \theta \operatorname{KL}(\mathbb{P}^ * ||\mathbb{P}^u) = \mathbb{E} _ {X\sim \mathbb{P}^{\overline{u}}}\frac{\mathrm{d}\mathbb{P}^ * }{\mathrm{d}\mathbb{P}^\overline{u}}(X)\mathbb{E} _ {m\sim\operatorname{Unif}\\{1,2,...,D\\}}\left[\frac{D}{m}\mathbb{E} _ {\mu' _ m(\widetilde{x}|X _ T)}\sum _ {d:\widetilde{x}^d=\mathbf{M}}-\log s _ \theta(\widetilde{x}) _ {d,x^d}\right],$$
> where $\mu' _ m(\widetilde{x}|x)$ means obtain $\widetilde{x}$ by uniformly selecting a subset of $\\{1,...,D\\}$ with size $m$ and mask the corresponding positions at $x$. In this way, by directly specifying the number of masks applied onto $x$ instead of independently masking each position of $x$ with probability $\lambda$, one can possibly reduce the variance of the loss, which has been observed in [2] for evaluating the log likelihood.
>
> Furthermore, one can also use the trick of *antithetic sampling*, i.e., if $m\ne D$, for each clean sample $x$ one can sample a pair of complementary masked positions whose intersection is empty and union is $\\{1,...,D\\}$, and compute the logits for these two masked versions of $x$. We believe that these two modifications would further reduce the variance in the logits.
>
> For the noise in importance weights, we agree that their variance may also amplify the variance in computing the loss. During training, we in fact self-normalize all $W^u(X^{(i)}),1\le i\le B$ with softmax in the batch to ensure numerical stability. This is equivalent to first using the weights to estimate the normalizing constant $Z$ then pluging it back into the obejctive.
>
> We will add more discussion of possible techniques for variance reduction and numerical stability improvement in the revised manuscript. Thank you for the insightful question!
>
> > **Weakness 3 & Question 3**: Motivation of using CTMC
>
> The use of CTMC originates from the fact that we adopt a **continuous-time** control pespective of the problem. While this is not always necesary to achieve our purpose, taking a continuous-time formulation makes it easier to compute objectives from the path measure perspective and use well-established stochastic analysis tools such as Girsanov's theorem. We believe that our algorithms can also be formulated under the use of a discrete-time dynamics on the finite state space, mimicking the case of GFlowNets. One major difference between our considered SOC approach and GFlowNets objectives is that ours does not require the estimation of normalizing constant during training, while GFlowNets training losses typically involve such process.
>
> > **Question 1**: Use of replay buffer and other choices
>
> The motivation of using replay buffer is to save the expensive computation costs of simulating the whole trajectories, rather than improve training. In fact, the optimal (yet most expensive) implementation of our algorithm would be not to use replay buffer and re-simulate trajectories for every optimization step, i.e., setting the resample frequency $k$ in Algorithm 1 as one so we are always using the samples from the current policy. Moreover, we only cache the final weights computed from each trajectory and discard the trajectory itself. We periodically drop all the buffered weights and resample new ones using current policy, without considering complicated buffer update algorithms. We did not explore prioritized replay as in fact the algorithms do not require the storage of transitions/trajectories, making prioritized replay largely inapplicable.
>
> > **Question 2**: Scaling to long trajectories
>
> Our algorithms are highly scalable for high dimensional problems, which incur long simulation trajectories. With the WDCE loss, the effectivenss of the training objective is not majorly influenced by the trajectory length. In the numerical experiments of this paper, we present successful results up to trajectory length of $16\times 16=256$, which is already decently long. We also have additional numerical results showing that the method remains effective for problems with trajectories length up to $24\times24=576$, which we will include in the revised manuscript. These empirical evidence suggests that our method can scale to long trajectories.
>
> ---
> Thank you again for your positive review and constructive feedback regarding the paper. We'd appreciate to discuss further if you have any additional questions or suggestions.
>
>
> ### References
>
> [1] Ou et al. Your Absorbing Discrete Diffusion Secretly Models the Conditional Distributions of Clean Data. ICLR 2025.
>
> [2] Nie et al. Large Language Diffusion Models. ArXiv 2502.09992.

---

> > ### Comment · Reviewer_7ypT · 2025-08-04
> >
> > Thanks for the responses. As my concerns are mostly address, I will keep my score positive.

---

> > > ### Author Response · Authors · 2025-08-05
> > >
> > > Thank you again for your thorough and insightful review and your recommendation for acceptance. We are pleased to hear that our rebuttal has successfully clarified your questions and concerns. Your constructive feedback and thoughtful insights have helped improve the quality of the paper, and we appreciate your recognition of this work's contribution to the field of discrete neural sampler training.
> > >
> > > Best regards,
> > >
> > > Authors

---

### Official Review · Reviewer_XE64 · 2025-07-05

**Clarity:** 2
**Significance:** 3
**Originality:** 3
**Rating:** 5
**Confidence:** 3

**Summary:**

The paper introduces a new diffusion sampler for discrete problems that is based on optimal control of a continuous-time Markov chain (CTMC). To do so, the authors first show that we can obtain a generator by minimizing the KL divergence between the optimal path distribution from a masked diffusion sampler and the generator. The authors discuss and compare different loss functions including REINFORCE loss, log-variance loss and cross entropy loss to solve the SOC problem. finally, they introduce a new weigthed cross entropy loss that is supposely computationally more efficient as it can reuse data. The authors compare their algorithm on challenging Ising models and potts models to a recent SOTA method (LEAPS) and show that it outperforms this SOTA method.

**Questions:**

- It should be clarified how the optimal control formulation for discrete samplers deviates from the standard optimal control formulation for continuous samplers
- From the study on the simpler 4x4 task, it seems that the log-variance loss is performing best. However, the paper proceeds with the weighted cross entropy loss (WCE). The advantage of the WCE to logVariance is unclear. Is it computation time due to data reuse? If so, that need to be evaluated as well
- Its unclear to me why we can not implement a data-reuse mechanism (e.g. with a similar importance weighting) for the other loss functions. Log-Variance loss seems to be better then cross entropy, so  the motivation why this was only done for the cross entropy loss is missing for me

**Ethical Concerns:**

["NO or VERY MINOR ethics concerns only"]

**Final Justification:**

Interesting new algorithm for discrete sampling with strong results. Main weakness was clarity, which was addressed by the authors.

**Limitations:**

yes

**Quality:**

3

**Strengths And Weaknesses:**

strength:
- interesting new algorithm for discrete sampling
- sound derivation (even though I could not fully follow everything) based on stochastic optimal control
- good experimental results with a clear improvement over SOTA on challenging problems:

weaknesses:
- clarity and motivation. The paper is hard to follow, notation is hard to grasp and it is unclear why certain things are introduced (e.g. Lemma 1). Moreover, basic definitions are hidden in equations and are quite hard to find (e.g. the defintion of W^u, that should be an own equation with W^u = ...) and no intution is given what this terms really mean.
- Results are not fully conclusive (see questions).

---

> ### Author Rebuttal · Authors · 2025-07-30
>
> We sincerely appreciate the reviewer's time and the constructive suggestions to help us improve the quality of this manuscript. We provide itemized responses below to each of them and hope they can help clarify your questions and concerns.
>
> > **Weakness 1**: Clarity and Motivation
>
> Thank you for raising the question and allow us to further clarify.
>
> - The main motivation of our work is to formulate the discrete neural sampler training problem as a path measure matching problem, which naturally connects to SOC for CTMC (Eq. (6)) and gives rises to other training objectives that are more favorable on discrete state spaces.
> - We introduce Lemma 1 as it is an important result for CTMC, which is also crucial for the derivation of our objectives. In Eq. (14), we use Lemma 1 to compute the log RN derivative between the optimal path measure $\mathbb{P}^ * $ and current path measure $\mathbb{P}^u$. We will update the manuscript to clearly reflect the use of Lemma 1 in such derivation to improve logical coherence.
> - As of the definition of $W^u$, we agree with the reviewer that it would certainly be clearer to definite it in one separate equation, and we will update it in the revision to clearly emphasize that. As for the meaning of $W^u(X)$, it is the log RN derivative up to an additional log normalizing constant, a weight factor associated with trajectory $X$ and policy $u$ that shows up ubiquitously in the formulation of the four introduced training objectives in this paper. We call it a weight (therefore denoted it with $W$) due to its connection to importance sampling and computation of the effective sample size (see Appendix D.2.1 and Eq. (24)).
>
> > **Question 1**: Connection to SOC for continuous samplers
>
> We discussed the connection of our method to the related literatures in neural samplers training and SOC in Appendix A. In fact, our formulation of neural sampler training as path measure matching is a general framework motivated by SOC and it indeed can be applied to continuous samplers. For example, when choosing $\mathcal{F}(\mathbb{P}^u, \mathbb{P}^ * ) = \operatorname{KL}(\mathbb{P}^u || \mathbb{P}^ * )$ and applied on continuous state space, we recover the training recipe of Path Integral Sampler [1]. Despite similar in a high-level, theoretical formulation, what distinguishes this work from the literature of SOC-based continuous samplers is the unique challenge of trajectory non-differetiability of CTMC on discrete state space, which renders many existing techniques on continuous samplers training inapplicable. Motivated by this, our work consider only losses that does not requires trajectory differetiability to numerically solve the problem.
>
> > **Question 2**:  Advantage of $\mathcal{F} _ {\operatorname{WDCE}}$ to $\mathcal{F} _ {\operatorname{LV}}$
>
> The motivation for introducing WDCE loss is to realize a more *scalable* training in problem with high dimensions, mostly in terms of GPU memory usage. We have explained this motivation in line 211-214 and 224-227.
>
> To be more specific, for training a discrete samplers of length $D$ with a batch of $B$ trajectories using the log variance loss $\mathcal{F} _ {\operatorname{LV}}$, the GPU memory usage is proportional to $BD$, as $\mathcal{F} _ {\operatorname{LV}}$ is a trajectory-level objective that requires backpropagating through all the gradient-tracking score values as is defined in Eq. (14). In high dimension scenarios, it's likely that $BD$ is significantly larger than $M$, the maximum possible number of gradient-tracking score values used in each loss evaluation (determined by the maximum GPU RAM). In an extremly high-dim case where $D > M$, $\mathcal{F} _ {\operatorname{LV}}$ will be completely inapplicable as it is unafforable to estimate the objective with even one trajectory.
>
> On the contrary, using $\mathcal{F} _ {\operatorname{WDCE}}$ is relatively free from such concern. The importance sampling factor $\exp(W^{\overline{u}})$ used in Eq. (16) is detached from the computation graph, and the number of gradient-tracking score values used in loss evaluation can be completely controlled by the remasking phase, and is not influenced of the problem dimension $D$. Besides, $\mathcal{F} _ {\operatorname{WDCE}}$ also increases the use rate of score network outputs and further amortize the computation cost through the use of replay buffers. In summary, $\mathcal{F} _ {\operatorname{WDCE}}$ is more memory-efficient and applicable even in super high-dim problem settings, and this promotes us to adopt it only for later experiments on $16\times 16$ distributions from statistical physics with $D = 256$.
>
> In the revision, we will further highlight this difference in GPU memory consumption and explain clearly why it drives us to only use $\mathcal{F} _ {\operatorname{WDCE}}$ in high-dim experiments. Thank you for bringing up this question!
>
> > **Question 3**: Importance Sampling for $\mathcal{F} _ {\operatorname{LV}}$ and $\mathcal{F} _ {\operatorname{RERF}}$
>
> Thank you for the interesting suggestion. We believe it is completely doable to implement data-reuse mechanism for other objectives as well. However, the main reason that we did not discuss them in this work is that importance-weighted versions of $\mathcal{F} _ {\operatorname{LV}}$ and $\mathcal{F} _ {\operatorname{RERF}}$ do not connect to simulation-free training of discrete diffusion models, which is the DCE loss described in Eq. (4). The main benefit of $\mathcal{F} _ {\operatorname{WDCE}}$ is that the simulation-free training with cached importance weights allows a flexible choice of batch size without worrying about being bounded by the problem dimension (see response to previous point). $\mathcal{F} _ {\operatorname{LV}}$ and $\mathcal{F} _ {\operatorname{RERF}}$ do not seem to enjoy similar connections despite the introduction of data-reuse scheme.
>
> This indeed brings up a new question of what is the optimal choice for the sampling path measure in the framework. In this work, we mainly explore the on-policy scenario with sampling path measure generated by the current policy $\overline{u}$. We believe that using importance-weighting to implement the choice of $\mathbb{P}^ * $ as the sampling path measure would be an intriguing direction to explore for future investigations.
>
> ---
> Thank you again for your positive review and constructive feedback regarding the paper. We'd appreciate to discuss further if you have any additional questions or suggestions.
>
>
> ### References
>
> [1] Zhang and Chen. Path integral sampler: a stochastic control approach for sampling. ICLR 2023.

---

> > ### Comment · Reviewer_XE64 · 2025-08-05
> >
> > Thanks for the detailed reply. My main concerns have been addressed and I agree with other reviewers that the paper should be accepted. Hence, I will raise my score.

---

> > > ### Author Response · Authors · 2025-08-05
> > >
> > > Thank you again for your thorough and insightful review and your recommendation for acceptance. We are pleased to hear that our rebuttal has successfully clarified your questions and concerns, and that you will raise the score. Your constructive feedback and thoughtful insights have helped improve the quality of the paper, and we appreciate your recognition of this work's contribution to the field of discrete neural sampler training. We will incorporate your constructive suggestions in the final version of the manuscript to reflect the fruitful discussion.
> > >
> > > Best regards,
> > >
> > > Authors

---

### Decision · Program_Chairs · 2025-09-17

**Decision:**

Accept (poster)

**Comment:**

This paper studies diffusion-based samplers of discrete distributions given by an unnormalised probability mass functions. Generalising the long line of existing work on this problem in the continuous setting, objectives for enforcing time reversal of the generative and the forward noising process -- now continuous-time, discrete-space Markov chains -- are derived and some methods for practical training are introduced. This sampler is evaluated on sampling two synthetic distributions from statistical physics (Isings and Potts models).

While the experiments are quite basic, the reviewers appreciate the theoretical/algorithmic contribution and give a unanimous Accept rating, and I agree with their assessment. However, it is strongly recommended that the authors address the following points in the revision:
- Clarify the connection to prior work on sampling via stochastic control in continuous space (e.g., PIS as mentioned by XE64, DIS from reference [11] in the paper).
- Clarify the connection to similar work on amortised sampling with discrete diffusion models (e.g., 7ypT mentions GFlowNet objectives, which have been used for discrete diffusion models -- for example, in [Venkatraman et al., NeurIPS'24](https://arxiv.org/abs/2405.20971) -- and can be trained with the LV objective).
- Improve the presentation of the method, taking reviewers' suggestions and minor corrections into account.
- Include discussion of complexity/cost in comparison to baselines.